# GUI-WORLD: A Dataset for GUI-Orientated Multimodal Large Language Models

## Abstract

Recently, Multimodal Large Language Models (MLLMs) have been used as agents to control keyboard and mouse inputs by directly perceiving the Graphical User Interface (GUI) and generating corresponding code. However, current agents primarily exhibit excellent understanding capabilities in static environments and are predominantly applied in relatively simple domains, such as Web or mobile interfaces. We argue that a robust GUI agent should be capable of perceiving temporal information on the GUI, including dynamic Web content and multi-step tasks. Additionally, it should possess a comprehensive understanding of various GUI scenarios, including desktop software and multi-window interactions. To this end, this paper introduces a new dataset, termed GUI-WORLD, which features meticulously crafted Human-MLLM annotations, extensively covering six GUI scenarios and eight types of GUI-orientated questions in three formats. We evaluate the capabilities of current state-of-the-art MLLMs, including ImageLLMs and VideoLLMs, in understanding various types of GUI content, especially dynamic and sequential content. Our findings reveal that ImageLLMs struggle with dynamic GUI content without manually annotated keyframes or operation history. On the other hand, VideoLLMs fall short in all GUI-orientated tasks given the sparse GUI video dataset. Based on GUI-WORLD, we take the initial step of leveraging a fine-tuned VideoLLM as a GUI agent, demonstrating an improved understanding of various GUI tasks. However, due to the limitations in the performance of base LLMs, we conclude that using VideoLLMs as GUI agents remains a significant challenge. We believe our work provides valuable insights for future research in dynamic GUI content understanding.

## 1 Introduction

Multimodal Large Language Models (MLLMs), such as GPT-4V(ision) [1] and LLaVA [2], have significantly contributed to the development of the visual-text domain [3]. These models bring forth innovative solutions and paradigms for traditional visual tasks, including visual reasoning [4], medical image interpretation [5, 6], and applications in embodied agents [7]. One particularly promising area is Graphical User Interface (GUI) understanding, which holds significant potential for real-world applications, such as webpage comprehension [8, 9] and navigation by GUI agents [10–12]. The key challenges of GUI understanding are twofold: effective GUI agents are expected to (1) possess a deep understanding of GUI elements, including webpage icons, text identified through Optical Character Recognition (OCR), and page layouts, and (2) exhibit an exceptional ability to follow instructions within GUI contexts, such as conducting searches through search engines.

Submitted to the 38th Conference on Neural Information Processing Systems (NeurIPS 2024) Track on Datasets and Benchmarks. Do not distribute.

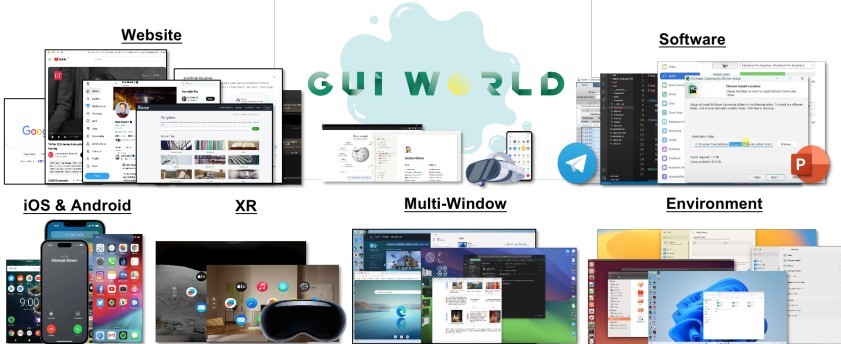

Figure 1: GUI-WORLD: a comprehensive dataset for GUI understanding, holding significant potential for real-world applications.

Despite significant progress, as illustrated in Table 1, existing works suffer from the following limitations: (1) Most studies predominantly focus on the static features of GUI scenarios, neglecting the need for MLLMs to effectively process sequential information and dynamic operations. For instance, an agent's task performance can be disrupted by unexpected elements such as pop-up advertisements, underscoring a gap in handling dynamic sequential tasks. (2) Current research is typically restricted to Web-based environments, which limits the models' generalization and robustness. For instance, GUI agents may need to operate across diverse platforms such as Windows, macOS, Linux, iOS, Android, and XR environments. Additionally, operations may sometimes involve multiple windows. Therefore, expanding the scope of research to encompass these varied environments will enhance the adaptability and effectiveness of GUI agents.

Table 1: Comparison of GUI datasets. 'Sem.': semantic instruction level, 'Seq.': Tasks for sequential images, 'Cro.': Cross-app or multi-window tasks, 'Dyn.': Tasks for dynamic GUI content.

| Dataset | Size | Sem. | VL | Video | Env Type | | | | Task Coverage | | | Task |
|---|---|---|---|---|---|---|---|---|---|---|---|---|
| | | | | | Web. | Mob. | Desk. | XR | Seq. | Cro. | Dyn. | |
| Rico [13] | 72,219 | Low | ✔ | ✔ | ✗ | ✔ | ✗ | ✗ | ✔ | ✔ | ✗ | UI Code/Layout Generation |
| MetaGUI [14] | 1,125 | Low | ✔ | ✗ | ✗ | ✔ | ✗ | ✗ | ✔ | ✗ | ✗ | Mobile Navigation |
| UGIF [15] | 523 | High | ✔ | ✗ | ✗ | ✔ | ✗ | ✗ | ✔ | ✗ | ✗ | UI Grounded Instruction Following |
| AITW [16] | 715,142 | High | ✔ | ✗ | ✗ | ✔ | ✗ | ✗ | ✔ | ✔ | ✗ | GUI Understanding |
| Ferret-UI [17] | 123,702 | Low | ✔ | ✗ | ✗ | ✔ | ✗ | ✗ | ✗ | ✗ | ✗ | UI Grounding & Understanding |
| MiniWoB++ [18] | 100 | Low | ✔ | ✗ | ✔ | ✗ | ✗ | ✗ | ✗ | ✗ | ✗ | Web Navigation |
| WebArena [19] | 812 | Low | ✔ | ✗ | ✔ | ✗ | ✗ | ✗ | ✔ | ✗ | ✗ | Web Navigation |
| Mind2Web [20] | 2,350 | High | ✔ | ✔ | ✔ | ✗ | ✗ | ✗ | ✔ | ✗ | ✗ | Web Navigation |
| OmniAct [21] | 9,802 | Low | ✔ | ✗ | ✔ | ✗ | ✔ | ✗ | ✔ | ✗ | ✗ | Code Generation |
| MMINA [22] | 1,050 | Low | ✔ | ✗ | ✔ | ✗ | ✗ | ✗ | ✔ | ✔ | ✗ | Web Navigation |
| AgentStudio [23] | 304 | High | ✔ | ✗ | ✔ | ✗ | ✔ | ✗ | ✔ | ✔ | ✗ | General Control |
| OSWorld [24] | 369 | High | ✔ | ✗ | ✔ | ✗ | ✔ | ✗ | ✔ | ✔ | ✗ | General Control |
| **GUI-WORLD (Ours)** | 12,379 | Both | ✔ | ✔ | ✔ | ✔ | ✔ | ✔ | ✔ | ✔ | ✔ | GUI Understanding Instruction Following |

To mitigate these gaps, this paper introduces GUI-WORLD, a comprehensive dataset containing over 12,000 GUI videos, specifically designed to evaluate and enhance the capabilities of GUI agents. This dataset encompasses a wide range of GUI scenarios, including popular websites, desktop and mobile applications across various operating systems, multi-window interactions, as well as XR environments. The data collection process involves sourcing GUI videos from screen recordings and instructional videos on YouTube. Subsequently, we utilize an Human-MLLM collaborative approach to generate a diverse set of questions and instructions and finally construct GUI-WORLD.

Likewise, we also establish a comprehensive benchmark for GUI understanding, which encompasses seven mainstream MLLMs, three keyframe selection strategies, six GUI scenarios, and a diverse array of queries in multiple-choice, free-form, and conversational formats, aiming to provide a thorough evaluation of the MLLMs' GUI-orientated capabilities. The assessment results indicate that most MLLMs struggle with GUI-WORLD, highlighting their limited dynamic understanding of graphical interfaces and underscoring the need for further enhancement.

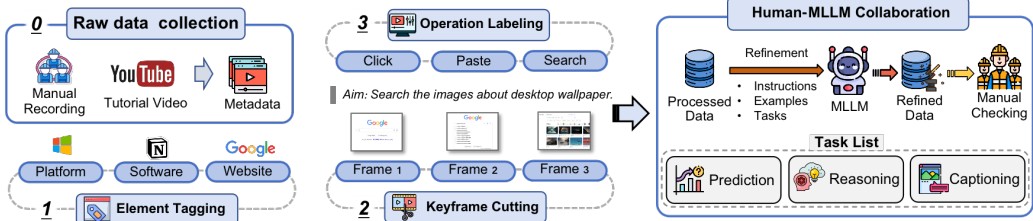

Figure 2: An overview construction pipeline of GUI-WORLD.

Leveraging this dataset, we take the first step of fine-tuning a Video GUI Agent proficient in dynamic and sequential GUI tasks, which results in significant improvements in the general capabilities of GUI agents, thereby demonstrating the utility and effectiveness of GUI-WORLD. Additionally, we delve into discussing various factors critical to GUI understanding, including the integration of textual information, the number of keyframes, and image resolutions.

Overall, the key contributions of this paper are three-fold:

▷ **A New Dataset.** We propose GUI-WORLD, a comprehensive GUI dataset comprising over 12,000 videos specifically designed to assess and improve the GUI understanding capabilities of MLLMs, spanning a range of categories and scenarios, including desktop, mobile, and extended reality (XR), and representing the first GUI-oriented instruction-tuning dataset in the video domain.

▷ **A Novel Model.** Based on GUI-WORLD, we propose GUI-Vid, a GUI-orientated VideoLLM with enhanced capabilities to handle various and complex GUI tasks. GUI-Vid shows a significant improvement on the benchmark and achieves results comparable to the top-performing models.

▷ **Comprehensive Experiments and Valuable Insights.** Our experiments indicate that most existing MLLMs continue to face challenges with GUI-oriented tasks, particularly in sequential and dynamic GUI content. Empirical findings suggest that improvements in vision perception, along with an increase in the number of keyframes and higher resolution, can boost performance in GUI-oriented tasks, thereby paving the way for the future of GUI agents.

## 2 GUI-WORLD: A Comprehensive Dataset for GUI Understanding

### 2.1 Overview

We introduce GUI-WORLD, a comprehensive dataset covering six GUI scenarios including video, human-annotated keyframes, as well as detailed captions and diverse types of QA produced by our data curation framework, aiming at benchmarking and enhancing the general GUI-orientated capabilities. These GUI scenarios encompass desktop operating systems (*e.g.*, macOS, Windows) and mobile platforms (*e.g.*, Android and iOS), websites, software, and even extended-range technologies (XR) (*e.g.*, GUI in Apple Vision Pro [25]). Discussion for each scenario are in subsection B.1.

As illustrated in Figure 2, the development of GUI-WORLD is structured around a two-stage process. Details regarding video and query statistics are provided in Table 2, which includes distributions of the number of keyframes, video lengths, and the lengths of queries and their corresponding golden answers, as displayed in Figure 3. Refer to Appendix G for details in the case study.

### 2.2 GUI Video Collection and Keyframe Annotation Process

We describe the pipeline for collecting screen recordings from student workers and GUI-related instructional videos from YouTube for GUI-WORLD and the procedures followed to convert these videos into keyframe sequences.

A significant portion of our video data is derived from screen recordings executed by student workers, which can directly reflect real-life GUI usage scenarios. A typical video collection scenario involves assigning a student worker a specific software task. The student begins by familiarizing themselves

Table 2: The statistics of GUI-WORLD. For Android, we automatically sample 10 frames. *Avg. Frame* refers to the average number of frames in each keyframe, and *Avg. Anno.* refers to the average number of manually annotated user actions in each keyframe.

| Category | Total Videos | Free-form | MCQA | Conversation | Total Frame. (Avg.) | Avg. Anno. |
|---|---|---|---|---|---|---|
| Software | 4,720 | 27,840 | 9,440 | 9,440 | 23,520 (4.983) | 7.558 |
| Website | 2,499 | 14,994 | 4,998 | 4,998 | 15,371 (6.151) | 6.862 |
| IOS | 492 | 2,952 | 984 | 984 | 2,194 (4.459) | 7.067 |
| Multi | 475 | 2,850 | 950 | 950 | 2,507 (5.277) | 7.197 |
| XR | 393 | 2,358 | 786 | 786 | 1,584 (4.030) | 10.970 |
| Android | 3,800 | 15,199 | 7,600 | 7,600 | 38,000 (10.000) | - |
| Summary | 12,379 | 76,673 | 24,758 | 24,758 | 83,176 (6.719) | 7.463 |

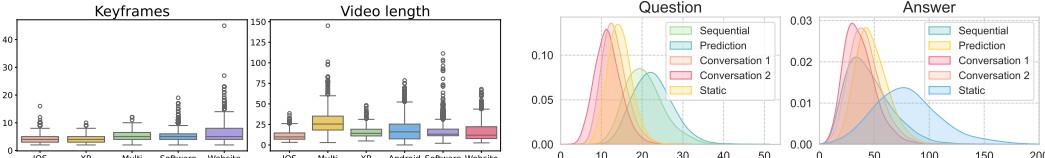

Figure 3: Left: Distribution of the number of keyframes and video lengths. Right: Length distribution for each type of question and its golden answer.

with the software, followed by recording a series of operations in a short video clip, such as "Sign up", "Sign in", "Create a New Page", and "Invite Other Collaborators" in the software "Notion[1]".

Despite the high fidelity of these manually recorded videos, we encounter several challenges: (1) Student workers often require substantial time to acquaint themselves with professional software (*e.g.*, MATLAB, Adobe After Effects (Ae)), which can hinder the progress of data collection. (2) The videos may lack comprehensiveness, typically capturing only commonly used operations and overlooking rarer functions crucial for dataset completeness. To address these issues, we also source videos from social media platforms that host a diverse array of GUI-related content. Specifically, we download tutorial videos from YouTube—given its prevalence as a video-sharing platform—because they richly detail various GUI operations. These videos are then segmented into shorter clips, each representing a distinct sequence of operations.

The subsequent step involves annotating these video clips with keyframes and textual descriptions of each keyframe using custom-designed annotation software. Although several algorithms exist for keyframe extraction [26–29], they typically underperform with GUI videos where changes between frames might be minimal (*e.g.*, a slight movement in the mouse cursor). To ensure high-quality datasets, we therefore perform manual extraction of these keyframes. Each keyframe is meticulously annotated to include details such as the operation performed, the purpose between two keyframes, the software or website used, mouse actions (*e.g.*, scroll, click), and keyboard inputs (*e.g.*, copy (Ctrl + C), paste (Ctrl + V), specific input). We detail our annotation process in subsection B.3.

## 2.3 GUI Tasks Generation from Human-MLLM Collaboration

Drawing insights from prior research [30–34], we develop a Human-MLLM collaboration pipeline to annotate captions and diverse types of QA specifically tailored for GUI comprehension. The process involves inputting an instructional prompt, a comprehensive description, key information (*e.g.*, system or application), and a sequence of human-annotated keyframes into GPT-4V. As depicted in Table 3, GUI-WORLD features an array of question types, as detailed in follows:

▷ **Detailed and Summarized Captioning:** This task challenges basic GUI knowledge and multimodal perception, also addressing the deficiency of detailed GUI content in video-caption pairs. Initially, GPT-4V generates two distinct descriptions for each video: one concentrating on fine-grained details and the other on the overall image sequences. Furthermore, GPT-4V provides a succinct summary, highlighting core operations and overarching objectives in the video.

---

[1]https://www.notion.so/

Table 3: Examples of diverse question types in GUI-WORLD.

| T. | Question | Examples |
|---|---|---|
| Caption | Detailed Description | **Q:** *Please provide a detailed description of what occurs throughout these sequential GUI images.* 
 **A:** The video shows a user taking the 16 Personalities test on a Windows desktop using the Edge browser... |
| Caption | Summarized Caption | **Q:** *Write a clear description of the video, make sure the key features are well covered.* 
 **A:** Creating a new IT team in Todoist by selecting industry, job function, role, team size, and inviting members. |
| Static | Layout, Icon Retrieval | **Q:** *What related searches are suggested on the right side of the Bing results for 'emnlp 2024'?* 
 **A:** The suggested related searches shown include 'emnlp 2024 miami', 'eacl 2024 call for papers'... |
| Static | Textual Retrieval | **Q:** *What is the estimated time to complete the content for Week 2 of the course?* 
 **A:** The estimated time to complete the content for Week 2 of the course is 1 hour... |
| Static | Interrelations in GUI Content | **Q:** *What is the name of the browser and the tab where the user performs the product search?* 
 **A:** The browser is Microsoft Edge, and the user performs the product search in the eBay tab. |
| Dynamic | Content Retrieval | **Q:** *What specific action does the user take after turning their head to the left to view the left side of the page?* 
 **A:** After turning their head to the left to view the left side of the page, the user performs... |
| Dynamic | Prediction | **Q:** *Given the mouse is over 'Add NeurIPS 2024 DB Track Submission,' what's the likely next step?* 
 **A:** It would be to click on the 'Add NeurIPS 2024 Datasets and Benchmarks Track Submission' button... |
| Dynamic | Sequential Reasoning | **Q:** *Scrolls down from the 'Moon Gravity', which of the following cheats? A. Change Weather B. Skyfall ...* 
 **A:** [[B]] |

▷ **Static GUI Content:** This task challenges MLLM with textual, layout, and iconographic analysis of static GUI content. We instruct GPT-4V to generate free-form queries with a golden answer concerning static GUI elements or specific scenes that recur in more than two keyframes, ensuring their consistent presence in the video. Additionally, GPT-4V also crafts QA pairs that evaluate inferential skills in static content, focusing on interrelations among icons or textual information.

▷ **Dynamic and Sequential GUI Content:** This task concentrates on temporal content in GUI video, such as dynamically changing interfaces, and aims to elucidate the sequential information and reasoning chains within GUI content. We direct GPT-4V to identify consistently changing elements to create queries for dynamic content. Moreover, predictive tasks are formulated on order and temporal relation in provided sequential images, challenging agents to anticipate future events or states.

In the last stage, human annotators will follow the guideline in subsection B.3 and carefully review the entire video and MLLM-generated QA pairs to correct inaccuracies and hallucinations, as well as supplement information for both questions and answers to make these tasks more challenging.

## 3   Progressive Enhancement on GUI Perception Ability

We introduce our strategy to enhance the GUI-oriented capabilities of current MLLMs on both static and dynamic GUI content. Inspired by previous studies [9, 35], we structure our methodology into two distinct fine-tuning stages, as illustrated in Figure 4. Initially, we fine-tune the MLLM on simpler tasks, such as description queries and captioning exercises, to instill a basic understanding of GUI elements. Subsequently, building on this foundation, the second stage aims to augment the MLLM's proficiency with more complex and challenging tasks. Our fine-tuning is all based on the Supervised Fine-Tuning (SFT): $\mathcal{L}_{\text{SFT}}(\pi_\theta) = -\mathbb{E}_{(x,y)\sim\mathcal{D}}[\log \pi_\theta(y \mid x)]$, where $x$ is the input, $y$ is LLMs' output, and $\pi_\theta$ denotes the model parameters that need to be optimized.

**Stage-1: Learning Preliminary for GUI Content.**   The initial phase focuses on aligning GUI content with a pre-trained vision encoder and a base LLM, utilizing GUI videos accompanied by detailed descriptions and captions. This phase aims to embed a robust understanding of fundamental GUI concepts and terminology within the MLLM. By engaging the model in basically captioning various GUI components, the model learns to recognize and articulate the functionalities and visual characteristics of these elements, thereby laying a solid groundwork for GUI knowledge.

**Stage-2: Mastering Advanced GUI Capability.**   Building on the foundational knowledge established in Stage 1, the second stage focuses on advancing the MLLM's proficiency in interacting with GUI elements through more complex tasks. These tasks are designed to simulate real-world scenarios that the MLLM might encounter in GUI environments, which include predicting based on

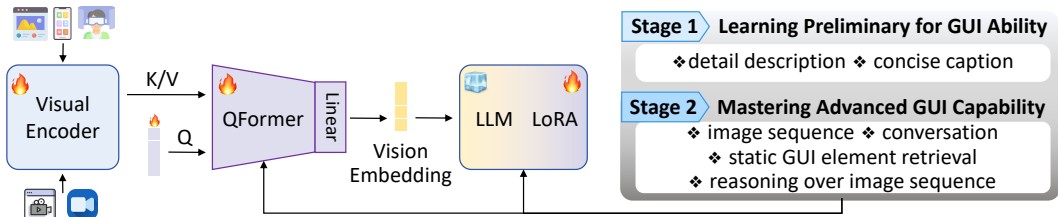

Figure 4: An overview of our fine-tuning architecture, focusing on GUI content alignment and instruction tuning.

image sequences, engaging in conversations, retrieving both static and dynamic GUI elements, and performing reasoning tasks.

As illustrated in Figure 4, We employ the two-stage training architecture utilizing VideoChat2 [35] as our foundational model. Initially, videos and images are encoded using the UMT-L visual encoder [36]. Subsequently, a QFormer compresses visual tokens into a smaller set of query tokens. Drawing inspiration from [37], we enhance the QFormer [38] by integrating instructions to enable it to extract visual representations pertinent to the given instructions. Additionally, we apply low-rank adaptation (LoRA [39]) to base LLM. This model is concurrently fine-tuned with the visual encoder and QFormer using a Vision-grounded Text Generation (VTG) loss: $\mathcal{L}_{\text{VTG}}(\theta) = -\mathbb{E}\left[\log p(y|v;\theta)\right]$, where $v$ represents the visual tokens derived from the QFormer, and $y$ represents the text output grounded in the visual context.

# 4 Experiments and Analysis

## 4.1 Experimental Setups

**Models.** We conduct evaluations on four of the most robust image-based MLLMs: GPT-4V(ision) [1], GPT-4o [40], Qwen-VL-Max [41], and Gemini-Pro-1.5 [42]. We benchmark on three keyframe selection settings: (1) *Random*, where frames are sampled at fixed time intervals within a video; (2) *Extracted*, with keyframes extracted using Katna[2]; and (3) *Human*, where keyframes are selected by humans during the annotation process. For the *Random* and *Extracted* settings, we input 10 frames into each MLLM, while the *Human* setting uses an average of 6.719 frames, as detailed in Table 2. Each model's responses employ a three-step Chain-of-Thought (CoT) [43] process, i.e., "Describe-Analyze-Answer", to evaluate their peak performance. Additionally, we assessed three advanced VideoLLMs—ChatUnivi [44], Minigpt4-video [45], and Videochat2 [46]—for their performance on GUI content. For detailed experimental setups are referred to Appendix D.

**Evaluation Metrics.** To assess free-form questions and multiple-round conversations, we utilize the LLM-as-a-Judge methodology, which assigns a similarity score ranging from 1 to 5 between MLLM's response and a predefined golden answer, already validated by previous studies[47–49]. For a comprehensive evaluation, we also provide BLEU [50] and BERTScore [51] in Appendix E. For multiple-choice questions, we measure performance using accuracy as the primary evaluation metric.

**Textual Information Integration.** To investigate the effectiveness of integrating image-caption models to enlarge the context window for LLMs—typically employed in natural videos—and the helpfulness of GUI history content in accomplishing GUI-oriented tasks, we implement three experimental settings: Detailed Caption, Concise Caption, and Vision + Detailed Caption. GPT-4V is utilized to provide captions of these keyframes, integrating human annotators' operational intents to more accurately describe each frame, being validated in subsection B.3.

**Keyframes and Resolution.** To explore the upper bound of GUI-orientated capabilities, particularly in dynamic and sequential tasks, we conduct ablation studies focusing on the impact of the number

---
[2]https://github.com/keplerlab/katna

Table 4: The overall performance in six GUI scenarios for MACQ and Free-form queries. 'D.C.' means detailed caption, and 'C.C.' means concise caption. 'R.', 'E.', and 'H.' denote random-selected, programmatic-selected, and human-selected keyframes, respectively. 'MC' means Multiple-Choice QA and 'Free' represents the average score of all free-form and conversational queries.

| Models | Setting | Software | | Website | | XR | | Multi | | IOS | | Android | | Avg. | |
|---|---|---|---|---|---|---|---|---|---|---|---|---|---|---|---|
| | | MC | Free | MC | Free | MC | Free | MC | Free | MC | Free | MC | Free | MC | Free |
| Gemini-Pro-1.5 | R. | 81.7% | 3.339 | 82.6% | 3.452 | 81.2% | 3.154 | 81.2% | 2.959 | 82.0% | 3.213 | 81.6% | 3.220 | 81.7% | 3.223 |
| | E. | 78.5% | 3.152 | 77.8% | 3.215 | 80.8% | 3.006 | 71.8% | 2.777 | 79.3% | 3.007 | 78.5% | 3.168 | 77.8% | 3.054 |
| Qwen-VL-Max | R. | 74.9% | 2.676 | 76.9% | 2.656 | 74.2% | 2.469 | 68.8% | 2.432 | 75.4% | 2.779 | 73.7% | 2.309 | 74.0% | 2.553 |
| | E. | 74.3% | 2.624 | 75.8% | 2.627 | 69.0% | 2.499 | 64.8% | 2.362 | 77.4% | 2.659 | 65.8% | 2.277 | 71.2% | 2.508 |
| | H. | 75.8% | 2.651 | 75.5% | 2.698 | 77.6% | 2.373 | 66.9% | 2.490 | 74.3% | 2.633 | - | - | 74.0% | 2.569 |
| GPT-4V | R. | 81.5% | 3.589 | 80.9% | 3.648 | 80.6% | 3.200 | 75.0% | 3.452 | 82.5% | **3.614** | 78.3% | 3.515 | 79.8% | 3.503 |
| | E. | 85.1% | 3.407 | 80.1% | 3.433 | 81.8% | 2.892 | 81.9% | 3.219 | **86.4%** | 3.427 | 79.9% | 3.176 | 82.6% | 3.259 |
| | H. | 86.0% | 3.520 | 79.8% | 3.655 | 83.4% | 3.265 | 76.9% | 3.449 | 79.9% | 3.453 | - | - | 81.2% | 3.469 |
| | D.C. | 85.0% | 3.350 | 83.1% | 3.380 | 82.3% | 3.056 | **84.2%** | 3.358 | 81.6% | 2.751 | 81.7% | 3.427 | 83.0% | 3.316 |
| | C.C | 80.7% | 3.028 | 72.2% | 3.025 | 82.8% | 2.809 | 81.3% | 3.160 | 76.5% | 2.868 | 76.4% | 2.939 | 78.3% | 2.971 |
| | H.+D.C. | 82.5% | 3.494 | 83.2% | 3.682 | **85.9%** | 3.191 | 83.9% | 3.617 | 80.9% | 3.516 | 84.9% | **3.758** | 83.5% | 3.543 |
| GPT-4o | H. | **86.5%** | **3.644** | 83.3% | **3.740** | 84.3% | **3.285** | 81.1% | **3.654** | 83.3% | 3.558 | **90.0%** | 3.561 | **84.8%** | **3.573** |
| ChatUnivi | - | 28.4% | 2.389 | 22.2% | 2.349 | 20.6% | 2.161 | 17.5% | 2.275 | 22.6% | 2.337 | 23.0% | 2.390 | 22.4% | 2.317 |
| Minigpt4Video | - | 18.9% | 1.475 | 15.3% | 1.520 | 16.3% | 1.362 | 15.4% | 1.457 | 20.1% | 1.501 | 14.6% | 1.342 | 16.8% | 1.443 |
| VideoChat2 | - | 45.5% | 2.144 | 42.6% | 2.221 | 44.0% | 2.005 | 40.4% | 2.222 | 40.2% | 2.169 | 44.7% | 2.119 | 42.9% | 2.147 |
| **GUI-Vid** | - | 59.9% | 2.847 | 54.1% | 2.957 | 55.6% | 2.764 | 52.9% | 2.861 | 51.8% | 2.773 | 53.4% | 2.572 | 54.6% | 2.796 |

of keyframes and image resolutions. We vary the number of keyframes (8, 16) fed into GUI-Vid. Additionally, we test the effect of different image resolutions on GPT-4o, using both low and high settings, to further assess how resolution influences performance.

## 4.2 Empirically Results

**Commercial ImageLLMs outperform Open-source VideoLLMs in Zero-shot Settings.** Commercial ImageLLMs, notably GPT-4V and GPT-4o, consistently outperform open-source VideoLLMs in zero-shot settings. As detailed in Table 4, GPT-4o exhibits superior performance across all GUI scenarios in complex tasks, reflected in its high scores in both multiple-choice and free-form queries, with an average of 84.8% and 3.573. Similarly, Gemini demonstrates strong capabilities in captioning and descriptive tasks within software and iOS environments, scoring 2.836 and 2.936, respectively, as shown in Table 13. Further analysis (Figure 5) reveals that GPT-4V excels in applications with minimal textual content and simple layouts, such as TikTok, health apps, and GitHub. In contrast, its performance drops in more intricate applications like Microsoft ToDo and XR software. As for VideoLLMs, their significantly poorer performance is attributed to two main factors: their inability to accurately interpret GUI content from user inputs and a lack of sufficient GUI-oriented pretraining, which is evident from their inadequate performance in basic captioning and description tasks. See Appendix E for BLEU and BERTScore, as well as detailed performance for complex tasks.

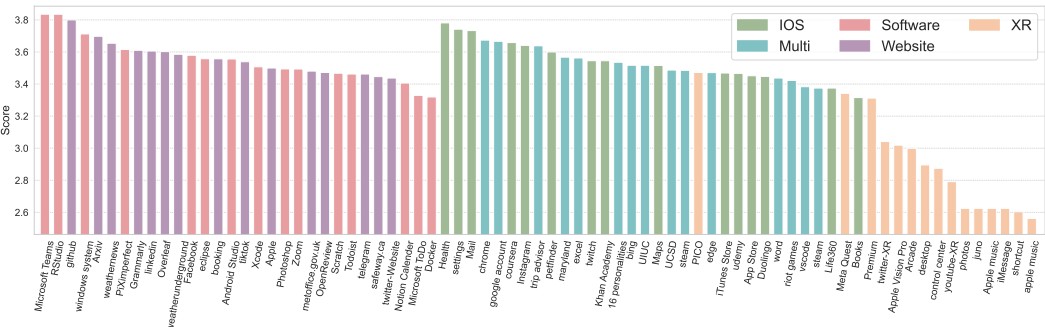

Figure 5: Fine-grained performance of GPT-4V in each software and website.

Table 5: Detailed scores for each tasks in **Software** scenarios. 'Dyn.' refers to queries on dynamic GUI content, and 'Pred.' indicates prediction tasks.

| Models | Setting | Caption | | Complex Tasks | | | Conversation | | Average |
|---|---|---|---|---|---|---|---|---|---|
| | | Concise | Detailed | Static | Dyn. | Pred. | Round 1 | Round 2 | |
| Gemini-Pro-1.5 | R. | 3.659 | 2.837 | 2.969 | 2.822 | 3.450 | 3.608 | 3.845 | 3.339 |
| | E. | 3.350 | 2.468 | 2.741 | 2.431 | 3.292 | 3.458 | 3.837 | 3.152 |
| Qwen-VL-Max | R. | 2.381 | 1.758 | 2.277 | 2.144 | 2.724 | 3.125 | 3.317 | 2.676 |
| | E. | 2.459 | 1.693 | 2.143 | 1.954 | 2.742 | 3.174 | 3.298 | 2.624 |
| | H. | 2.474 | 1.711 | 2.137 | 2.032 | 2.834 | 3.223 | 3.257 | 2.651 |
| GPT-4V | R. | 3.579 | 2.676 | **3.243** | 3.011 | 3.630 | 3.925 | 4.131 | 3.589 |
| | E. | 3.141 | 2.301 | 2.927 | 2.627 | 3.541 | 3.844 | 4.103 | 3.407 |
| | H. | 3.352 | 2.509 | 3.053 | 2.849 | 3.609 | 3.928 | 4.163 | 3.520 |
| | C.C. | 3.454 | 2.547 | 1.818 | 2.335 | 3.577 | 3.521 | 3.884 | 3.028 |
| | D.C. | 3.412 | 2.627 | 2.603 | 2.591 | 3.723 | 3.759 | 4.072 | 3.350 |
| | H.+D.C. | 3.436 | 2.677 | 2.927 | 2.750 | **3.791** | 3.857 | 4.148 | 3.494 |
| GPT-4o | H. | **4.048** | **3.028** | 3.125 | **3.117** | 3.562 | **4.129** | **4.318** | **3.644** |
| ChatUnivi | - | 1.587 | 1.240 | 1.705 | 1.656 | 2.524 | 2.698 | 3.366 | 2.389 |
| Minigpt4Video | - | 1.246 | 1.073 | 1.249 | 1.235 | 1.675 | 1.494 | 1.719 | 1.475 |
| VideoChat2 | - | 1.992 | 1.312 | 1.812 | 1.682 | 2.158 | 2.342 | 2.720 | 2.144 |
| **GUI-Vid** | - | 3.562 | 2.058 | 2.376 | 2.090 | 3.435 | 3.080 | 3.260 | 2.847 |

**Performance Variate in Different GUI Scenarios.** GPT-4V and Gemini excel in common scenarios such as mobile and website interfaces but show marked deficiencies in more complex GUI environments like XR and multi-window interactions, across both captioning and intricate tasks. This performance gap highlights a significant shortfall in understanding environments where GUI elements are scattered and demand sophisticated interpretation. It emphasizes the critical need for specialized benchmarks and datasets tailored to these complex GUI scenarios, which is essential for enhancing the GUI-oriented capabilities of MLLMs, paving the way for them to become truly reliable and high-performing general control agents.

**Keyframe Selection is Important for GUI-orientated Tasks.** Across both basic tasks such as captioning and more complex tasks like prediction and reasoning, significant variations are evident among keyframe selection methods. GPT-4V and Gemini significantly benefit from using random-selected and human-selected keyframes, scoring approximately 0.2-0.3 points higher in both captioning and free-form tasks than those using programmatic extraction. This suggests that traditional keyframe technologies, designed for natural videos, are less effective for detecting essential GUI operations, particularly when subtle movements like mouse clicks and dynamic changes are involved. Conversely, the difference in performance is relatively smaller in Qwen-VL-Max, indicating that while keyframe selection methods are crucial for models proficient in GUI content, they exert less influence on less capable models.

**Dynamic GUI Tasks Continue to Challenge MLLMs.** In the fine-grained tasks depicted in Table 5, GPT-4V and GPT-4o excel with static GUI content and prediction tasks over image sequences but struggle with providing detailed descriptions for entire videos and dynamic GUI content. This discrepancy is attributed to minor variations in GUI that significantly impact descriptions. Enhancing the number of keyframes and the granularity of perception might mitigate these issues. Among VideoLLMs, ChatUnivi excels in conversational tasks by effectively leveraging contextual nuances, particularly in subsequent rounds, yet it underperforms in GUI-oriented captioning tasks. In contrast, GUI-Vid demonstrates proficiency in sequential tasks but falls short in both captioning and static content handling. This gap is linked to deficiencies in GUI-Vid's pretraining, which lacked comprehensive GUI content crucial for effective vision-text alignment, as evidenced by its poor performance in Table 13 and an instruction tuning process also failed to fully address these shortcomings.

**Vision Perception is Important for Sequential GUI Tasks.** As demonstrated in Table 5, integrating detailed textual information slightly outperforms purely vision-based inputs or detailed captions, akin to a Chain of Thought (CoT) [43] setting. Surprisingly, GPT-4V excels in caption and prediction

Table 6: The overall results for ablation study on GUI-Vid finetuning. F.K. and E.K. mean keyframes during the finetuning and evaluation process respectively. I. means Image, and V. means Video.

| Setting | F.K. | E.K. | Data I. | Data V. | Software MC | Software Free | Website MC | Website Free | XR MC | XR Free | Multi MC | Multi Free | IOS MC | IOS Free | Android MC | Android Free | Avg. MC | Avg. Free |
|---|---|---|---|---|---|---|---|---|---|---|---|---|---|---|---|---|---|---|
| Baseline | - | 8 | - | - | 45.5% | 2.144 | 42.6% | 2.221 | 44.0% | 2.005 | 40.4% | 2.222 | 40.2% | 2.169 | 44.7% | 2.119 | 42.9% | 2.147 |
|  | - | 16 | - | - | 45.1% | 2.144 | 41.8% | 2.240 | 41.0% | 2.007 | 40.7% | 2.238 | 39.9% | 2.138 | 44.7% | 2.147 | 42.2% | 2.154 |
| GUI-Vid | 8 | 8 | ✗ | ✔ | 58.3% | 2.709 | 53.6% | 2.817 | 62.2% | 2.626 | **54.2%** | 2.627 | 53.1% | 2.708 | 54.9% | 2.501 | 56.0% | 2.665 |
|  |  | 8 | ✔ | ✔ | **59.9%** | **2.856** | 54.1% | 2.925 | 59.0% | 2.751 | 52.1% | 2.837 | 50.0% | 2.756 | 54.0% | 2.571 | 54.8% | 2.782 |
|  |  | 16 | ✗ | ✔ | 59.0% | 2.709 | **55.1%** | 2.821 | **62.8%** | 2.645 | 53.3% | 2.624 | **55.5%** | 2.727 | **55.7%** | 2.501 | **56.9%** | 2.671 |
|  |  | 16 | ✔ | ✔ | **59.9%** | 2.847 | 54.1% | **2.957** | 55.6% | **2.764** | 52.9% | **2.861** | 51.8% | **2.772** | 53.4% | **2.572** | 54.6% | **2.796** |

Table 7: GPT-4o average performance in six GUI scenarios under low and high resolution.

| Res. | Desc. | Conv. | Dyn. | Static | Caption | Free |
|---|---|---|---|---|---|---|
| Low | 2.794 | 3.912 | 3.150 | 2.869 | 3.672 | 3.394 |
| High | **3.031** | **4.056** | **3.318** | **3.131** | **3.911** | **3.573** |

tasks with just detailed captions, providing insights on enhancing specific GUI-oriented tasks through additional textual information. However, it still falls short in more challenging tasks, such as retrieving static or dynamic content. This underscores the critical role of visual perception in GUI environments, where even minor changes can significantly impact outcomes.

**Supreme Enhancement of GUI-Vid on Graphic-based Interface After Finetuned on GUI-WORLD.** As a pioneering study in training VideoLLMs as screen agents, GUI-Vid significantly outperforms the baseline model, showing an average improvement of 30% across various tasks and GUI scenarios, even surpassing the commercial ImageLLM, Qwen-VL-Max. This enhancement is particularly notable in captioning and prediction over image sequences, where GUI-Vid matches the performance of GPT-4V and Gemini-Pro. As shown in Figure 6, our two-stage progressive fintuning significantly enhances the performance in all GUI scenarios. Remarkably, GUI-Vid scored 3.747 in caption tasks within the

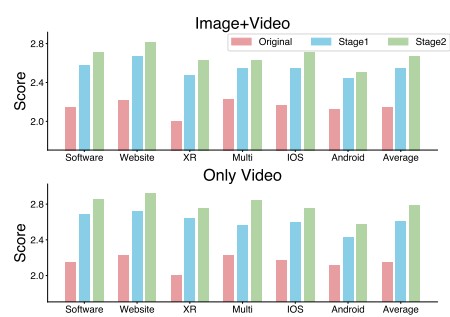

Figure 6: Two stages of progressive training enhance GUI ability.

XR scenario, highlighting its potential in XR applications and the high-quality annotations provided by our dataset. However, in Multiple-Choice QA and Chatbot tasks, GUI-Vid still lags behind industry leaders like GPT-4V and Gemini-Pro, a discrepancy likely due to the baseline LLM's weaker performance and the challenges of instruction-based fine-tuning.

**Upper Bound of GUI-orientated Capability with More Keyframes and High Resolution.** As depicted in Table 6, our two ablation studies during the fine-tuning phase demonstrate that utilizing GUI image-text captioning data significantly enhances the model's preliminary understanding of GUI elements, outperforming training that relies solely on videos. Additionally, an increased number of keyframes correlates with improved performance across various scenarios, notably in environments featuring multiple windows and software applications. Further evidence from Table 7 reveals that higher image resolutions substantially boost task performance, both basic and complex, for GPT-4o. These findings underscore the potential for further developing a more robust GUI Agent.

# 5 Conclusion

In this paper, we have introduced GUI-WORLD, a comprehensive GUI-oriented dataset designed to benchmark and enhance understanding of virtual interface, especially seqeuntial and dynamic tasks. This dataset extensively covers six scenarios and various tasks, addressing the previous research gap in comprehensively evaluating models' capabilities in graphic-based understanding. We conduct extensive benchmarks on leading MLLMs and the first Video Agent 'GUI-Vid' finetuned on our GUI-WORLD specifically for tasks requiring temporal information, achieving results comparable to top-performing models, providing detailed insights into enhancing GUI-related capabilities.

## 6 Limitations

While our work presents significant advancements in the field of GUI agents, there are several limitations that need to be addressed. Firstly, despite expanding the dataset to include various GUI scenarios, our models still show limited generalization capabilities when applied to environments not represented in the training data. This highlights the need for further research to improve the adaptability and robustness of GUI agents in diverse and unseen environments. Additionally, the accuracy of our models heavily relies on the selection of keyframes. Automatically extracted keyframes often fail to capture the essential elements needed for accurate GUI understanding, indicating the need for more sophisticated keyframe extraction techniques. Furthermore, although VideoLLMs have shown improvements in handling dynamic content, their ability to understand and predict sequential information in GUI tasks remains suboptimal. This suggests a necessity for future work to focus on enhancing the temporal understanding capabilities of these models. Finally, the training and fine-tuning processes for VideoLLMs require significant computational resources, which may not be accessible to all researchers.

## 7 Potential Negative Societal Impacts

While our work aims to advance the capabilities of GUI agents for beneficial applications, it is important to consider potential negative societal impacts. The use of GUI agents, especially those capable of operating across multiple environments and platforms, raises significant privacy concerns. Ensuring that these agents operate within strict ethical guidelines and that user data is handled securely and responsibly is paramount. There is also the risk of misuse of advanced GUI agents for malicious purposes, such as unauthorized access to sensitive information or automated exploitation of software vulnerabilities. Establishing robust security measures and ethical usage policies is essential to mitigate these risks.

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

# Part I

# Appendix

## Table of Contents

## A    Related Work

**MLLMs for GUI.**    Building upon the significant advancements in LLMs [52–55] and advanced
modality-mixing technologies [56, 57], groundbreaking MLLMs such as GPT-4V [1] and Gemini-Pro
[42], along with open-source MLLMs like the LLaVA-1.6 series [2, 58], CogVLM [59], and Qwen-
VL series [41], have shown outstanding performance across various tasks [60–65]. Venturing beyond
text and single image, several studies are now exploring the integration of video modalities for tasks
requiring dynamic or sequential visual content [44, 35, 66, 67]. In the GUI domain, leveraging the
robust vision perception capabilities of MLLMs, applications such as WebAgents [8, 68, 23] and
Mobile Agents [17, 12, 69] have gained popularity for handling everyday tasks like navigation and
VQA. Frontier research is also investigating the use of MLLMs as general control agents, such as
in playing computer games [70, 71] and serving as OS co-pilots [72, 24], paving the way for more
complex GUI operations.

**GUI Benchmark & Dataset.**    Building upon the foundational work of Rico [13], the first mobile
GUI video dataset, and AitW [16], which features 715k episodes of sequential images, research has
extensively covered mobile [14, 73, 74] and web GUI environments [75, 19, 76–78]. Mind2Web
[20] stands out in web-based datasets with over 2,000 tasks from 137 websites across 31 domains.
Advances continue into desktop GUIs with new toolkits [23], benchmarks [21, 79], and frameworks
[80, 81, 11]. Research on GUI also transfers from comprehending single images in a static workspace
[8] to sequential operations or multi-hop scenarios [24, 22], challenging the understanding and
operation capability of these powerful models.

## B    Details of Dataset Construction

### B.1    Six Main GUI Categories

In earlier endeavors pertaining to GUI, such as those involving GUI testing [82–84], the focus
was segmented into GUIs for Website, Software, IOS and Android platforms. However, as a
comprehensive GUI dataset,we included all potential GUI scenarios into our dataset to ensure that
our data is the most comprehensive knowledge that the GUI Agent needs to learn; we divided these
scenarios into six categories:

- **Android.** This category focuses on the GUI scenarios that occur within the Android operating
  system, which is predominantly used on smartphones. Android's ubiquity in the mobile market has
  led to a wide variety of GUI designs and interaction patterns, making it a rich field for study. This
  category has been the subject of extensive scrutiny in scholarly works such as  [13, 73, 16, 85].
- **Software.** This category encapsulates the GUI scenarios arising within software applications,
  whether they are standalone programs or components of a larger suite. The diversity of software
  applications, from productivity tools to creative suites, offers a wide range of GUI scenarios for
  exploration. The literature is rich with research in this area, such as  [86].
- **Website.** This category is concerned with the GUI scenarios that manifest within a web browser.
  Given the ubiquity of web browsing in modern digital life, this category holds significant relevance.
  It holds a substantial representation in academic literature, with pioneering papers such as  [20, 21]
  proposing excellent GUI datasets for websites.
- **IOS.** This category zeroes in on the GUI scenarios that transpire within the iOS operating system,
  the proprietary system for Apple devices like the iPhone and iPad. The iOS platform is known for
  its distinct design aesthetics and interaction patterns, providing a unique context for GUI research.
  A number of studies, such as  [87, 88] make use of GUI information in IOS.
- **Multi Windows.** This category is dedicated to GUI scenarios that necessitate simultaneous
  interaction with multiple windows, a common occurrence in desktop environments where users
  often juggle between several applications or documents. Despite the common use of multi-window
  interaction in everyday GUI usage, there has been relatively little research into this area [89]. The
  need for efficient multitasking in such scenarios presents unique challenges and opportunities for

GUI design and interaction research. As of our knowledge, there are no specific datasets catering to these multi-window GUI scenarios.

- **XR.** XR encompasses Virtual Reality (VR), Augmented Reality (AR), and Mixed Reality (MR) [90]. Given the advancements in XR technology and the growing accessibility of commercial-grade head-mounted displays [25, 91], XR has emerged as a novel medium for human-computer interaction. This necessitates the exploration of GUI within XR environments. In these scenarios, the GUI takes on a 3D, immersive form [92], demanding the agent to comprehend and navigate a 3D space. The emerging field of XR presents a new frontier for GUI research, with unique challenges and opportunities due to its immersive and interactive nature. To date, as far as we are aware, there are no datasets that specifically address GUI in the realm of XR.

## B.2 Selected Website/Software

In our study, we selected a diverse range of websites and software to comprehensively evaluate GUI understanding capabilities across various user scenarios. These selections cover essential categories such as social media, productivity tools, online shopping, and educational platforms, providing a broad spectrum of GUI environments.

The chosen websites, as shown in Figure 7, include popular social media platforms like Instagram, Twitter, and LinkedIn, which are integral to understanding dynamic and interactive GUI elements. We also included widely used productivity tools such as Microsoft Teams, Notion, and Slack to evaluate GUI tasks in professional and collaborative settings.

For software shown in Figure 8, we incorporated key applications like Adobe Photoshop and MAT-LAB to assess GUI operations in specialized and technical environments. Additionally, video conferencing tools like Zoom and cloud storage services like Google Drive were included to represent common remote work and file management scenarios.

These selections ensure that our study encompasses a wide array of user interactions and GUI complexities, thereby providing a robust evaluation of the current state-of-the-art in GUI understanding by MLLMs and comprehensively constructing a high-quality dataset.

## B.3 Human Keyframes Annotation Process

**Annotator's Information** The annotation is conducted by 16 authors of this paper and 8 volunteers independently. As acknowledged, the diversity of annotators plays a crucial role in reducing bias and enhancing the reliability of the benchmark. These annotators have knowledge in the GUI domain, with different genders, ages, and educational backgrounds. The education backgrounds of annotators are above undergraduate. To ensure the annotators can proficiently mark the data, we provide them with detailed tutorials, teaching them how to use software to record videos or edit video clips. We also provide them with detailed criteria and task requirements in each annotation process.

**Recording Video.** For self-recording videos, we employ OBS[3] on the Windows system for screen capturing and the official screen recording toolkit on the Mac/IOS system. This process necessitates human labelers to execute a series of targeted actions within specific websites or applications, which are subsequently captured as raw video footage. These actions, commonplace in everyday usage, enhance the reliability of our dataset. Subsequently, the raw videos are segmented into sub-videos, each encapsulating multiple actions (e.g., clicking a button) to achieve a specific objective (e.g., image search). The videos are then processed to extract keyframes annotated with detailed descriptions.

**Edition Based on YouTube Videos.** For sourcing videos from YouTube, we utilize a search protocol formatted as "[website name/application name] + tutorial" to compile relevant video lists. Human labelers first review these videos to understand the primary operations they depict. These videos are then divided into sub-videos, each containing several actions directed towards a

---

[3]https://obsproject.com/

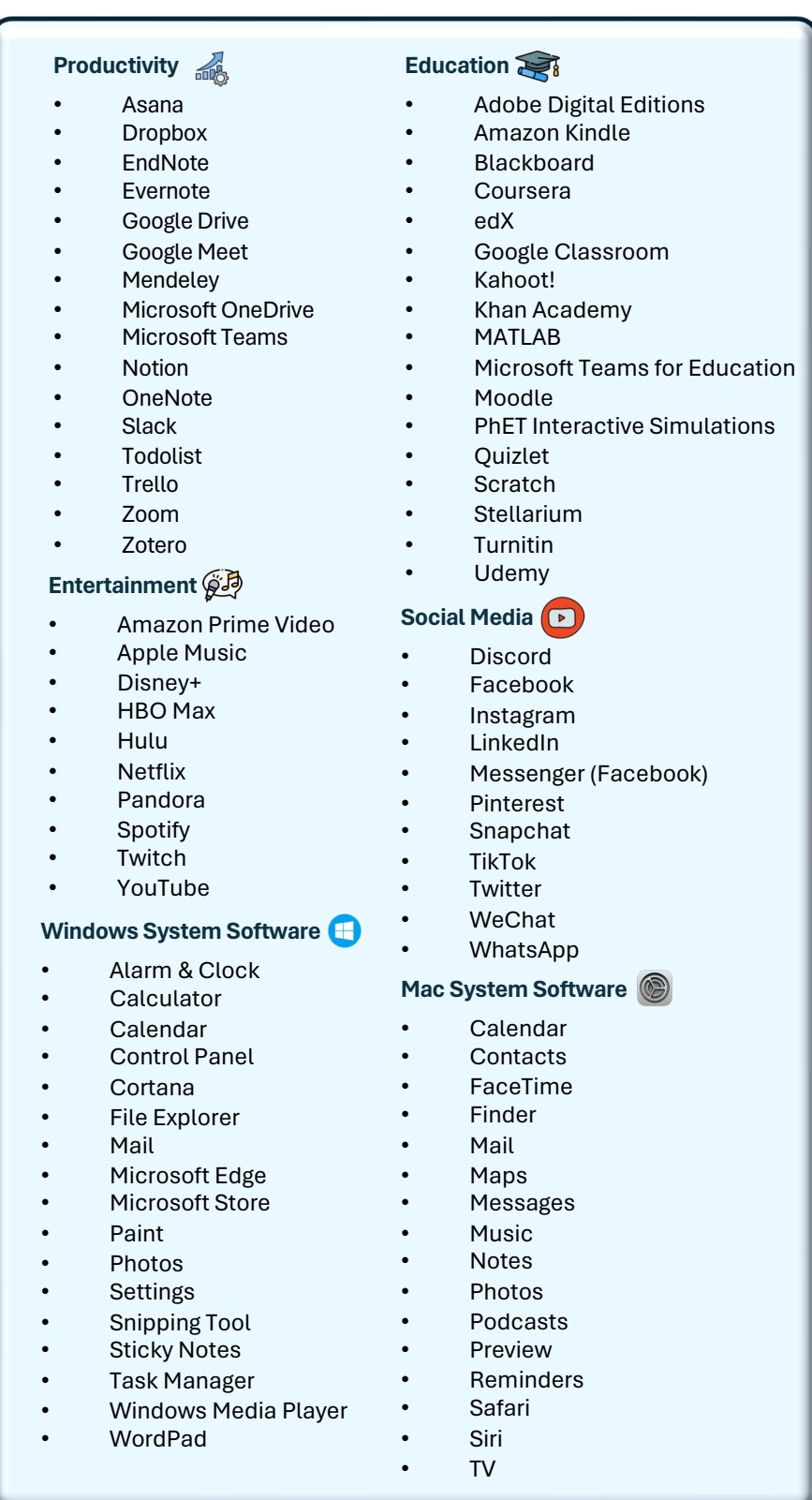

**Productivity**

- Asana
- Dropbox
- EndNote
- Evernote
- Google Drive
- Google Meet
- Mendeley
- Microsoft OneDrive
- Microsoft Teams
- Notion
- OneNote
- Slack
- Todolist
- Trello
- Zoom
- Zotero

**Entertainment**

- Amazon Prime Video
- Apple Music
- Disney+
- HBO Max
- Hulu
- Netflix
- Pandora
- Spotify
- Twitch
- YouTube

**Windows System Software**

- Alarm & Clock
- Calculator
- Calendar
- Control Panel
- Cortana
- File Explorer
- Mail
- Microsoft Edge
- Microsoft Store
- Paint
- Photos
- Settings
- Snipping Tool
- Sticky Notes
- Task Manager
- Windows Media Player
- WordPad

**Education**

- Adobe Digital Editions
- Amazon Kindle
- Blackboard
- Coursera
- edX
- Google Classroom
- Kahoot!
- Khan Academy
- MATLAB
- Microsoft Teams for Education
- Moodle
- PhET Interactive Simulations
- Quizlet
- Scratch
- Stellarium
- Turnitin
- Udemy

**Social Media**

- Discord
- Facebook
- Instagram
- LinkedIn
- Messenger (Facebook)
- Pinterest
- Snapchat
- TikTok
- Twitter
- WeChat
- WhatsApp

**Mac System Software**

- Calendar
- Contacts
- FaceTime
- Finder
- Mail
- Maps
- Messages
- Music
- Notes
- Photos
- Podcasts
- Preview
- Reminders
- Safari
- Siri
- TV

Figure 7: List of desktop softwares in GUI-WORLD.

**Social Media**

- https://instagram.com/
- https://twitter.com/
- https://whatsapp.com/
- https://pinterest.com/
- https://linkedin.com/
- https://tiktok.com/
- https://discord.com/
- https://reddit.com/
- https://telegram.org/

**Search Engines**

- https://google.com/
- https://yandex.com/
- https://bing.com/
- https://baidu.com/
- https://search.aol.com/

**Online Shopping**

- https://etsy.com/
- https://alibaba.com/
- https://ebay.com/

**Education and Learning**

- https://quora.com/
- https://byjus.com/
- https://cambridge.org/
- https://udemy.com/
- https://coursera.org/
- https://khanacademy.org/
- https://edx.org/
- https://academia.edu/

**Technology and Software**

- https://microsoft.com/
- https://apple.com/
- https://adobe.com/
- https://github.com/
- https://openai.com/
- https://oracle.com/
- https://vmware.com/

**Travel and Hospitality**

- https://booking.com/
- https://tripadvisor.com/
- https://yelp.com/
- https://airbnb.com/
- https://expedia.com/
- https://hotels.com/
- https://trivago.com/
- https://homeaway.com/

**Finance**

- https://moneycontrol.com/
- https://bloomberg.com/
- https://investing.com/
- https://marketwatch.com/
- https://seekingalpha.com/
- https://zacks.com/

**Food and Cooking**

- https://cookpad.com/
- https://allrecipes.com/
- https://foodnetwork.com/
- https://yummly.com/
- https://epicurious.com/

Figure 8: List of some websites in GUI-WORLD.

single goal (e.g., image search). Like the self-recorded footage, these segments are processed to isolate keyframes and furnish them with descriptive annotations.

**Keyframes Annotation.** After obtaining the GUI video clips, human annotators will filter out the keyframes of the operations based on the video content and the mouse and keyboard actions at that time. They will also label the sub-operations or targets between the two keyframes. Once the annotation is complete, the annotators will provide an overall description of the entire video, summarizing the main goal of the human operations in the video. After all the information is annotated, we will use a Large Language Model (LLM) to refine the text content, reducing any errors made by human annotators and adjusting the sentence structure. The prompt we use for the LLM to polish the human annotations is shown in Figure 9 and Figure 10.

**Human-LLM Cooperated Instruction Generation.** To curate and refine the golden answer of each video-instruction pair generated by GPT-4V, given that the raw response from GPT-4V may contain harmful content or hallucinations. The role of humans in the golden answer generation process is to enhance the difficulty of the questions and remove harmful and incorrect content, as shown in Table 8.

Table 8: Comparison of raw answer from GPT-4V and golden answer from humans.

| Raw response from GPT-4V | Golden answer from humans |
|---|---|
| "Q": "What subcategories are visible under the 'Lifestyle' section in the static GUI content of the video?" "A": "In the 'Lifestyle' section of the telegram.com website, the visible subcategories are 'Worcester Living', 'Faith', and 'Food'. These are clearly displayed in a horizontal list just below the main 'Lifestyle' category header, indicating different areas of content that users can explore within the lifestyle section." | "Q": "What subcategories are visible under the 'Lifestyle' section in the static GUI content of the video?" "A": "The 'Lifestyle' section of the telegram.com website features visible subcategories such as 'Worcester Living', 'Faith', and 'Food'. These are prominently shown in a horizontal list beneath the main 'Lifestyle' header, offering users various content areas to explore within the section." |

**Human verifying GPT-4V annotated captions.** We evaluated the quality of annotations from GPT-4V by selecting 1,000 detailed descriptions and captions generated by GPT-4V, which were then assessed by human annotators. The high satisfaction rate of 98% underscores the quality and relevance of the GPT-4V annotations.

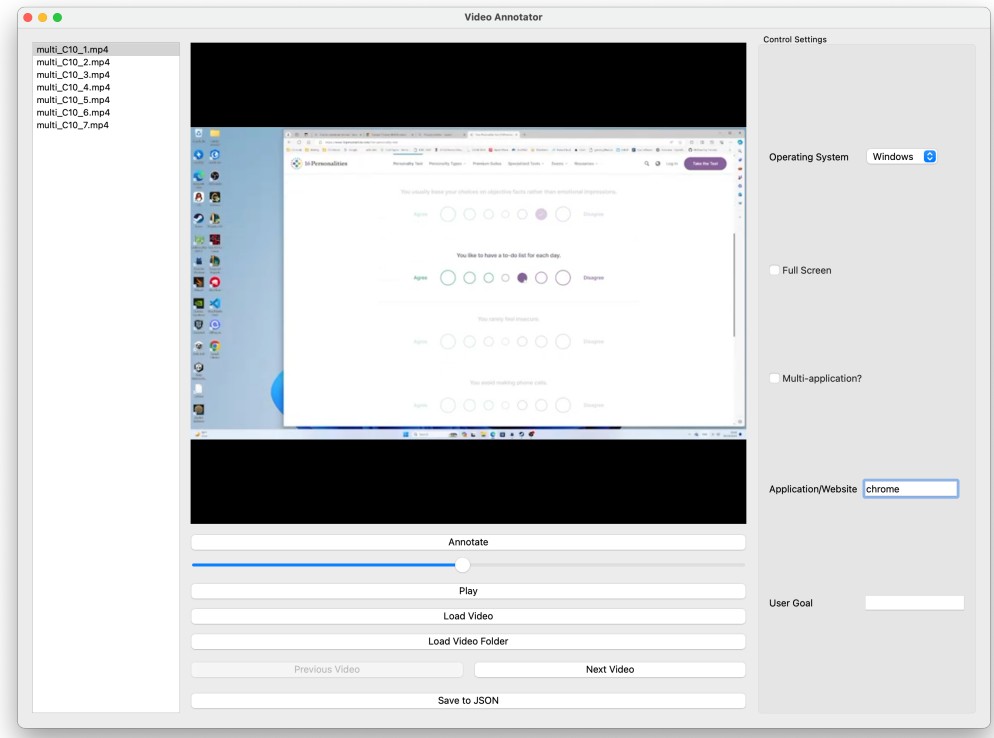

Figure 9: The overall preview of our annotating software.

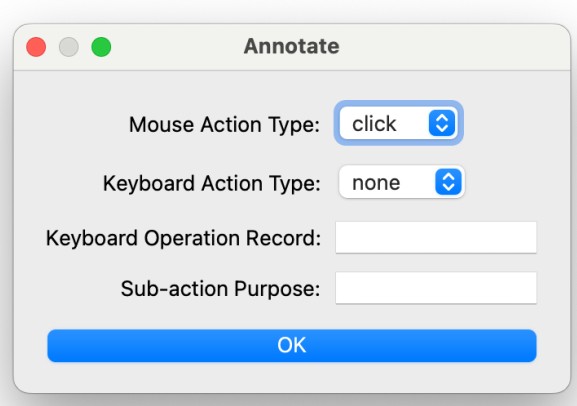

Figure 10: The interface for annotating a keyframe, consists of mouse action, keyboard action, and a short sub-action purpose.

## C Dataset Analysis

In this section, we provide an analysis of the length distribution of QA in each GUI scenario, as illustrated in Figure 11 and Figure 12. Question focus on sequential and predictical tasks are slightly longer than other types, while golden answer of static tasks tend to be longer. Length of Question-answer pair in various GUI scenarios are similarly distributed, with questions in Android environment are slightly shorter, and answers in XR environment are longer.

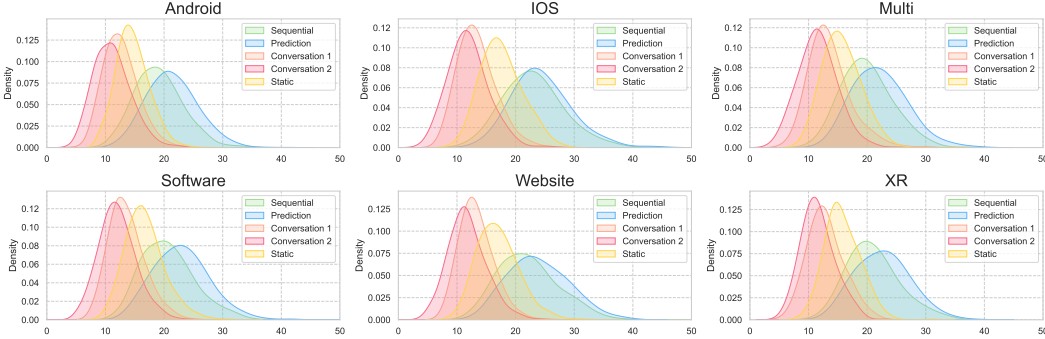

Figure 11: Length distribution of free-form questions.

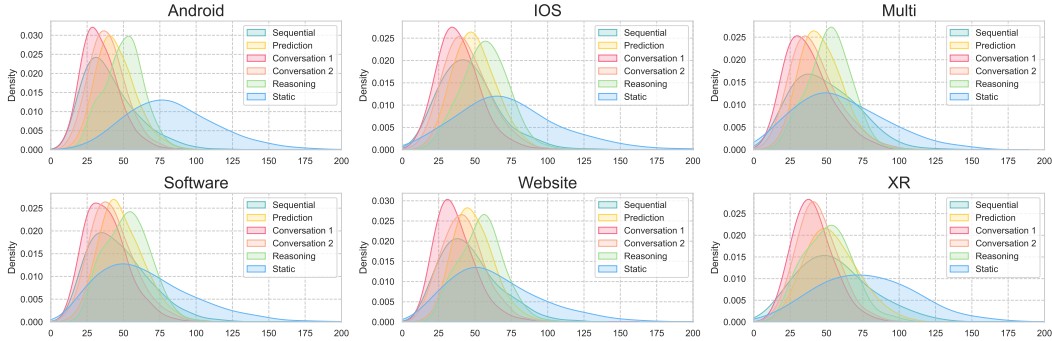

Figure 12: Length distribution of answers to free-form questions.

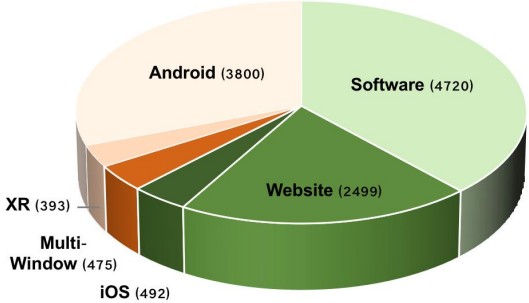

Figure 13: Statistic of different GUI scenarios in GUI-WORLD.

## D  Details of Experiments Setups

### D.1  Finetune dataset construction

We use two settings to finetune GUI-Vid, one with video-text pairs only, and the other with video-text and image-text pairs, which are all GUI content:

- **Video Only.** In this setting, we only trained GUI-Vid with video-text pairs in GUI-WORLD, as shown in Table 9.
- **Video-Image.** Inspired by the pre-trained process of Videochat2, we include image-text pairs to help the visual encoder align GUI knowledge. These images are selected from our GUI-WORLD, MetaGUI [14], and OmniAct [21] for high-quality GUI content. Subsequently, we use GPT-4V to generate a detailed description and a concise caption for each image. Finally, we construct a dataset consisting of video-text and image-text pairs for gaining comprehensive GUI-orientated capabilities.

Table 9: Video-only finetune dataset.

| Stage | Data types | Amount |
|-------|------------|--------|
| 1 | Detailed Description | 14,276 |
|   | Concise Caption | 7,138 |
| 2 | GUI VQA | 21,414 |
|   | Multiple-Choice QA | 14,276 |
|   | Conversation | 7,138 |

Table 10: Video-image finetune dataset.

| Stage | Data types | Source | Type | Amount |
|-------|------------|--------|------|--------|
| 1 | GUI-WORLD | Video | Detailed Description | 14,276 |
|   |            |       | Concise Caption | 7,138 |
|   |            | Image | Detailed Description | 5,555 |
|   |            |       | Concise Caption | 5,555 |
|   | METAGUI | Image | Detailed Description | 19,626 |
|   |         |       | Concise Caption | 19,626 |
|   | OmniAct |       | Detailed Description | 260 |
|   |         |       | Concise Caption | 260 |
| 2 | GUI-WORLD | Video | GUI VQA | 21,414 |
|   |           |       | Multiple-Choice QA | 14,276 |
|   |           |       | Conversation | 7,138 |

### D.2  Hyperparameter Settings

In this section, we will introduce the hyperparameters of MLLMs to facilitate experiment reproducibility and transparency. We divide them into three parts: the inference phase during benchmark and dataset construction, the LLM-as-a-Judge phase, and the fine-tuning phase. All our experiments were conducted on a server equipped with dual A800 and dual 4090 GPUs.

**Inference.**  We empirically study 6 MLLMs, involving 3 Image-LLMs and 3 Video-LLMs, with their hyperparameters detailed as follows:

- **GPT-4V [1]:** We set the temperature and top-p as 0.9, max-token as 2048, and both all images input are set as high quality in *Instruction Dataset Construction* and benchmarking.
- **Gemini-Pro-1.5 [42]:** We use the default settings, which set temperature as 0.4, top-p as 1, and max-token as 2048. It should be noted that during our project, Gemini-Pro-1.5 is still under the user request limit, which only provides 100 requests per day, making our benchmark difficult.

769   Given that Gemini hasn't launched Pay-as-you-go[4], we will include benchmark results on 'Human'
770   setting as soon as possible.

771 • **Qwen-VL-Max [41]:** We use the default settings for Qwen-VL-Max, with top-p as 0.8 and max-
772   token as 2048. Given that the input context window is merely 6,000 for Qwen, we scale the
773   resolution for all images to 0.3.
774 • **ChatUnivi [44]:** We use ChatUnivi-7B built upon Vicuna-v0-7B and set the max frame as 100,
775   temperature as 0.2, and max-token as 1024.
776 • **Minigpt4video [45]:** We use the suggested settings[5] for this model and the max-frame are set as
777   45, with only the max-token being modified to 1024.
778 • **VideoChat2 & GUI-Vid [46]:** For a fair comparison, we set the same hyperparameters for
779   VideoChat2 & GUI-Vid. We set the max-token as 1024, top-p as 0.9, temperature as 1.0, max-
780   frame as 8/16, repetition penalty as 1.2, and length penalty as 1.2.

781 **LLM-as-a-Judge.**   We studied four LLM-as-a-Judge in giving a similarity score for the MLLM's
782   response and ground truth, namely GPT-4 [52], ChatGPT [93], LLaMA-3-70b-instruct [54], and
783   Mixtral-8x22b-instruct-v0.1 [55]. Hyperparameter settings are detailed as follows:

784 • **GPT-4 & ChatGPT.** We set the temperature as 0.6 and others as default.
785 • **LLaMA-3-70b-instruct.** We set the temperature as 0.6, top-p as 0.9, top-k as 50.
786 • **Mixtral-8x22b-instruct-v0.1.** We set top-p as 0.7, top-k as 50, and temperature as 0.7.

787 **Finetune.**   We include several hyperparameter settings in experiment settings and ablation studies,
788   as shown in Table 11.

Table 11: Configuration settings for fine-tuning.

| Config | Setting |
| --- | --- |
| input frame | 8 |
| input resolution | 224 |
| max text length | 512 |
| input modal | I. + V. |
| optimizer | AdamW |
| optimizer momentum | $\beta_1, \beta_2 = 0.9, 0.999$ |
| weight decay | 0.02 |
| learning rate schedule | cosine decay |
| learning rate | 2e-5 |
| batch size | 4 |
| warmup epochs | 0.6 |
| total epochs | 3 |
| backbone drop path | 0 |
| QFormer drop path | 0.1 |
| QFormer dropout | 0.1 |
| QFormer token | 96 |
| flip augmentation | yes |
| augmentation | MultiScaleCrop [0.5, 1] |

### 789  D.3   Evaluation.

790   Given the complexity of free-form answers in GUI scenarios, the evaluation includes specific positions
791   of GUI elements, textual content, and comparing the response to the golden answer. LLM-as-a-judge
792   has been widely used in previous studies for complex evaluation tasks [47, 48]. Therefore, we
793   leverage LLM-as-a-Judge [47] in a similar setting to MM-vet [60], which compares the MLLM's
794   response to the golden answer. We carefully evaluate the accessibility of leveraging LLM-as-a-Judge,
795   selecting 1,000 samples covering 6 free-form questions mentioned in our dataset. As shown in
796   Table 12, GPT-4 outperforms other LLMs, exhibiting a better human alignment on providing a

---

[4] https://ai.google.dev/pricing
[5] https://github.com/Vision-CAIR/MiniGPT4-video

Table 12: Evaluating LLM-as-a-Judge as a replacement for human judging in the scoring setting.

| Models | Pearson($\uparrow$) | Spearman($\uparrow$) | Kendall($\uparrow$) | $ per Benchmark($\downarrow$) |
|---|---|---|---|---|
| GPT-4 | **0.856** | **0.853** | **0.793** | 120$ |
| ChatGPT | 0.706 | 0.714 | 0.627 | **12$** |
| Llama-3-70b-instruct | 0.774 | 0.772 | 0.684 | **12$** |
| Mixtral-8x22b-instruct-v0.1 | 0.759 | 0.760 | 0.670 | 15$ |

similarity score for the response compared to the golden answer, although it is approximately 10 times more expensive than other models.

Table 13: Scores of Caption (Cap.) and Description (Des.) tasks in six GUI scenarios.

| Models | Setting | Software | | Website | | XR | | Multi | | IOS | | Android | | Avg. | |
|---|---|---|---|---|---|---|---|---|---|---|---|---|---|---|---|
| | | Cap. | Des. | Cap. | Des. | Cap. | Des. | Cap. | Des. | Cap. | Des. | Cap. | Des. | Cap. | Des. |
| Gemini-Pro-1.5 | R. | 3.659 | 2.837 | 3.613 | 2.860 | 2.995 | 2.590 | 3.276 | 2.470 | 3.678 | 2.936 | - | - | 3.444 | 2.739 |
| | E. | 3.350 | 2.468 | 3.159 | 2.422 | 2.837 | 2.279 | 2.824 | 2.109 | 3.394 | 2.519 | 3.185 | 2.312 | 3.125 | 2.351 |
| Qwen-VL-Max | R. | 2.381 | 1.758 | 2.326 | 1.681 | 2.172 | 1.772 | 2.035 | 1.463 | 2.513 | 1.662 | 2.141 | 1.565 | 2.261 | 1.650 |
| | E. | 2.459 | 1.693 | 2.317 | 1.599 | 2.167 | 1.638 | 2.190 | 1.438 | 2.189 | 1.615 | 2.002 | 1.429 | 2.221 | 1.569 |
| | H. | 2.474 | 1.711 | 2.457 | 1.698 | 2.383 | 1.777 | 1.910 | 1.346 | 2.577 | 1.795 | 2.474 | 1.711 | 2.360 | 1.665 |
| GPT-4V | R. | 3.579 | 2.676 | 3.612 | 2.699 | 2.975 | 2.525 | 3.281 | 2.661 | 3.757 | 2.775 | 3.655 | 2.755 | 3.479 | 2.682 |
| | E. | 3.141 | 2.301 | 3.293 | 2.380 | 2.471 | 2.085 | 3.063 | 2.324 | 3.624 | 2.611 | 3.201 | 2.312 | 3.132 | 2.335 |
| | H. | 3.352 | 2.509 | 3.702 | 2.750 | 3.050 | **3.556** | 3.524 | 2.673 | 3.670 | 2.588 | - | - | 3.460 | 2.614 |
| GPT-4o | H. | **4.048** | **3.028** | **4.067** | **3.233** | 3.398 | 2.729 | **3.869** | **3.111** | **4.014** | **2.993** | **4.071** | **3.095** | **3.911** | **3.869** |
| ChatUnivi | - | 1.587 | 1.240 | 1.569 | 1.254 | 1.417 | 1.148 | 1.575 | 1.267 | 1.480 | 1.146 | 1.778 | 1.249 | 1.568 | 1.217 |
| Minigpt4Video | - | 1.246 | 1.073 | 1.200 | 1.057 | 1.320 | 1.106 | 1.130 | 1.034 | 1.190 | 1.076 | 1.184 | 1.061 | 1.212 | 1.068 |
| VideoChat2 | - | 1.992 | 1.312 | 1.817 | 1.307 | 1.838 | 1.426 | 2.222 | 1.433 | 2.169 | 1.270 | 2.119 | 1.294 | 1.900 | 1.340 |
| **GUI-Vid** | - | 3.562 | 2.085 | 3.655 | 2.167 | **3.747** | 2.153 | 3.370 | 1.742 | 3.566 | 2.071 | 2.662 | 1.248 | 3.427 | 1.911 |

Table 14: Detailed scores for each tasks in **Website** scenarios.

| Models | Setting | Static | Sequential | Prediction | Conversation1 | Conversation2 | Average |
|---|---|---|---|---|---|---|---|
| Gemini-Pro-1.5 | R. | 3.279 | 3.050 | 3.560 | 3.579 | 3.796 | 3.452 |
| | E. | 2.983 | 2.491 | 3.432 | 3.405 | 3.760 | 3.215 |
| Qwen-VL-Max | R. | 2.317 | 2.271 | 2.802 | 2.995 | 3.069 | 2.656 |
| | E. | 2.256 | 2.198 | 2.821 | 2.861 | 3.144 | 2.627 |
| | H. | 2.308 | 2.078 | 2.832 | 3.061 | 3.358 | 2.698 |
| GPT-4V | R. | 3.461 | 3.214 | 3.754 | 3.778 | 4.029 | 3.648 |
| | E. | 3.197 | 2.808 | 3.487 | 3.717 | 3.954 | 3.433 |
| | H. | **3.498** | 3.255 | 3.727 | 3.731 | 4.061 | 3.655 |
| | C.C. | 1.746 | 2.738 | 3.645 | 3.363 | 3.632 | 3.025 |
| | D.C. | 2.704 | 2.917 | 3.686 | 3.680 | 3.901 | 3.380 |
| | H.+D.C. | 3.313 | 3.221 | **3.852** | 3.850 | **4.171** | 3.682 |
| GPT-4o | H. | 3.443 | **3.373** | 3.672 | **4.086** | 4.122 | **3.740** |
| ChatUnivi | - | 1.701 | 1.668 | 2.524 | 2.514 | 3.338 | 2.349 |
| Minigpt4Video | - | 1.309 | 1.233 | 1.766 | 1.439 | 1.854 | 1.520 |
| VideoChat2 | - | 1.771 | 1.777 | 2.288 | 2.461 | 2.812 | 2.221 |
| GUI-Vid | - | 2.406 | 2.341 | 3.544 | 3.135 | 3.355 | 2.957 |

# E    Additional Experiments Results

In this section, we provide detailed result on each tasks in each GUI scenarios. For captioning tasks, Table 13 shows a comprehensive experimental results among six scenarios. For scores of LLM-as-a-Judge in specific task, see Table 14, Table 15, Table 16, Table 17, and Table 18. For BLEU [50] and BERTScore [51] in validating free-form and conversational questions, see Table 19, Table 20, Table 21, Table 24, Table 22, and Table 23. For performance in fine-grain (application level), see Figure 14 for Gemini-Pro and Figure 15 for Qwen-VL-Max.

Table 15: Detailed scores for each tasks in **XR** scenarios.

| Models | Setting | Static | Sequential | Prediction | Conversation1 | Conversation2 | Average |
|---|---|---|---|---|---|---|---|
| Gemini-Pro-1.5 | R. | 2.892 | 2.505 | 3.543 | 3.222 | 3.611 | 3.154 |
| | E. | 2.814 | 2.163 | 3.510 | 3.108 | 3.455 | 3.006 |
| Qwen-VL-Max | R. | 2.047 | 1.968 | 2.712 | 2.879 | 3.132 | 2.469 |
| | E. | 2.125 | 1.973 | 2.658 | 2.760 | 3.029 | 2.499 |
| | H. | 1.886 | 1.920 | 2.656 | 2.727 | 3.012 | 2.373 |
| GPT-4V | R. | **2.934** | 2.668 | 3.392 | 3.291 | 3.714 | 3.200 |
| | E. | 2.222 | 2.153 | 3.310 | 3.151 | 3.618 | 2.892 |
| | H. | 2.893 | 2.778 | 3.538 | **3.364** | 3.747 | **3.265** |
| | C.C. | 1.744 | 2.412 | 3.327 | 3.080 | 3.485 | 2.809 |
| | D.C. | 2.427 | 2.409 | 3.518 | 3.176 | **3.749** | 3.056 |
| | H.+D.C. | 2.775 | 2.635 | **3.580** | 3.235 | 3.734 | 3.191 |
| GPT-4o | H. | 2.871 | **2.745** | 3.370 | 3.596 | 3.836 | 3.285 |
| ChatUnivi | - | 1.660 | 1.420 | 2.205 | 2.250 | 3.270 | 2.161 |
| Minigpt4Video | - | 1.225 | 1.161 | 1.610 | 1.347 | 1.465 | 1.362 |
| VideoChat2 | - | 1.654 | 1.547 | 2.192 | 2.099 | 2.529 | 2.005 |
| GUI-Vid | - | 2.444 | 2.147 | 3.347 | 2.836 | 3.036 | 2.764 |

Table 16: Detailed scores for each tasks in **Multi-windows** scenarios.

| Models | Setting | Static | Sequential | Prediction | Conversation1 | Conversation2 | Average |
|---|---|---|---|---|---|---|---|
| Gemini-Pro-1.5 | R. | 2.538 | 2.410 | 3.296 | 3.152 | 3.402 | 2.959 |
| | E. | 2.545 | 2.049 | 2.972 | 2.930 | 3.389 | 2.777 |
| Qwen-VL-Max | R. | 1.793 | 1.872 | 2.770 | 2.897 | 3.122 | 2.432 |
| | E. | 1.866 | 1.780 | 2.730 | 2.627 | 3.105 | 2.362 |
| | H. | 1.884 | 1.969 | 2.913 | 2.689 | 3.104 | 2.490 |
| GPT-4V | R. | **3.185** | 2.655 | 3.745 | 3.699 | 3.973 | 3.452 |
| | E. | 2.902 | 2.406 | 3.636 | 3.420 | 3.729 | 3.219 |
| | H. | 3.000 | 2.952 | 3.801 | 3.597 | 3.889 | 3.449 |
| | C.C. | 2.097 | 2.973 | 3.774 | 3.331 | 3.621 | 3.160 |
| | D.C. | 2.671 | 2.979 | 3.849 | 3.466 | 3.822 | 3.358 |
| | H.+D.C. | 3.037 | **3.162** | **4.079** | 3.748 | 4.036 | 3.617 |
| GPT-4o | H. | 3.108 | 3.106 | 3.829 | **4.043** | **4.188** | **3.654** |
| ChatUnivi | - | 1.658 | 1.623 | 2.514 | 2.384 | 3.199 | 2.275 |
| Minigpt4Video | - | 1.205 | 1.186 | 1.690 | 1.400 | 1.801 | 1.457 |
| VideoChat2 | - | 1.754 | 1.774 | 2.479 | 2.420 | 2.699 | 2.222 |
| GUI-Vid | - | 2.485 | 2.067 | 3.537 | 2.954 | 3.247 | 2.861 |

Table 17: Detailed scores for each tasks in **IOS** scenarios.

| Models | Setting | Static | Sequential | Prediction | Conversation1 | Conversation2 | Average |
|---|---|---|---|---|---|---|---|
| Gemini-Pro-1.5 | R. | 3.076 | 2.637 | 3.370 | 3.366 | 3.615 | 3.213 |
| | E. | 2.852 | 2.356 | 3.137 | 3.126 | 3.566 | 3.007 |
| Qwen-VL-Max | R. | 2.438 | 2.244 | 2.923 | 3.102 | 3.273 | 2.779 |
| | E. | 2.303 | 2.150 | 2.614 | 3.145 | 3.264 | 2.659 |
| | H. | 1.884 | 1.969 | 2.913 | 2.689 | 3.104 | 2.490 |
| GPT-4V | R. | **3.364** | **3.080** | **3.684** | 3.766 | **4.184** | **3.614** |
| | E. | 3.209 | 2.774 | 3.545 | 3.611 | 4.006 | 3.427 |
| | H. | 3.107 | 2.830 | 3.631 | 3.680 | 4.011 | 3.453 |
| | C.C. | 1.788 | 2.291 | 3.511 | 3.212 | 3.542 | 2.868 |
| | D.C. | 2.751 | 2.732 | 3.654 | 3.642 | 3.842 | 3.324 |
| | H.+D.C. | 3.090 | 2.965 | 3.740 | 3.786 | 3.994 | 3.516 |
| GPT-4o | H. | 3.183 | 2.993 | 3.460 | **4.050** | 4.141 | 3.558 |
| ChatUnivi | - | 1.771 | 1.642 | 2.408 | 2.559 | 3.307 | 2.337 |
| Minigpt4Video | - | 1.291 | 1.219 | 1.698 | 1.556 | 1.737 | 1.501 |
| VideoChat2 | - | 1.955 | 1.803 | 2.145 | 2.315 | 2.626 | 2.169 |
| GUI-Vid | - | 2.262 | 2.133 | 3.401 | 2.843 | 3.224 | 2.773 |

Table 18: Detailed scores for each tasks in **Android** scenarios.

| Models | Setting | Static | Sequential | Prediction | Conversation1 | Conversation2 | Average |
|---|---|---|---|---|---|---|---|
| Gemini-Pro-1.5 | E. | 2.703 | 2.460 | 3.157 | 3.642 | 3.881 | 3.168 |
| Qwen-VL-Max | R. | 1.887 | 1.804 | 2.398 | 2.823 | 3.056 | 2.309 |
| | E. | 1.785 | 1.630 | 2.311 | 2.605 | 3.233 | 2.277 |
| GPT-4V | R. | **3.116** | 3.047 | **3.477** | 3.924 | 4.008 | 3.515 |
| | E. | 2.705 | 2.470 | 3.175 | 3.647 | 3.885 | 3.176 |
| | C.C. | 2.092 | 2.243 | 3.139 | 3.443 | 3.782 | 2.939 |
| | D.C. | 3.015 | 2.890 | 3.357 | 3.883 | 3.990 | 3.427 |
| GPT-4o | H. | 3.057 | **3.220** | 3.373 | **3.981** | **4.186** | **3.561** |
| ChatUnivi | - | 1.835 | 1.654 | 2.317 | 2.712 | 3.433 | 2.390 |
| Minigpt4Video | - | 1.183 | 1.159 | 1.507 | 1.342 | 1.521 | 1.342 |
| VideoChat2 | - | 1.732 | 1.754 | 2.125 | 2.340 | 2.645 | 2.119 |
| GUI-Vid | - | 2.010 | 1.928 | 3.053 | 2.755 | 3.105 | 2.572 |

Table 19: Detailed BLEU and BERTScore (B.S.) in **Software** scenarios.

| Models | Setting | Static | | Sequential | | Prediction | | Description | | Caption | | Conversation | | Avg. | |
|---|---|---|---|---|---|---|---|---|---|---|---|---|---|---|---|
| | | BLEU | B.S. | BLEU | B.S. | BLEU | B.S. | BLEU | B.S. | BLEU | B.S. | BLEU | B.S. | BLEU | B.S. |
| Gemini-Pro-1.5 | R. | 0.109 | 0.789 | 0.150 | 0.720 | 0.078 | 0.680 | **0.056** | 0.716 | 0.016 | **0.605** | 0.122 | 0.761 | 0.089 | 0.712 |
| | E. | 0.093 | 0.758 | 0.134 | 0.699 | 0.072 | 0.659 | 0.046 | 0.682 | 0.011 | 0.558 | 0.106 | 0.747 | 0.077 | 0.684 |
| Qwen-VL-Max | R. | 0.085 | 0.698 | 0.101 | 0.649 | 0.064 | 0.576 | 0.010 | 0.521 | 0.008 | 0.443 | 0.121 | 0.749 | 0.065 | 0.606 |
| | E. | 0.094 | 0.704 | 0.103 | 0.633 | 0.062 | 0.595 | 0.009 | 0.524 | 0.006 | 0.437 | 0.113 | 0.739 | 0.065 | 0.605 |
| | H. | 0.081 | 0.676 | 0.098 | 0.620 | 0.067 | 0.596 | 0.009 | 0.504 | 0.004 | 0.429 | 0.117 | 0.743 | 0.063 | 0.595 |
| GPT-4V | R. | **0.162** | **0.814** | 0.206 | 0.753 | **0.190** | **0.739** | 0.041 | 0.676 | **0.033** | 0.581 | **0.181** | 0.793 | **0.136** | 0.726 |
| | E. | 0.161 | 0.792 | 0.191 | 0.726 | 0.175 | 0.724 | 0.030 | 0.609 | 0.017 | 0.486 | 0.165 | 0.786 | 0.123 | 0.687 |
| | H. | 0.153 | 0.805 | 0.194 | 0.737 | 0.183 | 0.731 | 0.037 | 0.639 | 0.025 | 0.537 | 0.179 | 0.791 | 0.129 | 0.707 |
| GPT-4o | H. | 0.131 | 0.806 | **0.212** | **0.776** | 0.147 | 0.728 | 0.041 | 0.711 | 0.018 | 0.575 | 0.159 | **0.803** | 0.118 | **0.733** |
| ChatUnivi | - | 0.097 | 0.697 | 0.074 | 0.581 | 0.101 | 0.619 | 0.005 | 0.409 | 0.000 | 0.195 | 0.084 | 0.723 | 0.060 | 0.537 |
| Minigpt4Video | - | 0.019 | 0.516 | 0.022 | 0.470 | 0.029 | 0.516 | 0.000 | 0.399 | 0.000 | 0.249 | 0.013 | 0.510 | 0.014 | 0.443 |
| VideoChat2 | - | 0.095 | 0.698 | 0.080 | 0.595 | 0.076 | 0.574 | 0.004 | 0.341 | 0.000 | 0.193 | 0.100 | 0.733 | 0.059 | 0.523 |
| GUI-Vid | - | 0.142 | 0.758 | 0.145 | 0.681 | 0.114 | 0.698 | 0.049 | 0.658 | 0.004 | 0.519 | 0.093 | 0.717 | 0.091 | 0.672 |

Table 20: Detailed BLEU and BERTScore (B.S.) in **Website** scenarios.

| Models | Setting | Static | | Sequential | | Prediction | | Description | | Caption | | Conversation | | Avg. | |
|---|---|---|---|---|---|---|---|---|---|---|---|---|---|---|---|
| | | BLEU | B.S. | BLEU | B.S. | BLEU | B.S. | BLEU | B.S. | BLEU | B.S. | BLEU | B.S. | BLEU | B.S. |
| Gemini-Pro-1.5 | R. | 0.113 | 0.793 | 0.145 | 0.727 | 0.083 | 0.676 | **0.054** | 0.720 | 0.016 | **0.664** | 0.098 | 0.736 | 0.085 | 0.719 |
| | E. | 0.095 | 0.754 | 0.121 | 0.681 | 0.079 | 0.661 | 0.041 | 0.676 | 0.011 | 0.602 | 0.092 | 0.725 | 0.073 | 0.683 |
| Qwen-VL-Max | R. | 0.099 | 0.728 | 0.099 | 0.634 | 0.080 | 0.610 | 0.008 | 0.519 | 0.005 | 0.471 | 0.085 | 0.694 | 0.063 | 0.609 |
| | E. | 0.083 | 0.710 | 0.101 | 0.631 | 0.093 | 0.611 | 0.011 | 0.503 | 0.004 | 0.469 | 0.099 | 0.709 | 0.065 | 0.605 |
| | H. | 0.079 | 0.693 | 0.089 | 0.597 | 0.093 | 0.606 | 0.009 | 0.488 | 0.007 | 0.449 | 0.103 | 0.705 | 0.063 | 0.590 |
| GPT-4V | R. | 0.173 | **0.830** | **0.241** | 0.765 | 0.205 | 0.751 | 0.040 | 0.694 | 0.032 | 0.645 | 0.164 | 0.763 | 0.142 | 0.741 |
| | E. | 0.159 | 0.802 | 0.204 | 0.727 | 0.202 | 0.727 | 0.033 | 0.648 | 0.031 | 0.590 | 0.149 | 0.757 | 0.130 | 0.708 |
| | H. | **0.182** | 0.823 | 0.234 | **0.771** | **0.213** | **0.758** | 0.043 | 0.696 | **0.041** | 0.660 | **0.165** | **0.768** | **0.147** | **0.746** |
| GPT-4o | H. | 0.141 | 0.813 | 0.219 | 0.768 | 0.199 | 0.731 | 0.054 | 0.700 | 0.026 | 0.602 | 0.146 | 0.755 | 0.131 | 0.728 |
| ChatUnivi | - | 0.078 | 0.645 | 0.068 | 0.581 | 0.102 | 0.607 | 0.008 | 0.399 | 0.000 | 0.192 | 0.061 | 0.661 | 0.053 | 0.514 |
| Minigpt4Video | - | 0.022 | 0.527 | 0.016 | 0.448 | 0.027 | 0.501 | 0.000 | 0.344 | 0.000 | 0.186 | 0.011 | 0.522 | 0.013 | 0.421 |
| VideoChat2 | - | 0.073 | 0.619 | 0.075 | 0.579 | 0.049 | 0.511 | 0.004 | 0.328 | 0.000 | 0.167 | 0.067 | 0.678 | 0.045 | 0.480 |
| GUI-Vid | - | 0.114 | 0.731 | 0.158 | 0.674 | 0.129 | 0.694 | 0.049 | 0.667 | 0.002 | 0.553 | 0.075 | 0.681 | 0.088 | 0.667 |

Table 21: Detailed BLEU and BERTScore (B.S.) in **XR** scenarios.

| Models | Setting | Static | | Sequential | | Prediction | | Description | | Caption | | Conversation | | Avg. | |
|---|---|---|---|---|---|---|---|---|---|---|---|---|---|---|---|
| | | BLEU | B.S. | BLEU | B.S. | BLEU | B.S. | BLEU | B.S. | BLEU | B.S. | BLEU | B.S. | BLEU | B.S. |
| Gemini-Pro-1.5 | R. | 0.088 | 0.772 | 0.101 | 0.678 | 0.070 | 0.678 | 0.026 | **0.650** | 0.002 | 0.463 | 0.082 | 0.733 | 0.062 | 0.662 |
| | E. | 0.073 | 0.760 | 0.090 | 0.651 | 0.062 | 0.666 | 0.015 | 0.618 | 0.002 | 0.449 | 0.084 | 0.720 | 0.054 | 0.644 |
| Qwen-VL-Max | R. | 0.069 | 0.703 | 0.075 | 0.602 | 0.049 | 0.601 | 0.006 | 0.486 | 0.000 | 0.338 | 0.117 | 0.738 | 0.053 | 0.578 |
| | E. | 0.048 | 0.689 | 0.079 | 0.657 | 0.058 | 0.605 | 0.005 | 0.498 | 0.000 | 0.359 | 0.112 | 0.739 | 0.050 | 0.591 |
| | H. | 0.051 | 0.651 | 0.073 | 0.593 | 0.044 | 0.591 | 0.004 | 0.493 | 0.001 | 0.357 | 0.101 | 0.726 | 0.046 | 0.569 |
| GPT-4V | R. | 0.093 | 0.794 | 0.169 | 0.715 | 0.165 | 0.736 | 0.028 | 0.625 | 0.006 | 0.457 | 0.147 | 0.768 | 0.101 | 0.683 |
| | E. | 0.085 | 0.726 | 0.131 | 0.665 | 0.162 | 0.724 | 0.020 | 0.541 | 0.003 | 0.382 | 0.141 | 0.760 | 0.090 | 0.633 |
| | H. | 0.091 | 0.797 | **0.181** | **0.732** | **0.180** | **0.744** | 0.027 | 0.630 | **0.006** | 0.471 | **0.154** | **0.773** | **0.106** | **0.691** |
| GPT-4o | H. | 0.077 | **0.800** | 0.154 | 0.717 | 0.153 | 0.718 | 0.020 | 0.615 | **0.006** | 0.468 | 0.138 | 0.759 | 0.091 | 0.680 |
| ChatUnivi | - | 0.083 | 0.686 | 0.061 | 0.538 | 0.091 | 0.575 | 0.006 | 0.475 | 0.000 | 0.282 | 0.086 | 0.693 | 0.054 | 0.541 |
| Minigpt4Video | - | 0.014 | 0.545 | 0.016 | 0.466 | 0.027 | 0.502 | 0.001 | 0.453 | 0.000 | 0.262 | 0.013 | 0.474 | 0.012 | 0.450 |
| VideoChat2 | - | 0.077 | 0.679 | 0.079 | 0.595 | 0.073 | 0.577 | 0.004 | 0.378 | 0.000 | 0.211 | 0.101 | 0.721 | 0.056 | 0.527 |
| GUI-Vid | - | **0.096** | 0.754 | 0.149 | 0.689 | 0.131 | 0.700 | **0.051** | 0.637 | 0.003 | 0.460 | 0.082 | 0.705 | 0.085 | 0.657 |

Table 22: Detailed BLEU and BERTScore (B.S.) in **IOS** scenarios.

| Models | Setting | Static | | Sequential | | Prediction | | Description | | Caption | | Conversation | | Avg. | |
|---|---|---|---|---|---|---|---|---|---|---|---|---|---|---|---|
| | | BLEU | B.S. | BLEU | B.S. | BLEU | B.S. | BLEU | B.S. | BLEU | B.S. | BLEU | B.S. | BLEU | B.S. |
| Gemini-Pro-1.5 | R. | 0.108 | 0.797 | 0.142 | 0.717 | 0.080 | 0.682 | **0.075** | **0.714** | 0.011 | **0.602** | 0.117 | 0.746 | 0.089 | 0.710 |
| | E. | 0.099 | 0.768 | 0.136 | 0.700 | 0.075 | 0.655 | 0.066 | 0.695 | 0.011 | 0.592 | 0.113 | 0.743 | 0.083 | 0.692 |
| Qwen-VL-Max | R. | 0.087 | 0.704 | 0.098 | 0.650 | 0.112 | 0.639 | 0.009 | 0.519 | 0.003 | 0.465 | 0.106 | 0.725 | 0.069 | 0.617 |
| | E. | 0.075 | 0.638 | 0.095 | 0.647 | 0.094 | 0.600 | 0.009 | 0.512 | 0.009 | 0.475 | 0.103 | 0.712 | 0.064 | 0.597 |
| | H. | 0.080 | 0.632 | 0.083 | 0.589 | 0.092 | 0.617 | 0.013 | 0.520 | 0.007 | 0.452 | 0.099 | 0.703 | 0.062 | 0.585 |
| GPT-4V | R. | **0.159** | **0.824** | **0.224** | **0.772** | 0.206 | **0.766** | 0.040 | 0.673 | **0.030** | 0.579 | **0.174** | **0.777** | **0.139** | **0.732** |
| | E. | 0.149 | 0.813 | 0.201 | 0.752 | **0.207** | 0.746 | 0.035 | 0.659 | 0.017 | 0.566 | 0.160 | 0.762 | 0.128 | 0.716 |
| | H. | 0.156 | 0.805 | 0.205 | 0.745 | 0.203 | 0.748 | 0.034 | 0.644 | 0.025 | 0.559 | 0.159 | 0.763 | 0.130 | 0.711 |
| GPT-4o | H. | 0.137 | 0.802 | 0.196 | 0.761 | 0.199 | 0.732 | 0.035 | 0.683 | 0.022 | 0.533 | 0.154 | 0.774 | 0.124 | 0.714 |
| ChatUnivi | - | 0.093 | 0.679 | 0.085 | 0.604 | 0.106 | 0.616 | 0.005 | 0.437 | 0.000 | 0.258 | 0.076 | 0.698 | 0.061 | 0.548 |
| Minigpt4Video | - | 0.026 | 0.547 | 0.026 | 0.513 | 0.035 | 0.548 | 0.001 | 0.411 | 0.000 | 0.236 | 0.015 | 0.529 | 0.017 | 0.464 |
| VideoChat2 | - | 0.089 | 0.683 | 0.078 | 0.605 | 0.061 | 0.555 | 0.002 | 0.355 | 0.000 | 0.190 | 0.086 | 0.710 | 0.053 | 0.516 |
| GUI-Vid | - | 0.114 | 0.725 | 0.144 | 0.693 | 0.123 | 0.700 | 0.048 | 0.641 | 0.002 | 0.518 | 0.083 | 0.686 | 0.085 | 0.661 |

Table 23: Detailed BLEU and BERTScore (B.S.) in **Android** scenarios.

| Models | Setting | Static | | Sequential | | Prediction | | Description | | Caption | | Conversation | | Avg. | |
|---|---|---|---|---|---|---|---|---|---|---|---|---|---|---|---|
| | | BLEU | B.S. | BLEU | B.S. | BLEU | B.S. | BLEU | B.S. | BLEU | B.S. | BLEU | B.S. | BLEU | B.S. |
| Gemini-Pro-1.5 | E. | 0.089 | 0.771 | 0.189 | 0.704 | 0.189 | 0.710 | 0.023 | 0.619 | 0.016 | 0.570 | 0.149 | 0.749 | 0.109 | 0.687 |
| Qwen-VL-Max | R. | 0.041 | 0.640 | 0.084 | 0.528 | 0.066 | 0.549 | 0.008 | 0.484 | 0.004 | 0.445 | 0.089 | 0.673 | 0.049 | 0.553 |
| | E. | 0.037 | 0.634 | 0.074 | 0.498 | 0.065 | 0.541 | 0.005 | 0.443 | 0.003 | 0.383 | 0.089 | 0.683 | 0.045 | 0.530 |
| GPT-4V | R. | **0.106** | **0.809** | **0.242** | **0.757** | **0.210** | **0.733** | 0.029 | 0.653 | **0.028** | **0.619** | **0.170** | **0.763** | **0.131** | **0.723** |
| | E. | 0.089 | 0.771 | 0.192 | 0.705 | 0.190 | 0.713 | 0.023 | 0.619 | 0.016 | 0.571 | 0.150 | 0.750 | 0.110 | 0.688 |
| GPT-4o | H. | 0.075 | **0.809** | 0.241 | 0.755 | 0.188 | 0.719 | **0.038** | **0.677** | 0.014 | 0.581 | 0.137 | 0.747 | 0.116 | 0.715 |
| ChatUnivi | - | 0.076 | 0.675 | 0.079 | 0.588 | 0.096 | 0.594 | 0.007 | 0.482 | 0.001 | 0.368 | 0.063 | 0.670 | 0.054 | 0.563 |
| Minigpt4Video | - | 0.017 | 0.416 | 0.013 | 0.369 | 0.019 | 0.405 | 0.000 | 0.279 | 0.000 | 0.103 | 0.010 | 0.392 | 0.010 | 0.327 |
| VideoChat2 | - | 0.057 | 0.641 | 0.077 | 0.560 | 0.063 | 0.523 | 0.004 | 0.402 | 0.000 | 0.272 | 0.075 | 0.654 | 0.046 | 0.509 |
| GUI-Vid | - | 0.083 | 0.682 | 0.130 | 0.628 | 0.126 | 0.644 | 0.023 | 0.500 | 0.001 | 0.393 | 0.071 | 0.659 | 0.072 | 0.584 |

Table 24: Detailed BLEU and BERTScore (B.S.) in **Multiple-windows** scenarios.

| Models | Setting | Static | | Sequential | | Prediction | | Description | | Caption | | Conversation | | Avg. | |
|---|---|---|---|---|---|---|---|---|---|---|---|---|---|---|---|
| | | BLEU | B.S. | BLEU | B.S. | BLEU | B.S. | BLEU | B.S. | BLEU | B.S. | BLEU | B.S. | BLEU | B.S. |
| Gemini-Pro-1.5 | R. | 0.113 | 0.739 | 0.126 | 0.693 | 0.086 | 0.658 | **0.061** | **0.685** | 0.012 | 0.586 | 0.090 | 0.674 | 0.081 | 0.673 |
| | E. | 0.106 | 0.728 | 0.131 | 0.680 | 0.072 | 0.622 | 0.055 | 0.655 | 0.015 | 0.550 | 0.084 | 0.679 | 0.077 | 0.652 |
| Qwen-VL-Max | R. | 0.079 | 0.599 | 0.076 | 0.591 | 0.080 | 0.595 | 0.002 | 0.444 | 0.006 | 0.370 | 0.072 | 0.666 | 0.053 | 0.544 |
| | E. | 0.064 | 0.609 | 0.087 | 0.567 | 0.089 | 0.608 | 0.003 | 0.445 | 0.004 | 0.398 | 0.073 | 0.647 | 0.053 | 0.546 |
| | H. | 0.089 | 0.634 | 0.078 | 0.580 | 0.093 | 0.612 | 0.003 | 0.409 | 0.005 | 0.344 | 0.080 | 0.656 | 0.058 | 0.539 |
| GPT-4V | R. | 0.172 | **0.800** | 0.186 | 0.737 | 0.212 | 0.745 | 0.040 | 0.671 | **0.021** | 0.592 | 0.145 | **0.728** | 0.129 | 0.712 |
| | E. | 0.160 | 0.763 | 0.169 | 0.703 | 0.198 | 0.759 | 0.034 | 0.621 | 0.012 | 0.527 | 0.116 | 0.709 | 0.115 | 0.680 |
| | H. | **0.173** | 0.781 | **0.196** | 0.748 | **0.220** | **0.775** | 0.046 | 0.672 | **0.021** | 0.577 | 0.133 | 0.724 | **0.132** | 0.713 |
| GPT-4o | H. | 0.156 | 0.792 | 0.185 | **0.754** | 0.213 | 0.769 | 0.040 | 0.683 | 0.019 | 0.588 | 0.121 | 0.717 | 0.122 | **0.717** |
| ChatUnivi | - | 0.076 | 0.628 | 0.063 | 0.573 | 0.103 | 0.605 | 0.009 | 0.413 | 0.000 | 0.191 | 0.057 | 0.643 | 0.051 | 0.509 |
| Minigpt4Video | - | 0.015 | 0.504 | 0.024 | 0.473 | 0.023 | 0.527 | 0.001 | 0.326 | 0.000 | 0.155 | 0.009 | 0.469 | 0.012 | 0.409 |
| VideoChat2 | - | 0.098 | 0.657 | 0.081 | 0.593 | 0.067 | 0.577 | 0.007 | 0.344 | 0.000 | 0.162 | 0.065 | 0.654 | 0.053 | 0.498 |
| GUI-Vid | - | 0.128 | 0.737 | 0.144 | 0.664 | 0.133 | 0.721 | 0.041 | 0.605 | 0.004 | 0.452 | 0.058 | 0.644 | 0.084 | 0.637 |

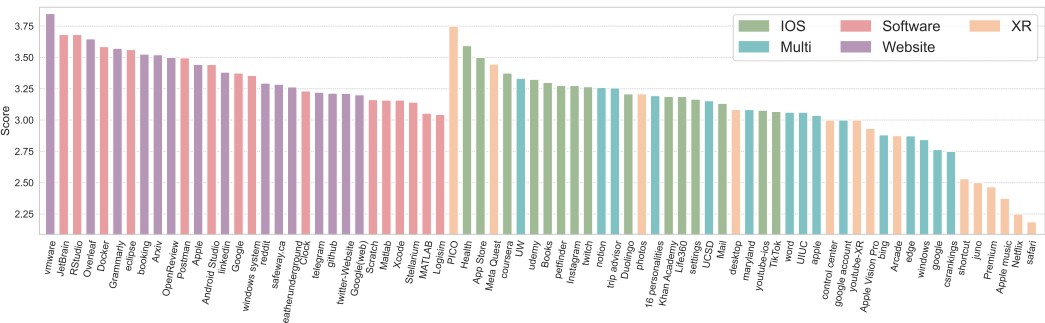

Figure 14: Fine-grained performance of Gemini-Pro-1.5 in each software and website.

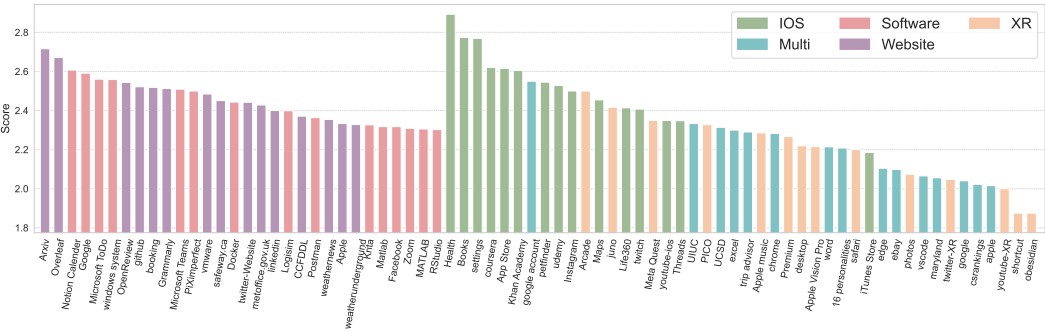

Figure 15: Fine-grained performance of Qwen-VL-Max in each software and website.

# F Prompts

In this section, we provide detailed prompts for models and human annotators. Figure 17 shows the guideline of human annotation, Figure 16 shows the prompt for leveraging LLMs to refine grammarly mistakes and polish sentence for human annotations. Figure 18, Figure 19, and Figure 20 present the prompt for Human-MLLM collaboration method to generate GUI-orientaed tasks. Figure 21 illustrate the prompt for benchmarking MLLMs, different GUI scenarios and different QA type has different prompt. Figure 22 and Figure 23 show prompt for LLM-as-a-Judge for free-form as well as conversational tasks and multiple-choice QA respectively.

---

**Refining Human Annotation on Goal and Sub-goal**

```
As an expert in English, please refine the following English
instructions (or objectives) into a polished phrase or a concise
sentence.
Avoid including irrelevant content and provide the polished output
directly.
Here is the English sentence:  {string}
```

Figure 16: Refining Human Annotation on Goal and Sub-goal.

**Main Interface**
1. Video List Panel (Left Panel): Displays a list of loaded video files. Each video file is shown with its name for identification.
2. Video Display Area (Center Panel): Shows the currently selected video for playback and annotation.
3. Control Settings (Right Panel):
Operating System: Select the operating system of the machine where the video was recorded.
Full Screen: Toggle full screen mode for the video display.
Multi-application?: Indicate if multiple applications in the video.
Application/Website: Enter the name of the application or website being used in the video.
User Goal: Enter the goal of the user performing the annotation.
4. Playback and Annotation Controls (Bottom Panel)
Annotate: Open a annotation window to add a new keyframe annotation.
Play: Starts or pauses the video playback.
Load Video: Allows you to load a single video file.
Load Video Folder: Allows to load multiple video files from a folder.
Previous Video / Next Video: Navigate through the loaded video files.
Save to JSON: Save the annotations in a JSON format.

**Annotation Window**
1. Mouse Action: Select a type of mouse action (e.g. click, drag).
2. Keyboard Action: Select the type of keyboard action (e.g., typing, key press).
3. Keyboard Operation Record: Enter details of the keyboard operation, if any.
4. sub-action Purpose: Describe the purpose of the action being annotated.

**How to Use**
**Loading Videos**
1. Load Multiple Videos
Click on the Load Video Folder button.
Select the folder containing your video files.
All video files in the folder will be loaded and listed in the Video List Panel.

**Playing Videos**
Select a video from the Video List Panel. Click the Play button to start or pause the video.

**Annotating Videos**
1. Start Annotation
Pause the video at the desired frame.
Click the Annotate button to open the annotation window.
2. Annotation Window
Select the Mouse Action Type and Keyboard Action Type from the dropdown menus.
If there is a keyboard action, enter the details in the Keyboard Operation Record field.
Describe the action's purpose in the Sub-action Purpose field.
Click OK to save the annotation.

**Saving Annotations**
Once all annotations are completed, click the Save to JSON button.

Figure 17: Guideline for Human Annotation.

**(Part 1) GPT-4V Generating GUI-orientated Tasks**

You are an AI visual assistant. This is a video of a mobile GUI, which I've divided into multiple frames and sent to you. Please provide a detailed description of what occurs throughout the entire video, focusing on the changes in the GUI elements or scenes rather than static aspects of a single frame. The detailed description should be placed under the key 'Description'. Based on your description, please design the following tasks:

Generate a precise caption for the video. This caption should encapsulate the main activities or changes observed throughout the video sequence. Place this caption under the key 'Caption'.

Create a free-form QA question related to the video's static GUI content, along with its answer. The question should delve into the details or changes in the static GUI elements or scenes captured in the video. The QA task should be nested under the key 'static QA', with 'Question' and 'Answer' as subkeys.

Develop a multiple-choice QA question about the video, with four options: one correct answer and three incorrect or irrelevant options. This task should assess the understanding of specific elements retieval or changes depicted in the video. Structure this task under the key 'MCQA', with 'Question' detailing the query, 'Options' listing the four choices including one correct answer, and 'Correct Answer' specifying the correct option, denoted, for example, as {[[B]]}.

Here are some key information of the video to help you understand the video comprehensively:

System: {item['system']}

Application: {item['app']}

Summary of the video: {item['goal']}

Key Operation/Sub goal in the video: {[i['sub_goal'] for i in item['keyframes']]}

Notice: Ensure that the questions you design for these tasks are answerable and the answers can be deduced from the GUI video content. The answerable question should be designed as difficult as possible. The tasks should be unambiguous and the answers must be definitively correct based on your understanding of the video content. Only include questions that have definite answers: (1) one can see the content in the image that the question asks about and can answer confidently; (2) one can determine confidently from the image that it is not in the image. Do not ask any question that cannot be answered confidently.

Each of these tasks should focus on the dynamic aspect of the GUI elements or scenes. Provide detailed answers when answering complex questions. For example, give detailed examples or reasoning steps to make the content more convincing and well-organized. The answers should be in a tone that a visual AI assistant is seeing the image and answering the question.

For the free-form QA tasks, please ensure that the answers are as detailed and lengthy as possible, with no concern for length. You can include multiple paragraphs if necessary to provide a comprehensive and thorough response. Please structure your response using JSON format and specific keys mentioned in the task requirements.

Figure 18: (Part 1) GPT-4V Generating GUI-orientated Tasks.

**(Part 2) GPT-4V Generating GUI-orientated Tasks.**

You are an AI visual assistant. This is a video of a <Scene Name> GUI, which I've divided into multiple frames and sent to you. Please provide a detailed description of what occurs throughout the entire video, focusing on the changes in the GUI elements or scenes rather than static aspects of a single frame. The detailed description should be placed under the key 'Description'. Based on your description, please design the following tasks:

A Sequential QA task: Design a question that requires understanding the sequence of GUI element changes or scene transformations in the video. The question should be free-form and necessitate the use of temporal information from the sequential images. The task should be structured under the key 'Sequential-QA' with subkeys 'Question' and 'Answer'.

A Next Stage Prediction task: Formulate a question that asks about the subsequent state or event following a certain frame in the video. The question should be designed in a free-form manner and predict future GUI elements or scene changes, structured under the key 'Prediction' with subkeys 'Question' and 'Answer'.

A two-round dialogue task: Create a dialogue with two rounds of interaction. The first round includes a user instruction and an assistant response, and the second round's user instruction should be based on the response from the first round. Both rounds should be free-form and nested under the key 'Conversation', with subkeys 'User 1', 'Assistant 1', 'User 2', and 'Assistant 2'.

A reasoning task: Design a multi-choice QA task that requires reasoning to identify the correct answer from four options. This task should test the reasoning ability to infer or deduce information that is not explicitly provided. It should be structured under the key 'Reasoning', with subkeys 'Question', 'Options', and 'Correct Answer'.

Here are some key information of the video to help you understand the video comprehensively:

System: {item['system']}
Application: {item['app']}
Summary of the video: {item['goal']}
Key Operation/Sub goal in the video: {[i['sub_goal'] for i in item['keyframes']]}

Figure 19: (Part 2) GPT-4V Generating GUI-orientated Tasks.

**(Part 3) GPT-4V Generating GUI-orientated Tasks.**

```
 Notice:  Ensure that the questions you design for these tasks are
answerable and the answers can be deduced from the GUI video content.
The answerable question should be designed as difficult as possible.
The tasks should be unambiguous and the answers must be definitively
correct based on your understanding of the video content.  Only in-
clude questions that have definite answers:  (1) one can see the con-
tent in the image that the question asks about and can answer confi-
dently; (2) one can determine confidently from the image that it is
not in the image.  Do not ask any question that cannot be answered con-
fidently.
Each of these tasks should focus on the dynamic aspect of the GUI el-
ements or scenes, with each answerable task as difficult as possible.
Provide detailed answers when answering complex questions.  For ex-
ample, give detailed examples or reasoning steps to make the content
more convincing and well-organized.  The answers should be in a tone
that a visual AI assistant is seeing the image and answering the ques-
tion.
For the free-form QA tasks, please ensure that the answers are as de-
tailed and lengthy as possible, with no concern for length.  You can
include multiple paragraphs if necessary to provide a comprehensive
and thorough response.  Please structure your response using JSON for-
mat and specific keys mentioned in the task requirements.
```

Figure 20: (Part 3) GPT-4V Generating GUI-orientated Tasks.

## Prompts for Benchmarking MLLMs

"XR": "You are an AI visual assistant. Here are sequential images of Mixed-Reality combining GUI interface and real world, which are selected from a GUI video.",
"software": "You are an AI visual assistant. Here are sequential GUI interface images of a specific software, which are selected from a GUI video.",
"website": "You are an AI visual assistant. Here are sequential GUI interface images of a desktop website, which are selected from a GUI video.",
"mobile": "You are an AI visual assistant. Here are sequential GUI mobile interface images, which are selected from a GUI video.",
"multi": "You are an AI visual assistant. Here are sequential GUI interface images of interaction among multiple softwares and websites, which are selected from a GUI video.",
"IOS": "You are an AI visual assistant. Here are sequential GUI IOS interface images, which are selected from a GUI video.",

"Sequential-QA": "This is a question about sequential information in sequential images.",
"Prediction": "This is a question about predicting the next action base on the previous actions in the sequential images.",
"Reasoning": "This is a multiple choice question with only one correct answer. This question may need multiple steps of reasoning according to the vision information in sequential images.",
"Description1": "Please give me a detail description of these sequential images.",
"Description2": "Offer a thorough analysis of these sequential images",
"Caption": "Please give me a concise caption of these sequential images.",
"static QA": "This is a question about static information such as text, icon, layout in these sequential images.",
"MCQA": "This is a multiple choice question with only one correct answer. This question may require sequential analysis ability to the vision information in these sequential images.",
"Conversation1": "Act as an assistant to answer the user's question in these sequential images.",
"Conversation2": "This is a multi-turn conversation task. You will be provide the first round conversation and act as an assistant to answer the user's question in the second round according to these sequential images."
Notice = "You can first provide an overall description of these sequential images, and then analyze the user's question according to the sequential images and description. Finally, give an answer based on this description and the image information. Please format your output in a Json format, with key 'Description' for the description of these sequential images, key 'Analysis' for your analysis on the user's question and key 'Answer' for your answer to the User's question."

Figure 21: Prompts for Benchmarking MLLMs.

**Prompt for LLM-as-a-Judge: Judging Free-form and Conversational Tasks**

You are an impartial judge. I will provide you with a question, a 'gold standard' answer, and a response that needs evaluation. Your task is to assess the quality of the response in comparison to the 'gold standard' answer. Please adhere to the following guidelines:

1. Start your evaluation by comparing the response to the 'gold standard' answer. Offer a brief explanation highlighting similarities and differences, focusing on relevance, accuracy, depth, and level of detail.
2. Conclude your evaluation with a score from 1 to 5, where 1 indicates the response is mostly irrelevant to the 'gold standard' answer, and 5 indicates it is very similar or equivalent.
3. Present your findings in JSON format, using 'Evaluation' for your textual analysis and 'Score' for the numerical assessment.
4. Ensure objectivity in your evaluation. Avoid biases and strive for an even distribution of scores across the spectrum of quality.
Your scoring must be as rigorous as possible and adhere to the following rules:
- Overall, the higher the quality of the model's response, the higher the score, with factual accuracy and meeting user needs being the most critical dimensions. These two factors largely dictate the final composite score.
- If the model's response is irrelevant to the question, contains fundamental factual errors, or generates harmful content, the total score must be 1.
- If the model's response has no severe errors and is essentially harmless, but of low quality and does not meet user needs, the total score should be 2.
- If the model's response generally meets user requirements but performs poorly in certain aspects with medium quality, the total score should be 3.
- If the model's response is close in quality to the reference answer and performs well in all dimensions, the total score should be 4.
- Only when the model's response surpasses the reference answer, fully addresses the user's problem and all needs, and nearly achieves a perfect score in all dimensions, can it receive a score between 5.
- As an example, the golden answer could receive a 4-5.

Here is the response for you to judge:
Question: {question}
Golden Answer: {golden_answer}
Response: {response}
Now, directly output your response in json format.

Figure 22: Prompt for LLM-as-a-Judge: Judging Free-form and Conversational Tasks .

**Prompt for LLM-as-a-Judge: Judging Multiple-Choice QA Tasks**

```
 You are a helpful assistant tasked with judging a Multiple Choice
Question Answering exercise.
I will provide a correct answer with only one option, and a response
that requires evaluation.
If the response matches the correct answer, simply output "Yes"; If it
does not, output "No".
Please avoid including any irrelevant information.
Here are some examples:

Example 1:
Question:  Based on the GUI video, why might the 'Loading' animation
continue without reaching the next stage?  A. The user has not yet
entered their login credentials.  B. There is a system update being
installed.  C. The server is taking time to authenticate the login cre-
dentials.  D. The 'Log In' button is malfunctioning.
Answer:  C
Response:  C. The server is taking time to authenticate the login cre-
dentials.
Output:  Yes

Example 2:
Question:  If the user wants to resume the group video call after
checking messages, what action should they take?  A. Turn their head
to the right.  B. Close the messaging app interface.  C. Say a voice
command to switch applications.  D. Turn their head to the left.
Answer:  A
Response:  B
Output:  No

Example 3:
Question:  What action does the user take to start playing music in
the video?  A. Closed the music player application B. Moved the music
player to a new position C. Clicked the play button D. Adjusted the
system volume
Answer:  [[B]]
Response:  C
Output:  No

Here is the question, answer, and response for you to judge:
Question:  {question}
Answer:  {answer}
Response:  {response}
Now, directly output "Yes" or "No".
```

Figure 23: Prompt for LLM-as-a-Judge: Judging Multiple-Choice QA Tasks.

## G    Case Study

In this section, we provide detailed case studies for six GUI scenarios, each divided into two parts. Figure 24 and Figure 25 show example frames and various tasks associated with them. Figure 26 and Figure 27 for IOS, Figure 28 and Figure 29 for multiple-windows interaction, Figure 32 and Figure 33 for website, and Figure 34 and Figure 35 for XR respectively.

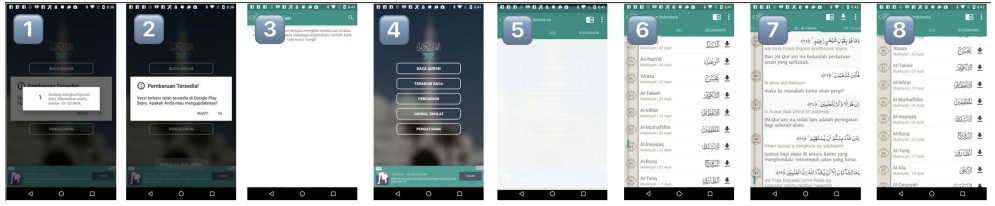

'Description1': "The video starts with a view of a mobile app interface, specifically an Al-Qur'an Indonesia app with a list of Surahs. A user scrolls through the Surah list, selects a specific Surah, and the view transitions to a page displaying the verses in Arabic and their Indonesian translation. The user then returns to the Surah list. Following this, a pop-up appears indicating a configuration process that takes approximately 10-20 seconds. After the configuration, another pop-up notifies of an available update from the Google Play Store, which the user can accept or postpone. The user dismisses this pop-up. The main menu is accessed, showing options such as 'Baca Quran', 'Terakhir Baca', 'Pencarian', 'Jadwal Sholat', and 'Pengaturan'. The user selects the 'Pencarian' option, which leads to a search page. Here, the user is instructed to enter a keyword to search in the translation database. Finally, the user navigates back to the main menu and then to the Surah list once more."

'Sequential-QA': 'Question': 'After selecting a specific Surah from the list, what does the user do next before a pop-up notification appears?'
'Answer': 'After selecting a Surah, the user views the verses in Arabic and their Indonesian translation. Then, the user returns to the Surah list before a pop-up notification appears.'

'Prediction': 'Question': 'What would likely be the next step in the app interface after the user is directed to the search page?'
'Answer': "After being directed to the search page, the likely next step would be for the user to enter a keyword into the search bar to find a translation or meaning within the app's database."

'Conversation': 'User 1': 'I need to find a specific verse in the Quran. How do I search for it?'
'Assistant 1': "You can search for a specific verse by tapping on the 'Pencarian' option in the main menu, which will take you to a search page where you can enter a keyword."
'User 2': 'I see. And how do I get back to the main menu from the search page?'
'Assistant 2': 'To return to the main menu from the search page, you can tap on the back arrow located at the top left corner of the screen.'

'Reasoning': 'Question': "If the user wants to read the verses of Surah 'Al-Infitar', what should they do after launching the app?"
'Options': ["A. Go to the settings and select 'Al-Infitar'.", "B. Scroll through the Surah list and select 'Al-Infitar'.", "C. Choose the 'Pencarian' option and type 'Al-Infitar'.", "D. Wait for a pop-up and select 'Al-Infitar' from there."]
'Correct Answer': "B. Scroll through the Surah list and select 'Al-Infitar'."

Figure 24: Case study for Android (part 1).

'Description2': "The video begins by displaying a mobile GUI with
a list of chapters from the Quran in Indonesian. Each chapter has a
downward arrow suggesting expandable content. As the video progresses,
a popup appears with a loading icon and a message in Indonesian indi-
cating a configuration is in progress, which takes about 10-20 seconds.
After this, another popup appears notifying of a new update available
on the Google Play Store with options to update or postpone. Subse-
quently, the screen shows a search interface where users can input
keywords for searching within the Quran's translated database. The
main menu is then accessed, with options such as 'Read Quran', 'Last
Read', 'Search', 'Prayer Schedule', and 'Settings'. The GUI transi-
tions back to the list of chapters, and a specific chapter, At-Takwir,
is selected. The video then displays the verses of this chapter, both
in Arabic and Indonesian translation, with an option to listen to the
audio. Finally, it navigates back to the list of chapters."
'Caption': "Navigating through a Quran app's GUI, interacting with
chapter lists, update notifications, search function, and viewing spe-
cific verses with translations."
'static QA': 'Question': 'What options are available in the main
menu of the mobile Quran application?'
'Answer': "The main menu of the mobile Quran application provides sev-
eral options for the user to choose from. These include 'BACA QURAN'
(Read Quran) for accessing the chapters to read, 'TERAKHIR BACA' (Last
Read) to resume reading from where the user left off last time, 'PEN-
CARIAN' (Search) to search the Quran's database for specific keywords,
'JADWAL SHOLAT' (Prayer Schedule) to check the prayer times, and 'PEN-
GATURAN' (Settings) to modify app settings. This menu provides a sim-
ple and efficient way for users to navigate through the app's features
and customize their reading and learning experience."
'MCQA': 'Question': 'What happens after the user is notified about
the new update available on the Google Play Store?'
'Options': 'A': 'The app closes automatically.', 'B': 'The search in-
terface is displayed.', 'C': 'The list of chapters disappears.', 'D':
'An advertisement for shopping deals is shown.'
'Correct Answer': '[[B]] The search interface is displayed.'

Figure 25: Case study for Android (part 2).

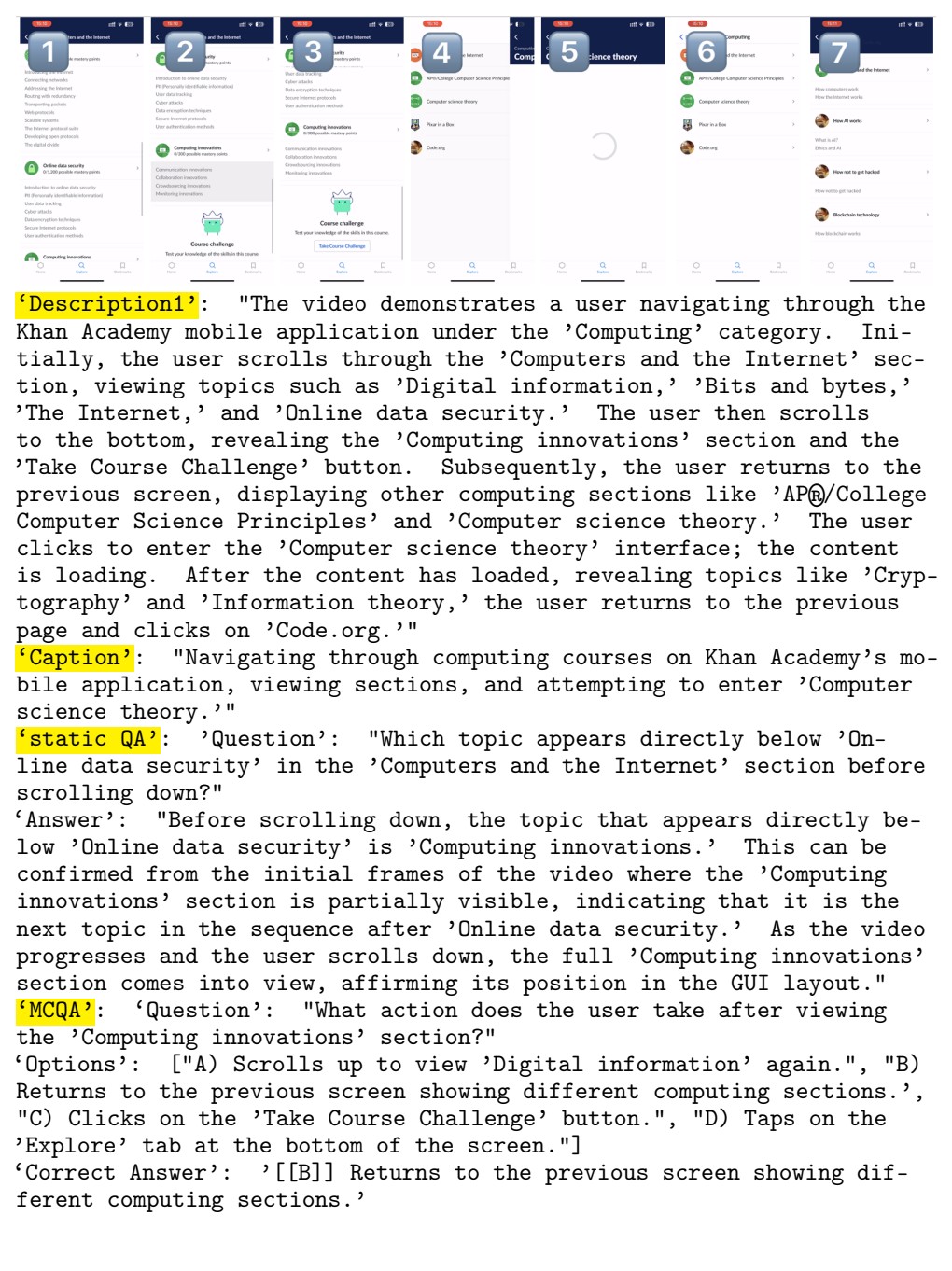

‘Description1’: "The video demonstrates a user navigating through the Khan Academy mobile application under the 'Computing' category. Initially, the user scrolls through the 'Computers and the Internet' section, viewing topics such as 'Digital information,' 'Bits and bytes,' 'The Internet,' and 'Online data security.' The user then scrolls to the bottom, revealing the 'Computing innovations' section and the 'Take Course Challenge' button. Subsequently, the user returns to the previous screen, displaying other computing sections like 'AP®/College Computer Science Principles' and 'Computer science theory.' The user clicks to enter the 'Computer science theory' interface; the content is loading. After the content has loaded, revealing topics like 'Cryptography' and 'Information theory,' the user returns to the previous page and clicks on 'Code.org.'"

‘Caption’: "Navigating through computing courses on Khan Academy's mobile application, viewing sections, and attempting to enter 'Computer science theory.'"

‘static QA’: ‘Question’: "Which topic appears directly below 'Online data security' in the 'Computers and the Internet' section before scrolling down?"
‘Answer’: "Before scrolling down, the topic that appears directly below 'Online data security' is 'Computing innovations.' This can be confirmed from the initial frames of the video where the 'Computing innovations' section is partially visible, indicating that it is the next topic in the sequence after 'Online data security.' As the video progresses and the user scrolls down, the full 'Computing innovations' section comes into view, affirming its position in the GUI layout."

‘MCQA’: ‘Question’: "What action does the user take after viewing the 'Computing innovations' section?"
‘Options’: ["A) Scrolls up to view 'Digital information' again.", "B) Returns to the previous screen showing different computing sections.', "C) Clicks on the 'Take Course Challenge' button.", "D) Taps on the 'Explore' tab at the bottom of the screen."]
‘Correct Answer’: ’[[B]] Returns to the previous screen showing different computing sections.’

Figure 26: Case study for IOS (part 1).

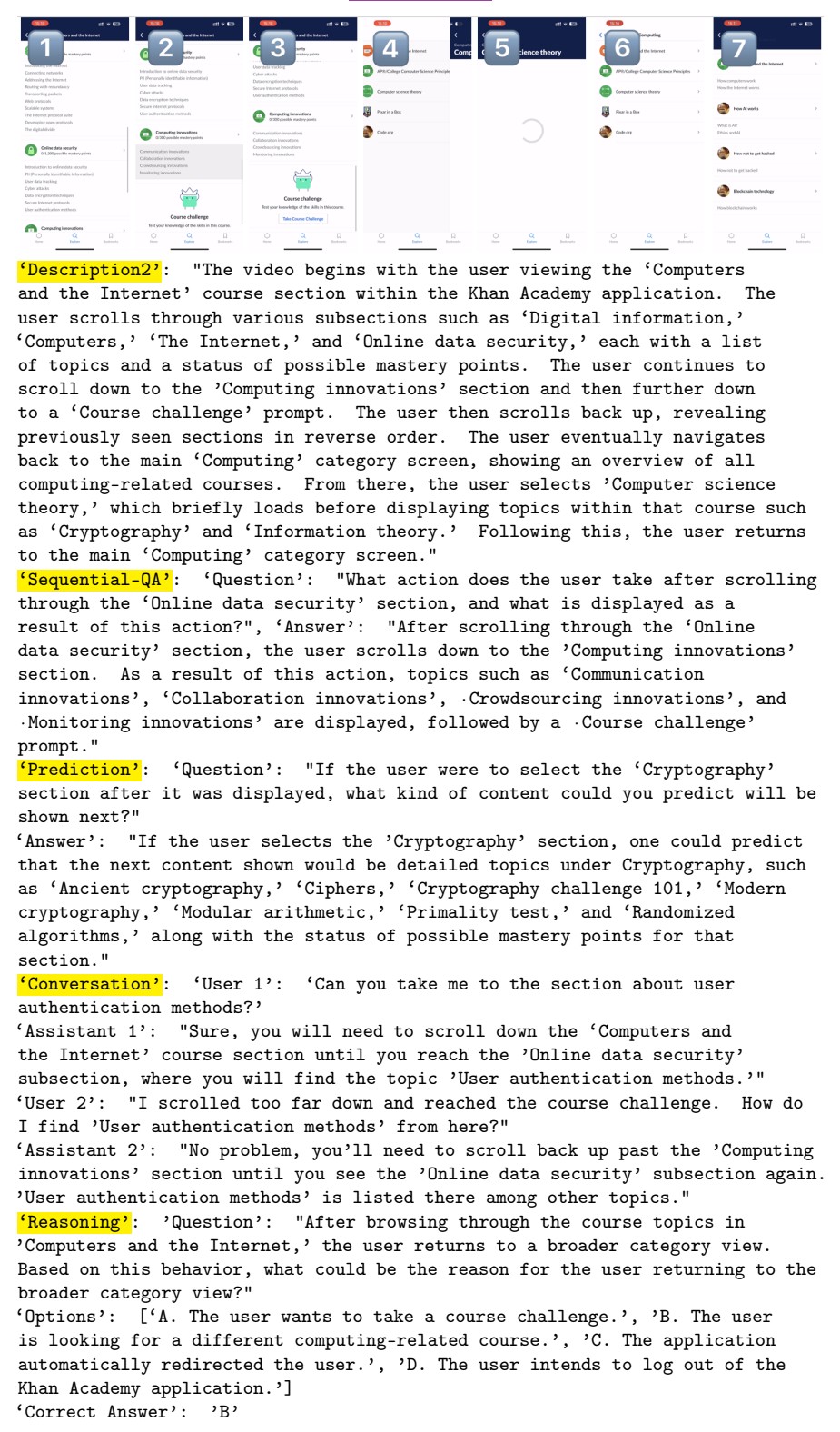

'Description2':  "The video begins with the user viewing the 'Computers and the Internet' course section within the Khan Academy application.  The user scrolls through various subsections such as 'Digital information,' 'Computers,' 'The Internet,' and 'Online data security,' each with a list of topics and a status of possible mastery points.  The user continues to scroll down to the 'Computing innovations' section and then further down to a 'Course challenge' prompt.  The user then scrolls back up, revealing previously seen sections in reverse order.  The user eventually navigates back to the main 'Computing' category screen, showing an overview of all computing-related courses.  From there, the user selects 'Computer science theory,' which briefly loads before displaying topics within that course such as 'Cryptography' and 'Information theory.'  Following this, the user returns to the main 'Computing' category screen."

'Sequential-QA':  'Question':  "What action does the user take after scrolling through the 'Online data security' section, and what is displayed as a result of this action?", 'Answer':  "After scrolling through the 'Online data security' section, the user scrolls down to the 'Computing innovations' section.  As a result of this action, topics such as 'Communication innovations', 'Collaboration innovations', ·Crowdsourcing innovations', and ·Monitoring innovations' are displayed, followed by a ·Course challenge' prompt."

'Prediction':  'Question':  "If the user were to select the 'Cryptography' section after it was displayed, what kind of content could you predict will be shown next?"
'Answer':  "If the user selects the 'Cryptography' section, one could predict that the next content shown would be detailed topics under Cryptography, such as 'Ancient cryptography,' 'Ciphers,' 'Cryptography challenge 101,' 'Modern cryptography,' 'Modular arithmetic,' 'Primality test,' and 'Randomized algorithms,' along with the status of possible mastery points for that section."

'Conversation':  'User 1':  'Can you take me to the section about user authentication methods?'
'Assistant 1':  "Sure, you will need to scroll down the 'Computers and the Internet' course section until you reach the 'Online data security' subsection, where you will find the topic 'User authentication methods.'"
'User 2':  "I scrolled too far down and reached the course challenge.  How do I find 'User authentication methods' from here?"
'Assistant 2':  "No problem, you'll need to scroll back up past the 'Computing innovations' section until you see the 'Online data security' subsection again.  'User authentication methods' is listed there among other topics."

'Reasoning':  'Question':  "After browsing through the course topics in 'Computers and the Internet,' the user returns to a broader category view.  Based on this behavior, what could be the reason for the user returning to the broader category view?"
'Options':  ['A. The user wants to take a course challenge.', 'B. The user is looking for a different computing-related course.', 'C. The application automatically redirected the user.', 'D. The user intends to log out of the Khan Academy application.']
'Correct Answer':  'B'

Figure 27: Case study for IOS (part 2).

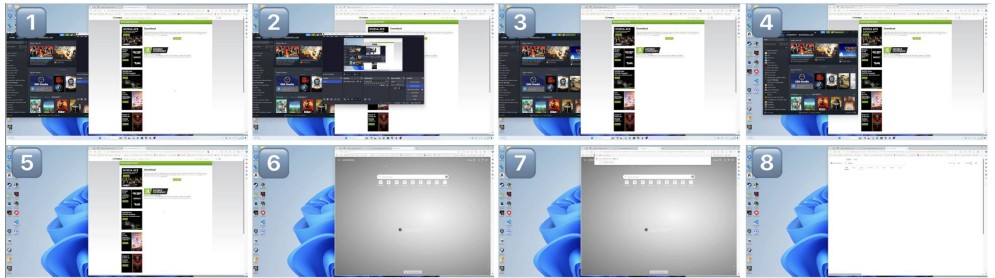

'Description1':  "The video begins with a Windows desktop displaying
multiple open applications, including Steam, OBS Studio, and a web
browser with NVIDIA's website loaded.  The user starts by clicking on
the back page of the browser, which partially obscures the OBS window.
Then, the user clicks on the OBS application, bringing it to the fore-
front.  The user minimizes OBS, followed by dragging the Steam window
to the center of the screen and minimizing it as well.  A new web page
is opened in the Edge browser's navigation bar, and the user types 'of-
fice' into the search bar.  The browser navigates to the Bing search
interface, and 'office' is successfully searched."
'Caption':  'Navigating and Managing Multiple Applications on Windows
Including Steam, OBS Studio, and Edge Browser'
'static QA': 'Question':  "Which web browser is used in the video and
which website is prominently featured before the search for 'office'?"
'Answer':  "The web browser used in the video is Microsoft Edge.
The prominently featured website before the search for 'office' is
NVIDIA's official website where the 'Download Drivers' page is dis-
played."
'MCQA':  'Question':  'What action is taken after the OBS application
is minimized?', 'Options':  ['A. The Steam window is closed.'
'B. The Steam window is moved to the center of the screen and mini-
mized.', 'C. The Edge browser is closed.', 'D. A file is opened from
the desktop.']
'Correct Answer':  '[[B]] The Steam window is moved to the center of
the screen and minimized.'

Figure 28: Case study for multiple-windows interaction (part 1).

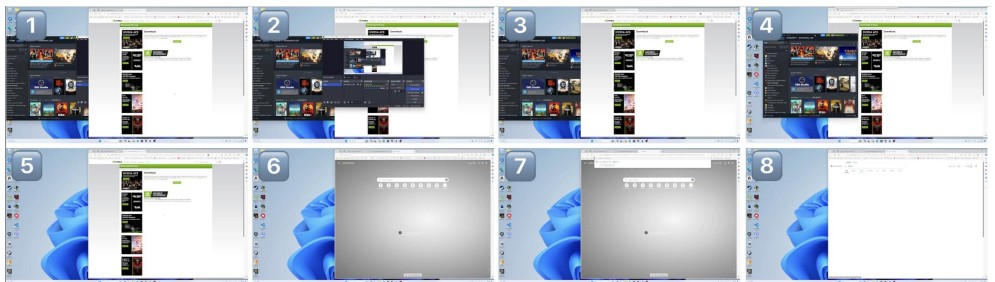

'Description2': "The video shows a Windows desktop with several application windows open, including Steam, OBS, and Edge. Initially, the OBS window is partially covering the Edge browser. The user clicks on the Edge browser, bringing it to the foreground, and then minimizes the OBS window, clearing the view. Next, the user moves the Steam window to the center of the screen and minimizes it as well. Afterward, the user opens a new tab in Edge and enters 'office' into the search bar, which leads to a Bing search results page for 'office'. The video ends with the user on the Bing search page with 'office' results displayed."

'Sequential-QA': 'Question': 'After moving the Steam window to the center, what did the user do next in the Edge browser?'
'Answer': "After moving the Steam window to the center, the user minimized the Steam application. Then, they opened a new tab in the Edge browser and entered 'office' into the search bar, which led them to the Bing search results page for 'office'."

'Prediction': 'Question': "What would be the likely next action the user performs after searching for 'office' on Bing?"
'Answer': "The likely next action the user would perform after searching for 'office' on Bing would be to click on one of the search results to find more information or to visit the Microsoft Office website to purchase or learn more about the office suite."

'Conversation': 'User 1': 'Can you minimize the OBS application for a better view of the browser?'
'Assistant 1': 'Certainly, the OBS application has been minimized, providing a clear view of the Edge browser.'
'User 2': 'Great, now can you search for Microsoft Office in the Edge browser?'
'Assistant 2': "Of course, a new tab has been opened in the Edge browser and 'office' has been entered into the search bar. The Bing search results for 'office' are now displayed."

'Reasoning': 'Question': 'If the user needs to record gameplay footage next, which application should they interact with and what would be their first step?'
'Options': ["A. They should open the Steam application and click on the 'STORE' tab.", "B. They should open the Edge browser and search for 'game recording software'.", "C. They should reopen the OBS application and click on the 'Start Recording' button.", "D. They should access the Windows Start menu and search for the 'Camera' app."]
'Correct Answer': 'C'

Figure 29: Case study for multiple-windows interaction (part 2).

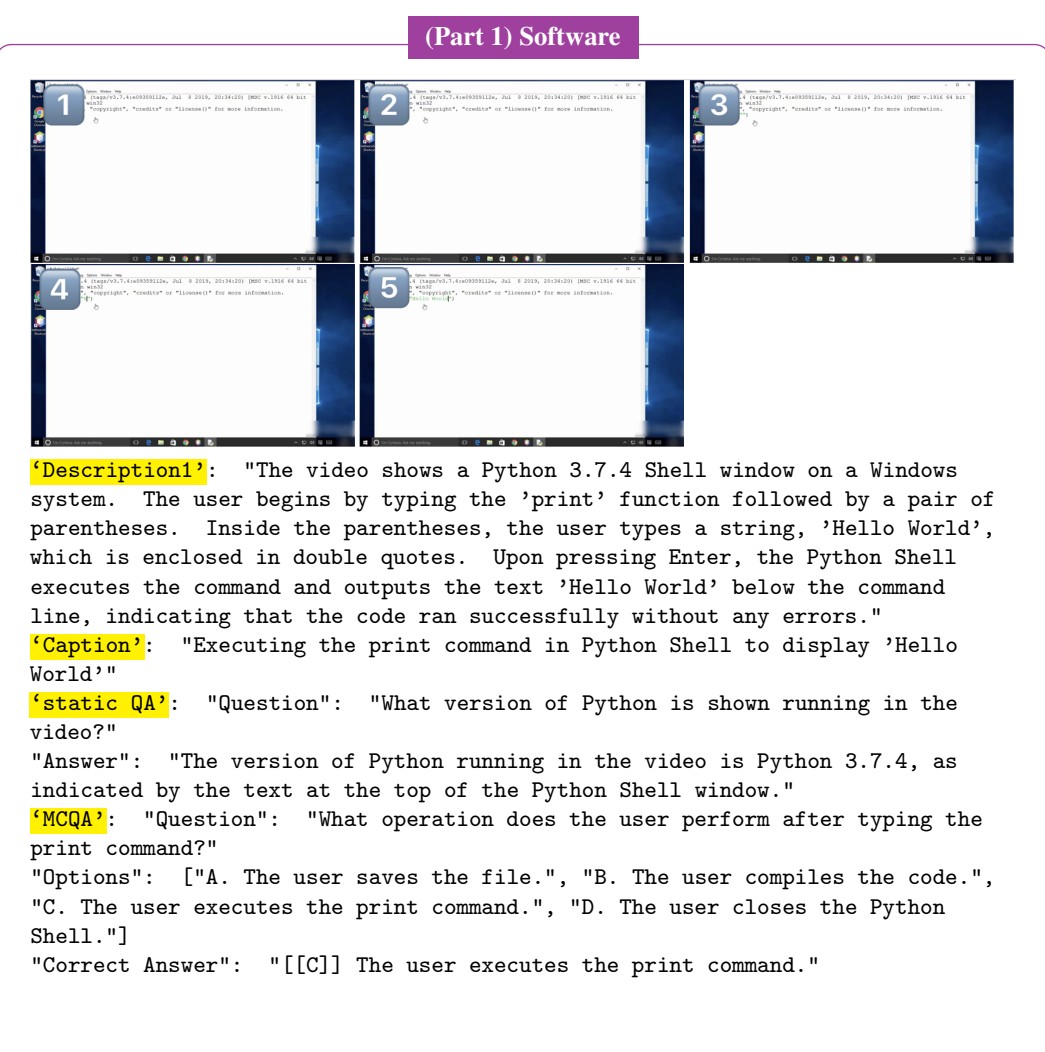

'Description1': "The video shows a Python 3.7.4 Shell window on a Windows system. The user begins by typing the 'print' function followed by a pair of parentheses. Inside the parentheses, the user types a string, 'Hello World', which is enclosed in double quotes. Upon pressing Enter, the Python Shell executes the command and outputs the text 'Hello World' below the command line, indicating that the code ran successfully without any errors."
'Caption': "Executing the print command in Python Shell to display 'Hello World'"
'static QA': "Question": "What version of Python is shown running in the video?"
"Answer": "The version of Python running in the video is Python 3.7.4, as indicated by the text at the top of the Python Shell window."
'MCQA': "Question": "What operation does the user perform after typing the print command?"
"Options": ["A. The user saves the file.", "B. The user compiles the code.", "C. The user executes the print command.", "D. The user closes the Python Shell."]
"Correct Answer": "[[C]] The user executes the print command."

Figure 30: Case study for software (part 1).

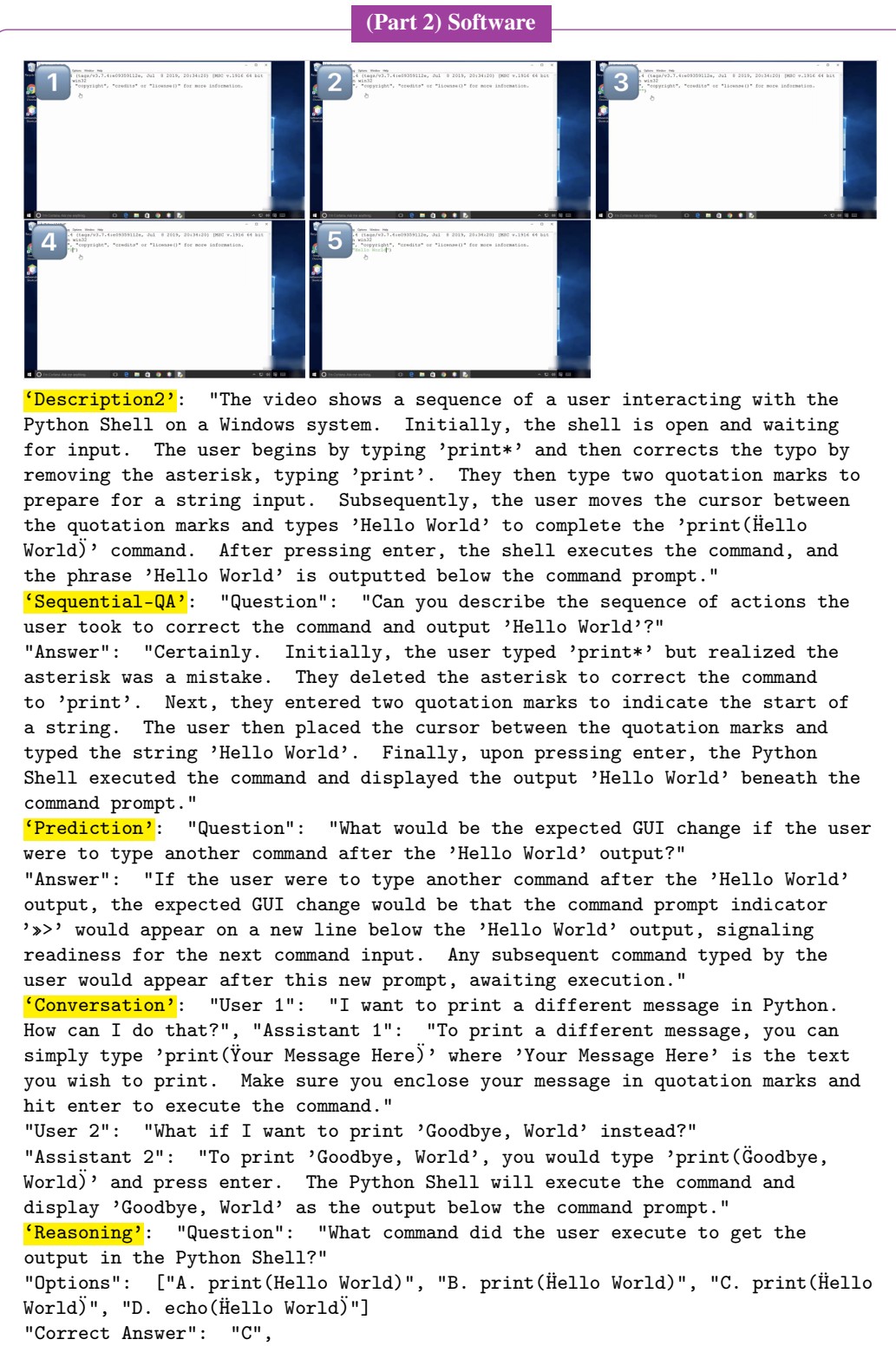

**'Description2'**: "The video shows a sequence of a user interacting with the Python Shell on a Windows system. Initially, the shell is open and waiting for input. The user begins by typing 'print*' and then corrects the typo by removing the asterisk, typing 'print'. They then type two quotation marks to prepare for a string input. Subsequently, the user moves the cursor between the quotation marks and types 'Hello World' to complete the 'print(Ḧello World)' command. After pressing enter, the shell executes the command, and the phrase 'Hello World' is outputted below the command prompt."

**'Sequential-QA'**: "Question": "Can you describe the sequence of actions the user took to correct the command and output 'Hello World'?"
"Answer": "Certainly. Initially, the user typed 'print*' but realized the asterisk was a mistake. They deleted the asterisk to correct the command to 'print'. Next, they entered two quotation marks to indicate the start of a string. The user then placed the cursor between the quotation marks and typed the string 'Hello World'. Finally, upon pressing enter, the Python Shell executed the command and displayed the output 'Hello World' beneath the command prompt."

**'Prediction'**: "Question": "What would be the expected GUI change if the user were to type another command after the 'Hello World' output?"
"Answer": "If the user were to type another command after the 'Hello World' output, the expected GUI change would be that the command prompt indicator '≫' would appear on a new line below the 'Hello World' output, signaling readiness for the next command input. Any subsequent command typed by the user would appear after this new prompt, awaiting execution."

**'Conversation'**: "User 1": "I want to print a different message in Python. How can I do that?", "Assistant 1": "To print a different message, you can simply type 'print(Ÿour Message Here)' where 'Your Message Here' is the text you wish to print. Make sure you enclose your message in quotation marks and hit enter to execute the command."
"User 2": "What if I want to print 'Goodbye, World' instead?"
"Assistant 2": "To print 'Goodbye, World', you would type 'print(G̈oodbye, World)' and press enter. The Python Shell will execute the command and display 'Goodbye, World' as the output below the command prompt."

**'Reasoning'**: "Question": "What command did the user execute to get the output in the Python Shell?"
"Options": ["A. print(Hello World)", "B. print(Ḧello World)", "C. print(Ḧello World)", "D. echo(Ḧello World)"]
"Correct Answer": "C",

Figure 31: Case study for software (part 2).

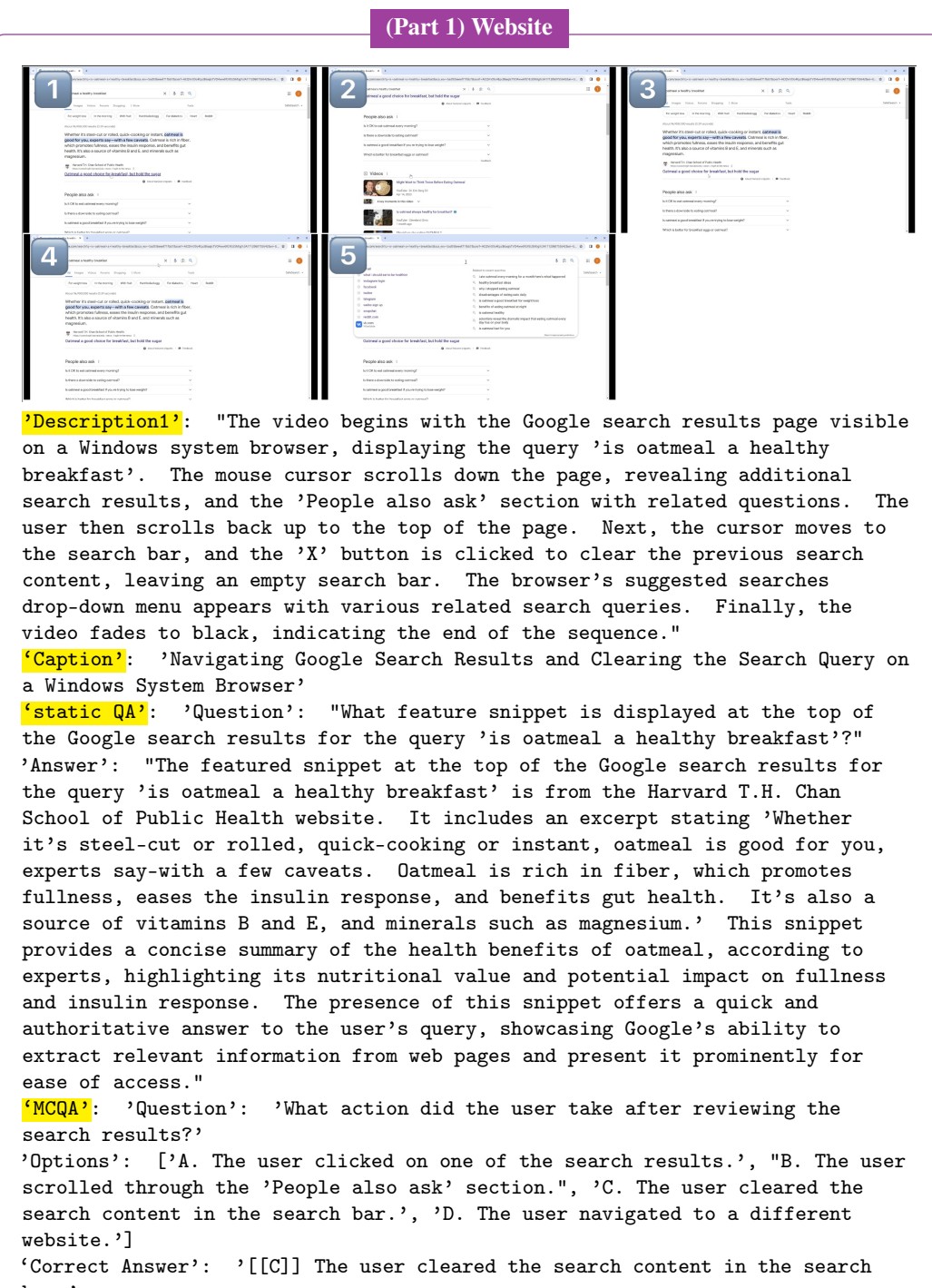

**'Description1'**: "The video begins with the Google search results page visible on a Windows system browser, displaying the query 'is oatmeal a healthy breakfast'. The mouse cursor scrolls down the page, revealing additional search results, and the 'People also ask' section with related questions. The user then scrolls back up to the top of the page. Next, the cursor moves to the search bar, and the 'X' button is clicked to clear the previous search content, leaving an empty search bar. The browser's suggested searches drop-down menu appears with various related search queries. Finally, the video fades to black, indicating the end of the sequence."

**'Caption'**: 'Navigating Google Search Results and Clearing the Search Query on a Windows System Browser'

**'static QA'**: 'Question': "What feature snippet is displayed at the top of the Google search results for the query 'is oatmeal a healthy breakfast'?"
'Answer': "The featured snippet at the top of the Google search results for the query 'is oatmeal a healthy breakfast' is from the Harvard T.H. Chan School of Public Health website. It includes an excerpt stating 'Whether it's steel-cut or rolled, quick-cooking or instant, oatmeal is good for you, experts say-with a few caveats. Oatmeal is rich in fiber, which promotes fullness, eases the insulin response, and benefits gut health. It's also a source of vitamins B and E, and minerals such as magnesium.' This snippet provides a concise summary of the health benefits of oatmeal, according to experts, highlighting its nutritional value and potential impact on fullness and insulin response. The presence of this snippet offers a quick and authoritative answer to the user's query, showcasing Google's ability to extract relevant information from web pages and present it prominently for ease of access."

**'MCQA'**: 'Question': 'What action did the user take after reviewing the search results?'
'Options': ['A. The user clicked on one of the search results.', "B. The user scrolled through the 'People also ask' section.", 'C. The user cleared the search content in the search bar.', 'D. The user navigated to a different website.']
'Correct Answer': '[[C]] The user cleared the search content in the search bar.',

Figure 32: Case study for website (part 1).

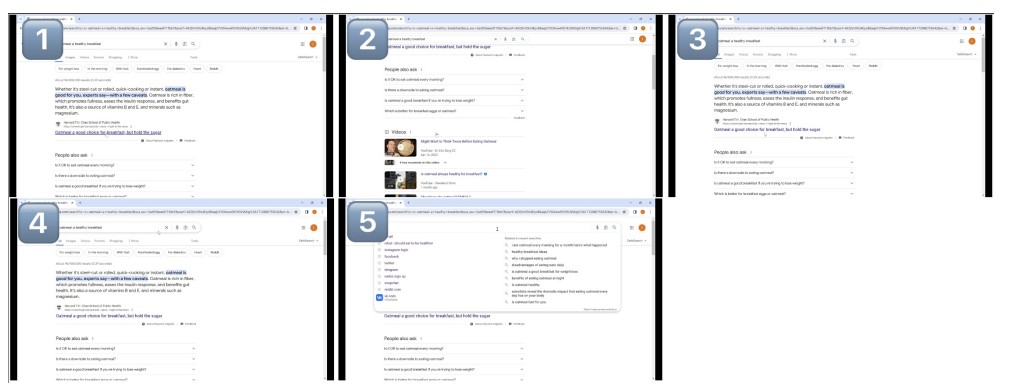

**'Description2'**:  "The video shows a sequence of actions on a Google search results page within a web browser on a Windows system.  Initially, the mouse cursor moves over a search result discussing the health benefits of oatmeal. Next, the user scrolls down, revealing a 'People also ask' section with questions related to oatmeal and a 'Videos' section showcasing related content. Subsequently, the user scrolls back up to the original position, highlighting the same search result about oatmeal's health benefits.  Finally, the user moves the cursor to the search bar and clicks the 'X' to clear the previous search content, resulting in a blank search bar with suggestions and related searches listed below it.  The screen then goes black, indicating the end of the video."

**'Sequential-QA'**:  'Question':  "After reading about oatmeal's health benefits, what section of the page did the user explore next and did they return to read more about the health benefits afterward?"

'Answer':  "After reading about the health benefits of oatmeal, the user scrolled down to explore the 'People also ask' section and the 'Videos' section related to oatmeal.  Following this, the user scrolled back up to revisit the original search result that discussed the health benefits of oatmeal."

**'Prediction'**:  'Question':  'Based on the sequence of events, what action is the user likely to perform after clearing the search bar?'

'Answer':  'After clearing the search bar, the user is likely to type in a new search query or select one of the suggested searches that appear below the search bar.'

**'Conversation'**:  'User 1':  "I want to find more articles on oatmeal's health benefits.  Can you scroll back up?"

'Assistant 1':  "I've scrolled back up and the article on oatmeal's health benefits from the Harvard T.H. Chan School of Public Health is highlighted again."

'User 2':  'Great, now can you clear the search and look for something else?'

'Assistant 2':  'The search content has been cleared, and the search bar is now empty, showing a list of related searches and previous search history suggestions for a new query.'

**'Reasoning'**:  'Question':  'If the user wants to perform a new search after clearing the search bar, which of the following actions would they need to take next?', 'Options':  ['A. Scroll down to view more search results'
'B. Type a new query into the search bar', "C. Click on one of the 'People also ask' questions", 'D. Close the browser window']
'Correct Answer':  'B',

Figure 33: Case study for website (part 2).

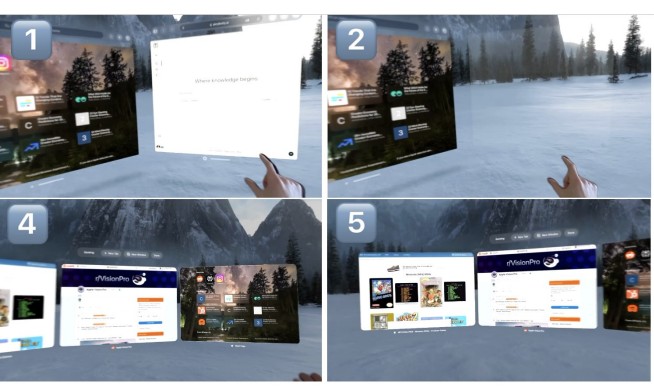

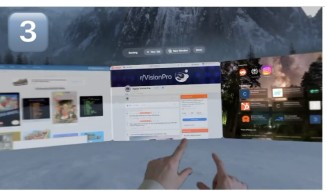

**'Description1':** "The video showcases a user navigating through various pages within the Apple Vision Pro browser on a Windows system. Initially, the browser displays the start page with Favorites and Reading List. The user then turns their head to the right, which triggers the transition to view a webpage on the right side. Following this, the user pinches with both hands to exit the page and then pinches with both hands and fingers moving towards the middle to expand the browser's various pages. This reveals multiple open browser tabs side by side. The user continues to turn their head left and right to view different pages on each side. Lastly, the user selects and expands a specific tab to fill the screen, displaying its content."

**'Caption':** 'Navigating through multiple browser pages using head movement and hand gestures in Apple Vision Pro on Windows'

**'static QA':** 'Question': "What is the main category listed under the Favorites section on the browser's start page?"
'Answer': "The main category listed under the Favorites section on the browser's start page is 'Perplexity', denoted by a unique icon, followed by other favorites like Instagram and various websites."

**'MCQA':** 'Question': 'How does the user switch between different open tabs in the Apple Vision Pro browser?'
'Options': ['A. Using keyboard shortcuts', 'B. Turning their head left and right', 'C. Scrolling with a mouse', 'D. Typing the tab number']
'Correct Answer': '[[B]] Turning their head left and right'

Figure 34: Case study for XR (part 1).

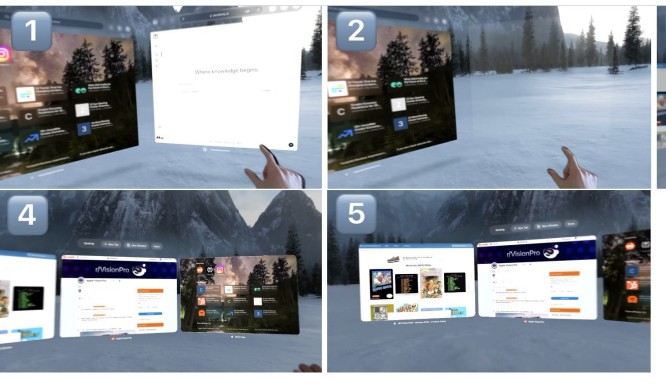

'Description2': "The video starts with a full-screen view of a browser
interface titled 'Apple Vision Pro' displaying various website thumbnails
and bookmarks. The user then turns their head to the right, causing the right
side of the browser to come into view. Next, the user looks at a dot at the
bottom of the page and pinches with both hands, which causes the browser to
exit the full-screen view and shrink to a smaller, windowed mode. The user
then performs a pinching motion with both hands, bringing the fingers towards
the middle, which causes the browser's various pages to expand, giving an
overview of multiple open tabs. The user again turns their head to the right
to view the right side page and then to the left to view the left side page.
Throughout the video, the GUI elements such as tabs, the address bar, and
website thumbnails respond dynamically to the user's head movements and hand
gestures."

'Sequential-QA': 'Question': 'What action did the user take immediately
after turning their head to the right to view the right side page?'
'Answer': 'After turning their head to the right to view the right side
page, the user looked at a dot at the bottom of the page and pinched with
both hands, which caused the browser to exit full-screen mode and shrink to a
smaller windowed interface.'

'Prediction': 'Question': 'If the user pinches with both hands and moves
their fingers towards the middle after the browser shrinks to a smaller
window, what will happen next?'
'Answer': "If the user pinches with both hands and moves their fingers
towards the middle after the browser shrinks, the action will cause the
browser's various pages to expand, providing an overview of multiple open
tabs."

'Conversation': 'User 1': 'How do I view all the pages I have open in the
browser?'
'Assistant 1': 'You can view all the open pages by pinching with both hands
and moving your fingers towards the middle. This will expand the browser and
show an overview of all the open tabs.'
'User 2': 'Great, and how can I exit this overview?'
'Assistant 2': 'To exit the overview of open pages, you can look at the dot
at the bottom of the page and pinch with both hands. This will exit the
overview and return you to the individual page view.'

'Reasoning': 'Question': 'How can the user access the options to open a new
tab or window from the current state?'
'Options': ['A. Turn their head to the left and select the plus sign.', 'B.
Swipe left on the touchpad.', 'C. Turn their head to the right and select the
'Done' button.', 'D. Pinch with both hands to exit the current view and access
the toolbar.']
'Correct Answer': 'D'

Figure 35: Case study for XR (part 2).

