# OpenReview forum: "GUI-World: A Dataset for GUI-Orientated Multimodal Large Language Models"
_NeurIPS.cc/2024/Datasets_and_Benchmarks_Track — Submitted to NeurIPS 2024 Track Datasets and Benchmarks_

### Official Review · Reviewer_XMtM · 2024-07-15
**Review for "GUI-World"**

**Rating:** 7
**Confidence:** 3

**Review:**

This paper focuses on the GUI understanding capabilities of MLLMs. It proposes a benchmarking dataset and a model to simultaneously measure the performance of these models and provide insights on how to improve the performance of open-source models on such tasks. The three contributions are as follows:

- A dataset containing 12,379 videos to assess the GUI understanding capabilities of MLLMs
- A model (GUI-Viv) trained on the purpose-built dataset to show possible improvement on the desired task.
- Benchmarking existing MLLMs on the proposed dataset to show their promise and limitations


This paper compares different models, ranging from open source to closed source. Table 4 shows the accuracy and scores of various models on the task, with GPT-4 achieving the best results in several categories. While interesting, the paper does not provide a clear way to interpret these numbers. Is GUI understanding a solved task, and should the community focus on more challenging problems? Do we face significant challenges in closing the gap to 100% accuracy?

Additionally, some of the takeaways and results are not surprising and are easy to predict. For example, commercial models perform better than open-source models, and performance varies in different GUI scenarios.


Question: What data was used to train GUI-Vid in section 3? How much data is shared between training and the benchmark?

**Strengths:**

- The dataset contains videos rather than still images, showing the authors’ attempt to make the dataset future-proof as more agents become capable of working with videos.
- Size of the dataset: Although 12,379 might seem like a small number compared to many datasets, the effort required to collect such data is significant.
- The data collected in this dataset is not tied to any specific model/application and is related to the general daily use of various applications on multiple platforms (from iOS and VisionOS to the web), which is another positive point.

**Additional Feedback:**

Please see my comments above.

**Clarity:**

Yes with a minor issues.

- Regarding page 10: Adding this page was totally unnecessary and contains minor information. Please pay attention to the page limit.
- Suggestion: When referring to the size of the dataset, be exact. For example, in line 64, don’t say “over 12,000”; use the exact number: 12,379

**Correctness:**

Yes, the dataset is sourced from online videos, and annotations and task descriptions are added by humans and GPT-4V.

**Documentation:**

Yes, but for the trained model, there is no documentation (or I should say, I could not find any).

**Ethics:**

No, the paper is good.

**Limitations:**

- The main motivation of the work is related to achieving agents that can accomplish tasks in various applications. However, the paper lacks discussion and connection between the designed benchmark and the actual ability of models to act as agents. (How does this benchmark help or rank models in the actual task of being a generalist agent?, In other words, if model A is better than model B in this benchmark, is model A actually better at acting as an agent or not?)
- I believe the multiple-choice format is not the correct method for assessing knowledge in the current task discussed in the benchmark, as it might be very easy for large models to guess the answer without paying attention to details.

**Opportunities For Improvement:**

- One of the perils in the current multimodal benchmarks is the nonnecessity of images/videos to answer questions. [1]. Did the authors conduct a study to see the performance of models on questions without images/videos?


[1] - Chen, Lin, et al. "Are We on the Right Way for Evaluating Large Vision-Language Models?." arXiv preprint arXiv:2403.20330 (2024).

**Relation To Prior Work:**

Yes, I like Table 1 of this paper, which tries to compare the dataset in this paper with prior work in a very elegant way.

**Summary And Contributions:**

This paper introduces a benchmarking dataset for assessing GUI understanding in multimodal large language models (MLLMs). The benchmark contains various tasks related to understanding interactions inside applications, ranging from mobile to desktop and mixed reality. The authors have evaluated various models, both closed and open source, on the proposed benchmark. Additionally, the authors have trained a model to demonstrate that it is possible to improve performance on GUI understanding tasks.

---

> ### Author Rebuttal · Authors · 2024-08-15
>
> Thank you for your support and appreciation of our work! We are also grateful for the time and effort you have invested in reviewing our paper. We will address each of your concerns step by step:
>
> ---
>
> **Q1**: What data was used to train GUI-Vid in section 3? How much data is shared between training and the benchmark?
>
> **A1:** Thanks for pointing this out. We have placed the statistics for the training dataset in Appendix D, specifically in Tables 9 and 10. Our training set and test set are completely separate, with no data leakage between them, to ensure the fairness and integrity of the benchmark.
>
> ---
>
> **Q2:** Did the authors conduct a study to see the performance of models on questions without images/videos?
>
> **A2:** Your question is very constructive. Not using image/video for multimodal benchmarks can reveal whether the model is using the vision side to solve problems. We have added relevant experiments for testing, comparing GPT-4 with and without vision input. Due to word limit constraints, we reported an average of 6 scenarios, as shown in **Table 1 2**, with detailed results in **PDF**. We found that for conversation and MCQA, the model can solve certain problems through questions and context. However, for description and questions requiring reasoning, such as dynamic or static scenarios, there is still a significant gap.
>
> ---
> **Table 1: Average Performance of GPT-4o with and without Vision Input.**
> | Setting | MCQA_all | Description | Conversation | Dynamic | Static | Caption |
> |---------|----------|-------------|--------------|---------|--------|---------|
> | w.o. vision | 0.790 | 1.872 | 3.915 | 2.979 | 2.486 | 2.187 |
> | w. vision | **0.848**| **3.031**| **4.056** | **3.318** | **3.131** | **3.911** |
>
> **Table 2: Performance of GPT-4o in Software Scenarios with and without Vision Input.**
> | Setting | MCQA_all | Description | Conversation | Dynamic | Static | Caption |
> |-----------|----------|-------------|--------------|---------|--------|---------|
> | w.o. vision | 0.831    | 1.878       | 4.123        | 3.034   | 2.633  | 2.042   |
> | w. vision   | **0.865**    | **3.028**       | **4.223**        | **3.341**   | **3.125**  | **4.048**   |
>
> ---
> **Q3:** The main motivation of the work is related to achieving agents that can accomplish tasks in various applications. However, the paper lacks discussion and connection between the designed benchmark and the actual ability of models to act as agents. (How does this benchmark help or rank models in the actual task of being a generalist agent? In other words, if model A is better than model B in this benchmark, is model A actually better at acting as an agent or not?)
>
> **A3:**- Regarding the definition of GUI agents, there are those that write code to operate GUI **[1]**, and those that assist people through chat **[2]**. We conducted experiments from these two perspectives, verifying the strong correlation between GUI understanding capability and GUI agents.
>
> - We compared the benchmark results on GUI-world with existing benchmarks **[3] [4] [5]** for operating on GUI as shown in **Table 3**, and found that the results generally match, *i.e.*, the stronger the understanding ability, the stronger the agent performance.
> - For the definition of chat helping humans, we selected 180 videos from the benchmark, choosing 30 videos for each scenario. We asked 5 human annotators to pose the question they most wanted to ask after watching each video. We then used GUI-Vid, both before and after fine-tuning, to answer these questions. The human annotators who asked the questions were then asked to indicate which answer was more helpful. The results are shown in **Table 4**, demonstrating that models trained in GUI understanding are more favored by people when acting as GUI agents.
>
> Both task settings fully demonstrate that models perform well on our benchmark also excel when acting as GUI agents, proving the effectiveness of our benchmark and dataset.
>
> ---
>
> **Table 3: Strong Correlation Between Our Benchmark (GUI Understanding) and Other GUI Agent Benchmarks  (Number means Ranking)**
> | Model         | GUI-World | VisualAgentBench | VideoGUI | OS-World |
> |---------------|-----------|-------------------|----------|----------|
> | GPT-4o        | **1**         | **1**                 | **1**        | 2        |
> | GPT-4V        | **2**        | **2**                 | **2**        | 1        |
> | Gemini-1.5-Pro| **3**        | **3**                 | **3**        | **3**        |
> | Qwen-VL-Max   | **4**         | **4**                 | **4**        | /        |
>
> **Table 4: User Preference: GUI-Vid vs VideoChat2 (With and Without Fine-tuning on GUI-World)**
> | Scenarios | GUI-Vid     | Tie    | VideoChat2   |
> |-----------|---------|--------|--------|
> | Software  | **82.7%**   | 13.3%  | 4.0%   |
> | Website   | **86.0%**   | 12.0%  | 2.0%   |
> | XR        | **88.0%**   | 8.7%   | 3.3%   |
> | Multi     | **85.3%**   | 10.0%  | 8.7%   |
> | IOS       | **92.0%**   | 6.0%   | 2.0%   |
> | Android   | **82.0%**   | 16.0%  | 2.0%   |
> | Average   | **86.0%**   | 11.0%  | 3.7%   |
>
> ---
>
> **[1]** SeeClick: Harnessing GUI Grounding for Advanced Visual GUI Agents
>
> **[2]** CogAgent: A Visual Language Model for GUI Agents
>
> **[3]** OSWorld: Benchmarking Multimodal Agents for Open-Ended Tasks in Real Computer Environments
>
> **[4]** VideoGUI: A Benchmark for GUI Automation from Instructional Videos
>
> **[5]** VisualAgentBench: Towards Large Multimodal Models as Visual Foundation Agents

---

> > ### Author Rebuttal · Authors · 2024-08-15
> >
> > **Q4:** I believe the multiple-choice format is not the correct method for assessing knowledge in the current task discussed in the benchmark, as it might be very easy for large models to guess the answer without paying attention to details.
> >
> > **A4:** Thanks for your valuable comment. Indeed, MCQA is only a small part of our benchmark, as shown in Table 2 in our manuscript, while most of the questions are free-form or conversational.
> >
> > Based on the experiments where only language is used to answer questions (your Q2), MCQA is indeed not an ideal method for benchmarking. However, it can enrich the question formats in the training dataset. Therefore, we will remove MCQA from the benchmark in the next version. Thank you for pointing this out!
> >
> > ---
> >
> > **Q5:** Regarding page 10: Adding this page was totally unnecessary and contains minor information. Please pay attention to the page limit.
> >
> > **A5:** This page is a requirement for the NeurIPS DB track. We will inquire with the Area Chair about the possibility of moving the content of this page to the appendix to enhance the paper's aesthetics and readability.
> >
> > ---
> >
> > **Q6:** When referring to the size of the dataset, be exact. For example, in line 64, don’t say “over 12,000”; use the exact number: 12,379
> >
> > **A6:** Thank you for your suggestion. In the next version, we will replace all of these with specific numbers.
> >
> > ---
> >
> > **Q7:** Problem about documentation.
> >
> > **A7:** Thank you for your suggestion. By the supplementary material deadline, we had released scripts on how to use the model, but we indeed hadn't yet provided documentation for training. However, we have now released the scripts used for training as well.
> >
> > ---
> > We greatly appreciate your insights and will continue to contribute to the GUI field based on your suggestions. We plan to use our proposed dataset to train more powerful GUI expert models, aiming to advance the open-source community's development in the GUI domain.

---

> > > ### Comment · Reviewer_XMtM · 2024-08-25
> > >
> > > I thank the author for their detailed response and for clarifying some aspects of the paper. Before I update my rating, I have one final question: Given the new results for the benchmark with and without the vision input, is the gap between the two setups meaningful? Is this benchmark a good measure for assessing the GUI understanding of models? In other words, what should our expectation be for the ‘gap’ between these two setups ? What does ‘significant’ mean in this context?

---

> > > > ### Author Response · Authors · 2024-08-25
> > > > **Official Comment by Authors**
> > > >
> > > > Thank you for your question. We will first explain our expectations for the 'gap' between these two setups, and then explain why the additional experiment results are meaningful and why we consider our benchmark a good measure of assessing the GUI understanding of models.
> > > >
> > > > ---
> > > > **Q1:** What should our expectation be for the ‘gap’ between these two setups? What does "significant" mean in this context?
> > > >
> > > > **A1:** I believe our expectation for a vision-language benchmark should be that the "with vision" setting performs better than the "without vision" setting. Otherwise, it could not be considered a true vision-language benchmark. In our review, "significant" means that there is a huge performance difference between "with" and "without" vision input, particularly in the dynamic and static tasks.
> > > >
> > > > ---
> > > > **Q2:** Is the gap between the two setups meaningful? Is this benchmark a good measure for assessing the GUI understanding of models?
> > > >
> > > > **A2:** This gap between the "with" and "without" vision input demonstrates that our benchmark require visual perception capabilities (unlike some vision-language benchmarks, where stronger backbone LLMs always lead to better performance **[1]**), effectively proves the validity of our benchmark, as it can verify whether models have the ability to perceive GUI visual elements. It also highlights the novelty of our benchmark, showing that there is no benchmark leakage problem, as models relying solely on LLM pre-training knowledge cannot successfully complete tasks in our benchmark. Therefore, we consider our benchmark as a good measure for assessing GUI understanding.
> > > >
> > > > ---
> > > > **[1]** Cambrian-1: A Fully Open, Vision-Centric Exploration of Multimodal LLMs

---

> > > > > ### Comment · Reviewer_XMtM · 2024-08-26
> > > > >
> > > > > I revised my original score as I believe this paper makes a valuable contribution. However, I still have some questions for the authors regarding the interpretation of the results, as their previous answer was not clear to me.
> > > > >
> > > > > In a well-established benchmark like ImageNet, we know that even a 1 percentage point improvement is significant when comparing accuracies over 80%; ImageNet is familiar to many people. However, for GUI understanding benchmark, it is not clear whether a difference between 79% and 84% (with vision vs. without vision) is significant or even meaningful. Is this gap substantial? When we move to other benchmark settings like "Description," interpreting the gap becomes even more challenging. I would like to know the authors' perspective on this.

---

> > ### Author Response · Authors · 2024-08-26
> > **Official Comment by Authors**
> >
> > Thank you for raising the score!
> >
> > Regarding the 5% difference in multiple-choice questions, we consider this gap substantial. With two multiple-choice questions per video and 1,823 videos in total, this 5% difference translates to approximately 182 questions that cannot be answered correctly, which is definitely a significant gap, as it is highly related to whether this model can solve users' GUI problems in daily life.
> >
> > Your question about interpreting the gap in open-ended tasks like "Description" is intriguing and highlights an important issue for the entire LLM community to consider: the role and interpretation of benchmark results. In our case, we use LLM-as-a-Judge to score free-form QA, which to some extent can reflect the performance gap between models, as demonstrated by the wide use of MT-bench **[1]**. However, the use of LLM-as-a-Judge does present challenges in interpreting score differences and quantifying exact performance improvements.
> >
> > In a word, my view is that even subtle score improvements can indicate enhanced model performance on our benchmark, at least demonstrating progress in specific GUI-related tasks.
> >
> > ---
> >
> > Your question has made a deep reflection on my part. I believe the interpretability of benchmark results is an area that requires further investigation by the benchmark research community, as explored in studies like **[2] [3] [4] [5]**. I hope this response addresses your concerns. If you have any further questions, feel free to ask. Our exchange has been extremely enlightening for me, and thank you again for your insightful review.
> >
> > ---
> > **[1]** Judging llm-as-a-judge with mt-bench and chatbot arena
> >
> > **[2]** Are Emergent Abilities of Large Language Models a Mirage?
> >
> > **[3]** Benchmarks as Microscopes: A Call for Model Metrology
> >
> > **[4]** ECBD: Evidence-Centered Benchmark Design for NLP
> >
> > **[5]** https://physics.allen-zhu.com/

---

### Official Review · Reviewer_6RQG · 2024-07-15
**GUI-World**

**Rating:** 6
**Confidence:** 3

**Review:**

The data is interesting but it is unclear how relevant the benchmark is for the intended purpose of the data. The paper presentation needs to improve.

**Strengths:**

A lot of manual effort was put into creating the benchmark.

**Additional Feedback:**

-

**Clarity:**

* The writing is quite rough in places. Please do a thorough proofread + check all figures.
These are some examples but there are many more issues throughout the paper:
- 'As illustrated in Figure 2, the development of GUI-World is structured around a two-stage process.'   Figure 2 contains no clear indication of stages - so what are the two stages?
- A lot of the methodology details are described in the appendix. Would be nice if the methodology section was a bit more self-contained.
- 147 'learning preliminary for GUI content' -> ?
- 247 after finetuned -> after fine-tuning
- 256 fintuning -> fine-tuning
- Figure 4 misses a legend - what do the icons mean?
- Table 4 what is MACQ?
- 4.2 'Empirically results' -> empirical results
- Fig 5 misses Android (at least in the legend). Also - how are the elements ordered? Software/website and the other 3 seem to be intermingled.

**Correctness:**

The dataset construction seems to be sound, but I don't think the evaluation method in the benchmark is appropriate for the goal of the benchmark (see above).

**Documentation:**

I may have overlooked it but I wasn't able to find a link to the data in the paper. The authors point to Appendix D in their checklist, but there is only some documentation. The supplementary data includes only a very small portion of the data.

**Limitations:**

Yes, the limitations and potential negative societal impact are discussed on page 10 of the paper.

**Opportunities For Improvement:**

* Do we need this mixed benchmark?
The benchmark mixes Mobile, Website, Desktop and XR GUIs. Why do we need a single model to handle all those? Would it not make more sense to have a good model for each of those types of GUIs (since they tend to be quite different)? In fact - GPT-4o already seems to do quite well on the benchmark.. so what is the point of publishing the benchmark? These questions are not answered in the paper.

* Unclear how tasks relate to GUI development
While I acknowledge that GUI understanding is important for performing GUI-related tasks with an LLM, it is not clear how some of the tasks are a good proxy for how well an LLM can perform such tasks. For example, is captioning really important for a GUI-related task? It would be good to extend Table 3 with an example of a related GUI task that would be impacted by how well the LLM handles the question type.


Note:
The paper exceeds the page limit and should have been desk rejected.

MINOR questions/comments:
- How is a keyframe defined?
- The authors state that automated keyframe extraction mechanisms do not work well with GUI videos - did they evaluate this systematically, or is it just a gut feeling? How do these approach perform vs. the manual approach that was taken?
- Findings like 'commercial imageLLMs outperform open-source videoLLMs' do not really have a place in dataset papers. The findings should be about the data, and not about which model happens to be better on the data.

**Relation To Prior Work:**

Yes

**Summary And Contributions:**

The authors present GUI-World, a dataset containing over 12k GUI videos, designed to evaluate and enhance the capabilities of GUI agents. They also present a benchmark for GUI understanding and evaluate 7 MLLMs on it, and they fine-tune a model GUI-Vid which is a GUI-oriented VideoLLM. The GUI videos come from student workers and from instruction/tutorial videos, which are manually annotated with textual descriptions.

---

> ### Author Rebuttal · Authors · 2024-08-15
>
> Thank you very much for your feedback. We will explain step by step to clarify your doubts and thus, as much as possible, eliminate any misunderstanding about your contribution to the article.
>
> ---
> **W1.1:** Do we need this mixed benchmark? The benchmark mixes Mobile, Website, Desktop and XR GUIs. Why do we need a single model to handle all those? Would it not make more sense to have a good model for each of those types of GUIs (since they tend to be quite different)?
>
> **A1.1:** We designed the mixed benchmark with two objectives:
>
> - We aim for a general model capable of handling all user queries across various GUI scenarios, such as GPT-4o. While some models excel in certain downstream tasks, they cannot serve as general-purpose GUI assistants, helping users through chatting.
> - GUI elements across different scenarios are relatively similar and possess a degree of generalizability. Scenarios often overlap; for instance, desktop environments may include mobile GUI screenshots or videos. XR scenarios (like in Apple Vision Pro) can encompass both desktop and mobile interfaces. As current models still struggle with these diverse scenarios, we believe an effective GUI assistant should be able to handle all GUI contexts we've proposed in our benchmark.
>
> ---
>
> **W1.2:** In fact - GPT-4o already seems to do quite well on the benchmark.. so what is the point of publishing the benchmark? These questions are not answered in the paper.
>
> **A1.2:** To address your concern, we would like to reaffirm our primary contribution: a high-quality GUI video dataset designed for training open-source models and a benchmark pioneering in its focus on GUI, especially dynamic content, tackling a critical gap in the current understanding of GUI assistants' capabilities to interpret dynamic elements.
>
> Regarding our benchmark results, while GPT-4o demonstrates satisfactory performance, it still falls short of the ideal score of 5, indicating room for improvement in creating a truly powerful GUI assistant. Additionally, the poor performance of open-source models on this benchmark underscores the need for advancement. By publishing this benchmark, we aim to stimulate progress within the open-source community on GUI-oriented tasks. We appreciate your observations and will clarify our motivations more explicitly in the next version of our paper.
>
> ---
>
> **W2:** Unclear how tasks relate to GUI development While I acknowledge that GUI understanding is important for performing GUI-related tasks with an LLM, it is not clear how some of the tasks are a good proxy for how well an LLM can perform such tasks. For example, is captioning really important for a GUI-related task? It would be good to extend Table 3 with an example of a related GUI task that would be impacted by how well the LLM handles the question type.
>
> **A2:** We conducted additional analysis and experiments to show how GUI understanding capability helps mainstream GUI-related tasks, including generating code to operate GUI **[1]** and assist people through chat **[2]**. Both demonstrate the strong correlation between GUI understanding capability and specific tasks for GUI agents.
>
> - We compared the benchmark results on GUI-world with existing benchmarks **[3] [4] [5]** for operating on GUI as shown in **Table 1**, and found that the results generally match, i.e., the stronger the understanding ability, the stronger the agent performance.
> - For the definition of chat helping humans, we selected 180 videos from the benchmark, choosing 30 videos for each scenario. We asked 5 human annotators to pose the question they most wanted to ask after watching each video. We then used GUI-Vid, both before and after fine-tuning, to answer these questions. The human annotators who asked the questions were then asked to indicate which answer was more helpful. The results are shown in **Table 2**, demonstrating that models trained in GUI understanding are more favored by people when acting as GUI agents.
>
> As for caption tasks, we consider them as a basic task in GUI understanding. We will follow your suggestion to extend Table 3 in our manuscript with examples of related GUI tasks. Thanks for your suggestion!
>
> ---
> **Table 1: Strong Correlation Between Our Benchmark (GUI Understanding) and Other GUI Agent Benchmarks (Number means Ranking)**
> | Model         | GUI-World | VisualAgentBench | VideoGUI | OS-World |
> |---------------|-----------|-------------------|----------|----------|
> | GPT-4o        | **1**         | **1**                 | **1**        | 2        |
> | GPT-4V        | **2**        | **2**                 | **2**        | 1        |
> | Gemini-1.5-Pro| **3**        | **3**                 | **3**        | **3**        |
> | Qwen-VL-Max   | **4**         | **4**                 | **4**        | /        |
>
> **Table 2: User Preference: GUI-Vid vs VideoChat2 (With and Without Fine-tuning on GUI-World)**
> | Scenarios | GUI-Vid     | Tie    | VideoChat2   |
> |-----------|---------|--------|--------|
> | Software  | **82.7%**   | 13.3%  | 4.0%   |
> | Website   | **86.0%**   | 12.0%  | 2.0%   |
> | XR        | **88.0%**   | 8.7%   | 3.3%   |
> | Multi     | **85.3%**   | 10.0%  | 8.7%   |
> | IOS       | **92.0%**   | 6.0%   | 2.0%   |
> | Android   | **82.0%**   | 16.0%  | 2.0%   |
> | Average   | **86.0%**   | 11.0%  | 3.7%   |
>
> ---
>
> **[1]** SeeClick: Harnessing GUI Grounding for Advanced Visual GUI Agents
>
> **[2]** CogAgent: A Visual Language Model for GUI Agents
>
> **[3]** OSWorld: Benchmarking Multimodal Agents for Open-Ended Tasks in Real Computer Environments
>
> **[4]** VideoGUI: A Benchmark for GUI Automation from Instructional Videos
>
> **[5]** VisualAgentBench: Towards Large Multimodal Models as Visual Foundation Agents

---

> > ### Author Rebuttal · Authors · 2024-08-15
> >
> > **Q3:** The paper exceeds the page limit.
> >
> > **A3:** Our main content (up to the conclusion section) is within 9 pages. The 10th page contains the "Limitations" and  "Potential Negative Societal Impacts" sections, which are required for submissions to the DB track. Thank you for your inquiry and concern.
> >
> > ---
> > **Q4.1:** How is a keyframe defined?
> >
> > **A4.1:** A keyframe is essentially a snapshot from a video. Our selection of keyframes includes human-selected, random, and programmatically selected frames. During the annotation process, annotators mark the changes occurring in the graphical user interface (GUI) at that specific frame or the operator's intended actions. Additionally, they record mouse and keyboard input signals to provide a comprehensive set of information.
> >
> > ---
> > **Q4.2:** The authors state that automated keyframe extraction mechanisms do not work well with GUI videos - did they evaluate this systematically, or is it just a gut feeling? How do these approach perform vs. the manual approach that was taken?
> >
> > **A4.2:** The conclusion that programmatic methods 'do not work well' is demonstrated through the benchmark results. In our paper, we analyze the issues with programmatic approaches, which typically select frames with the most significant visual changes. However, in GUI interactions, small changes often carry rich semantic information. For instance, when text is entered into a search box, programmatic methods may not select frames before and after the text input as key frames because the visual difference is minimal. Instead, they might choose the frame after the search button is pressed. This leads to a situation where the MLLM doesn't know what the user input was, and consequently, cannot answer questions effectively.
> >
> > ---
> >
> > **Q4.3:** Findings like 'commercial imageLLMs outperform open-source videoLLMs' do not really have a place in dataset papers. The findings should be about the data, and not about which model happens to be better on the data.
> >
> > **A4.3:** Although our paper primarily focuses on a dataset, it also includes a benchmark section. We believe that reporting the performance of these models and detailing their strengths and weaknesses across various scenarios is beneficial to the community. Thank you for your observation. In the next version of our paper, we will dedicate more space to provide a more detailed introduction to the dataset portion.
> >
> > ---
> > **Q5:** I don't think the evaluation method in the benchmark is appropriate for the goal of the benchmark.
> >
> > **A5:** Since we use a QA format to test MLLMs' GUI understanding capabilities, we employed the LLM-as-a-Judge approach. This method compares the model's output with the benchmark's ground truth. The advantage of this judging system is that it can provide a detailed analysis of the response in relation to the ground truth through Chain-of-Thought (COT), rather than just a numerical score. Additionally, we have included BERTScore and BLEU score metrics in the appendix for further evaluation.
> >
> > ---
> > **W6.1:** As illustrated in Figure 2, the development of GUI-World is structured around a two-stage process.' Figure 2 contains no clear indication of stages - so what are the two stages
> >
> > **A6.1:** Thank you for your correction. In fact, the 0, 1, 2, and 3 on the left side are all part of Stage 1, while the Human-MLLM Collaboration on the right side is Stage 2.
> >
> > ---
> > **Q6.2:** A lot of the methodology details are described in the appendix. Would be nice if the methodology section was a bit more self-contained.
> >
> > **A6.2:** Thank you for your suggestion! We will move more of the methodology details into the main text in the next version to make the paper more solid.
> >
> > ---
> > **Q6.3:** 147 'learning preliminary for GUI content' -> ?
> >
> > **A6.3:** The 'preliminary for GUI content' mentioned here refers to a process similar to Stage 2 in LLaVA-like models, which involves training on image-text pairs. The purpose of this stage is to learn what these GUI contents represent, i.e., align text embedding with basic GUI content. Following this, instruction tuning is used to learn more complex GUI-oriented tasks, such as reasoning problems within GUI environments.
> >
> > ---
> > **Q6.4:** after finetuned -> after fine-tuning，'Empirically results' -> empirical results
> >
> > **A6.4:** Thank you for pointing this out! We will correct these grammatical errors in the next version.
> >
> > ---
> > **Q6.5:** Table 4 what is MACQ?
> >
> > **A6.5:** Thank you for pointing this out! This is a typo. It should be MCQA, which stands for multiple-choice QA.
> >
> > ---
> > **Q6.6:** Fig 5 misses Android (at least in the legend). Also - how are the elements ordered? Software/website and the other 3 seem to be intermingled.
> >
> > **A6.6:** We apologize for any confusion we may have caused. Since we obtained the Android dataset from RICO [6], there are relatively few videos from each individual software application, which doesn't provide statistically significant reference value. This figure is modeled after Mind2Web [7], with the first part showing Software and Website data, and the latter part displaying iOS, XR, and Multi-platform data. In the next version, we will add a border to better distinguish between these sections.
> >
> > ---
> > **Q7:** Problem related to Documentation.
> >
> > **A7:** We have uploaded our dataset to a Hugging Face repository and placed the documentation on how to use the dataset on GitHub. Therefore, the link points to GitHub. We apologize for any confusion this may have caused. As our dataset is ~90GB in size, we only included a portion of it in the supplementary materials.
> >
> > ---
> >
> > We greatly appreciate your feedback and will continue to contribute to the GUI field based on your suggestions. We plan to use our proposed dataset to train more powerful GUI expert models, aiming to advance the open-source community's development in the GUI domain.
> >
> > ---
> > **[6]** Rico: A Mobile App Dataset for Building Data-Driven Design Applications
> >
> > **[7]** Mind2Web: Towards a Generalist Agent for the Web

---

> > > ### Comment · Reviewer_6RQG · 2024-08-25
> > >
> > > Thanks for the response! Assuming that the authors make the changes as described in their response, they addressed most of my concerns. I'll increase my score for the paper.

---

### Official Review · Reviewer_pKiX · 2024-07-19
**A promising work with potential application but not fully realized value.**

**Rating:** 6
**Confidence:** 3
**Correctness:** Yes.
**Clarity:** Yes.

**Review:**

Overall, I believe that the GUI-World dataset proposed in this paper is valuable and comprehensive. However, GUI-vid does not fully showcase the value of this dataset or its potential in enabling MLLMs to understand GUI tasks. This might be due to the base model, but it is necessary to switch to a more advanced open-source base model. Therefore, I think the current version of the paper is slightly below the acceptance threshold.

**Strengths:**

1. GUI-WORLD is an extensive GUI dataset with a broader coverage compared to existing datasets.
2. The GUI-Vid model demonstrates significant performance improvements in handling complex GUI tasks compared to the base model.
3. The paper is well-structured and clearly written.

**Additional Feedback:**

N/A

**Documentation:**

The introduction of the dataset is sufficient.

**Limitations:**

The authors have mentioned limitations and discussed them thoroughly. In my opinion, some of these limitations are necessary and urgently need to be addressed.

**Opportunities For Improvement:**

1. The reliance on manual keyframe selection instead of automatic extraction might lead to suboptimal performance in practical applications. There are advanced Keyframe Identifier methods under multimodal LLMs, such as KISA[1], which could be referenced and the rationale for not using such advanced methods should be explained.
2. Although the GUI-Vid model is introduced, its performance is still limited by the base LLM, lagging behind industry-leading models. Why not adopt advanced multimodal large models as the base model? For instance, models like [2][3] are fine-tuned on LLaMA for video understanding; [4] is fine-tuned on LLaVa for understanding embodied operation videos. A thorough discussion is necessary.
3.There are some typos in the manuscript that need careful checking and correction, such as the inappropriate capitalization in line 159: "As illustrated in Figure 4, We."

[1] Kou, Longxin, et al. "KISA: A Unified Keyframe Identifier and Skill Annotator for Long-Horizon Robotics Demonstrations." Forty-first International Conference on Machine Learning.

[2] Lin, Bin, et al. "Video-llava: Learning united visual representation by alignment before projection." arXiv preprint arXiv:2311.10122 (2023).

[3] Zhang, Hang, Xin Li, and Lidong Bing. "Video-llama: An instruction-tuned audio-visual language model for video understanding." arXiv preprint arXiv:2306.02858 (2023).

[4] Liu, Jinyi, et al. "Enhancing Robotic Manipulation with AI Feedback from Multimodal Large Language Models." arXiv preprint arXiv:2402.14245 (2024).

**Relation To Prior Work:**

The choice of base model did not adequately reference existing work.

**Summary And Contributions:**

This paper presents GUI-WORLD, a comprehensive GUI dataset containing over 12,000 videos, designed to evaluate and enhance the GUI understanding capabilities of multimodal large models (MLLMs). Based on GUI-WORLD, GUI-Vid is proposed as a multimodal large model for GUI tasks. Experiments indicate that most existing MLLMs still face challenges when dealing with GUI tasks, particularly with sequential and dynamic GUI content. Empirical results show that improving visual perception, increasing the number of keyframes, and enhancing image resolution can improve the performance of GUI tasks, providing valuable insights for future GUI agent research.

---

> ### Author Rebuttal · Authors · 2024-08-15
>
> Thank you very much for your valuable feedback. We will address each of your concerns and provide explanations and additional experiment results to help you better understand the contributions of this paper step by step:
>
> ---
> **Q1:** GUI-Vid does not fully showcase the value of this dataset or its potential in enabling MLLMs to understand GUI tasks. This might be due to the base model, but it is necessary to switch to a more advanced open-source base model.
>
> **A1:** Thank you for recognizing the value of our dataset. As the first dataset to focus on the GUI video domain and address dynamic GUI content in a benchmark, we have tried to demonstrate its effectiveness, proposing the first expert open-source VideoLLM in the GUI domain. Despite the limitations of our base model, our trained model has shown promising results. While it may not match the performance of Gemini-Pro or GPT-4, it has outperformed the closed-source model Qwen-VL-Max, which significantly validates the quality of our dataset.
>
> We are aware of the recent emergence of new VideoLLM models **[1] [2]**. In light of this, we plan to utilize more advanced and powerful models as our base for fine-tuning in our next version, to better showcase the full potential and value of our dataset.
>
> As pioneers in this specific area of GUI video analysis and dynamic content benchmarking, we remain committed to pushing the boundaries of what's possible in this field. Our ongoing efforts aim to further enhance the capabilities of AI models in understanding and interpreting complex, dynamic GUI interactions.
>
> ---
> **Q2.1:** The reliance on manual keyframe selection instead of automatic extraction might lead to suboptimal performance in practical applications.
>
> **A2.1:** The number of human-selected keyframes is fewer than automatic extraction frames, as shown in Table 2 in the manuscript. This potentially results in less information compared to random selection, leading to inconsistent performance. Thank you for your suggestion and we will continue to find replacement methods for manual keyframe selection while keeping competitive results.
>
> ---
> **Q2.2:** There are advanced Keyframe Identifier methods under multimodal LLMs, such as KISA **[3]**, which could be referenced and the rationale for not using such advanced methods should be explained.
>
> **A2.2:** Regarding the other Keyframe Identifier methods you mentioned, KISA has not yet open-sourced its code. Additionally, while some papers have explored aggregating keyframe selection processes with LLMs **[2]**, they too have not released their code. Consequently, we plan to update our paper's next version with a more comprehensive discussion on keyframe identifiers. We will also conduct experiments using these alternative keyframe selection methods once their code becomes available.
>
> ---
> **Q3.1:** Although the GUI-Vid model is introduced, its performance is still limited by the base LLM, lagging behind industry-leading models.
>
> **A3.1:** During our submission period, new VideoLLMs emerged. We evaluated two latest models,  LLaVA-Next-Video-7B-DPO **[1]** and  Video-LLaVA **[4]**. We show average performance compared to GUI-Vid in **Table 1**, with details shown in **PDF**. Our model outperformed these in most tasks, except Conversation, likely due to their use of DPO during training. We'll report on Video-LLaMA **[5]** in our next version due to code reproducibility issues.
>
> **Table 1: Performance Comparison of Recent Models on GUI-Oriented Tasks**
>
> | Model       | MCQA | Description | Conversation | Dynamic | Static | Caption |
> |-------------|----------|-------------|--------------|---------|--------|---------|
> | LLaVA-Next  | 0.471 | 1.644 | **3.169** | 2.464 | 1.961 | 2.148
> | Video-LLaVA | 0.494    | 1.156       | 2.911        | 2.005   | 1.722  | 1.622   |
> | **GUI-Vid (Ours)**     | **0.546**    | **1.911**       | 3.070        | **2.755**   | **2.331**  | **3.427**   |
>
> ---
>
> **Q3.2:** A thorough discussion for VideoLLMs.
>
> **A3.2:** Thank you for your suggestion. We will test the zero-shot performance of more recent VideoLLMs on our benchmark, such as the latest LLaVA-Next-Video-32B, and also conduct fine-tuning experiments. We will include a discussion of these in the related works section.
>
> ---
>
> **W4:** Typos in the manuscript.
>
> **A4:** Thank you for your thorough review. We will correct these typos in the next version.
>
> ---
>
> We greatly appreciate your insights and will continue to contribute to the GUI field based on your suggestions. We plan to use our proposed dataset to train more powerful GUI expert models, aiming to advance the open-source community's development in the GUI domain.
>
> ---
> **[1]** https://llava-vl.github.io/blog/2024-04-30-llava-next-video/
>
> **[2]** KeyVideoLLM: Towards Large-scale Video Keyframe Selection
>
> **[3]** KISA: A Unified Keyframe Identifier and Skill Annotator for Long-Horizon Robotics Demonstrations
>
> **[4]** Video-LLaVA: Learning United Visual Representation by Alignment Before Projection
>
> **[5]** Video-llama: An instruction-tuned audio-visual language model for video understanding

---

> > ### Comment · Reviewer_pKiX · 2024-08-17
> >
> > Thank you to the authors for their detailed response.
> >
> > I acknowledge new results mentioned in the authors' reply. For MCQA, the newly added results, specifically those related to LLaVA-Next and Video-LLaVA, are indeed significantly better than the previously recorded results for Minigpt4Video and VideoChat2. However, this still doesn't address my concern: how much better would the results be if fine-tuning were based on Video-LLaVA compared to the current GUI-vid? Could it be that the fine-tuning gains are minimal due to the superior performance of the base model? As the authors mentioned, utilizing more advanced and powerful models for fine-tuning to better showcase the full potential and value of our dataset is indeed necessary and important.
> >
> > I also acknowledge the author's commitment that the revised version will address typos and add discussions on related work. I would like to note that related work is not limited to what I originally listed. A more comprehensive discussion is always better.
> >
> > Furthermore, regarding the Keyframe Identifier method, my concern remains, specifically that manually selecting keyframes may result in suboptimal performance in real-world applications. This is important, but it may be challenging to resolve within a short timeframe of one or two weeks.
> >
> > I look forward to the author's further response. My core concerns have not yet been fully addressed, so I will refer to the opinions and discussions of other reviewers before making a final decision on whether to adjust the score.

---

> > > ### Author Response · Authors · 2024-08-17
> > > **Official Comment by Authors**
> > >
> > > Thank you for the timely response!
> > >
> > > We will try to fine-tune Video-LLaVA as you suggested in the following week to prove further the high quality and potential usage of our dataset.
> > >
> > > As for the keyframe identifier method, we will try to use the second-best method mentioned in KISA (because KISA hasn't published its code).
> > >
> > > Thanks for the valuable suggestion. Hopefully, our additional experiment can meet your expectations.

---

> > > ### Author Response · Authors · 2024-08-25
> > > **Official Comment by Authors**
> > >
> > > **Q2:** Try to fine-tune on Video-LLaVA.
> > >
> > > **A2:** Due to limited computational resource (we only have access to two A800 GPUs), we can only perform LoRA fine-tuning on Video-LLaVA rather than full SFT. Additionally, we could only set a global batch size of 4, resulting in nearly 24 hours of training time for a single epoch on GUI-World. Although aligning our fine-tuning hyperparameters with the authors' recommendations, we encountered some challenges. The use of LoRA led to a degree of catastrophic forget in the model, in both the one-epoch and two-epoch checkpoints.
> > >
> > > While we made our best efforts to conduct these experiments, we are sorry and unable to complete this portion of the study due to these limitations. In the future, when we have access to more computational resources, we intend to revisit this by fine-tuning the most effective videoLLM models available, such as the recently released LLaVA-OneVision-7B or even larger 72B models.
> > >
> > > ---
> > > Finally, we would like to express our sincere gratitude for your suggestions regarding our paper. Your input has significantly contributed to enhancing the quality of our work. We hope that our responses have addressed your concerns. If you have any further questions, please don't hesitate to ask.

---

> > > > ### Comment · Reviewer_pKiX · 2024-08-26
> > > >
> > > > Thank you for promptly improving the experiment.
> > > >
> > > > I appreciate the authors' further research and validation of automated keyframe identifier methods. The authors' supplementary experimental conclusion that Automated Keyframe Identification Methods can partially replace the manual annotation process enriches this paper, making it more coherent and complete.
> > > >
> > > > Moreover, fine-tuning on more advanced and powerful foundational MLLMs to more comprehensively showcase the value of the dataset is crucial. I look forward to seeing these conclusions in future versions.
> > > >
> > > > Overall, the authors' supplementary experiments and their explanations of my concerns are sufficient for me to raise the score above the borderline, so I will increase the score to 6. I look forward to the authors enhancing the details mentioned in the discussion phase (the automated Keyframe Identifier methods, multimodal LLM fine-tuning related papers, new experimental results, etc.) and continuing to improve the paper.

---

> > > > > ### Author Response · Authors · 2024-08-26
> > > > > **Official Comment by Authors**
> > > > >
> > > > > Thank you for raising the score!
> > > > >
> > > > > Your feedback indeed helped improve our work, and it was a pleasure to engage in multiple rounds of discussion about our paper. We will continue to refine our work based on your insightful suggestions.
> > > > >
> > > > > Thank you again for your time and valuable comments. We are happy to have had you as a reviewer.

---

> > ### Author Response · Authors · 2024-08-25
> > **Official Comment by Authors**
> >
> > Thank you for your patience. During this time, we have attempted to implement two new keyframe identifier methods and fine-tune Video-LLaVA. I will present our additional experiment results step-by-step:
> >
> > ---
> > **Q1:** Other keyframe identifier methods.
> >
> > **A1:** As of now, the code for KISA **[1]** has not been released. Consequently, we opted to implement the previous state-of-the-art methods mentioned in the KISA paper, specifically UVD **[2]**. We implemented two settings from UVD: UVD + VIP **[3]** and UVD + R3M **[4]**, which were reported as the top-performing keyframe identifier methods in their paper.
> >
> > During our implementation, we observed that these methods sometimes extracted more than 10 keyframes for some videos. However, due to GPT's image input limitations (which allows a maximum of 10 images) and to maintain the fairness of our benchmark , we performed a skip selection for videos keyframes to 10 keyframes. For videos with fewer than 10 keyframes selected by these methods, we maintained consistency with our previous benchmark setting by including the first and last frames to reach the 10-frame input, thus ensuring the robustness of our experiments.
> >
> > The overall experimental results are presented in **Table 1**, with detailed results shown in **Tables 2 3**. The new keyframe identifier methods indeed brought some performance improvements, particularly in static and description tasks. However, for dynamic tasks, the performance still falls short of human selection. These findings suggest that keyframe identifier methods represent a promising direction for future research and have the potential to replace human effort in certain scenarios.
> >
> > This investigation into automated keyframe selection methods has provided valuable insights into GUI field on how to replace human effort in selecting video keyframes. We appreciate your constructive suggestion, as it has led us to explore this important aspect of GUI analysis and has contributed significantly to our research.
> >
> > ---
> > **Table 1: Average Performance of GPT-4o among 6 scenarios: Two Automated Keyframe Identification Methods vs. Human-Selected Keyframes.**
> > | Setting | MCQA | Description | Conversation | Dynamic | Static | Caption | Average |
> > |---------|------|-------------|--------------|---------|--------|---------|---------|
> > | Human   | **0.848**| 3.031       | 4.056        | 3.318   | 3.131  | 3.911   | 3.573   |
> > | UVD+VIP | 0.835| 3.150       | 4.044        | 3.265   | 3.346  | 3.923   | 3.581   |
> > | UVD+R3M | 0.845| **3.136**       | **4.058**        | **3.292**   | **3.363**  | **3.940**   | **3.612**   |
> >
> > **Table 2: Detailed Performance of GPT-4o using UVD+ViP Keyframe Identification Method.**
> > | Scenario | MCQA | Description | Conversation | Dynamic | Static | Caption | Average |
> > |----------|------|-------------|--------------|---------|--------|---------|---------|
> > | Software | 0.862| 3.297       | 4.282        | 3.354   | 3.478  | 4.112   | 3.749   |
> > | Website  | 0.820| 3.248       | 4.155        | 3.415   | 3.567  | 4.074   | 3.744   |
> > | XR       | 0.842| 2.980       | 3.775        | 3.034   | 3.122  | 3.587   | 3.347   |
> > | Multi    | 0.821| 3.391       | 4.165        | 3.466   | 3.404  | 3.868   | 3.659   |
> > | IOS      | 0.860| 3.157       | 4.017        | 3.353   | 3.492  | 4.050   | 3.648   |
> > | Android   | 0.807| 2.827       | 3.871        | 2.970   | 3.014  | 3.844   | 3.340   |
> > | Average  | 0.835| 3.150       | 4.044        | 3.265   | 3.346  | 3.923   | 3.581   |
> >
> > **Table 3: Detailed Performance of GPT-4o using UVD+R3M Keyframe Identification Method.**
> > | Scenario | MCQA | Description | Conversation | Dynamic | Static | Caption | Average |
> > |----------|------|-------------|--------------|---------|--------|---------|---------|
> > | Software | 0.858| 3.290       | 4.273        | 3.352   | 3.458  | 4.134   | 3.741   |
> > | Website  | 0.827| 3.282       | 4.114        | 3.460   | 3.591  | 4.065   | 3.746   |
> > | XR       | 0.877| 3.010       | 3.861        | 3.142   | 3.161  | 3.600   | 3.433   |
> > | Multi    | 0.836| 3.237       | 4.129        | 3.503   | 3.417  | 3.897   | 3.737   |
> > | IOS      | 0.864| 3.165       | 4.094        | 3.328   | 3.480  | 4.078   | 3.663   |
> > | Android  | 0.806| 2.835       | 3.876        | 2.968   | 3.072  | 3.865   | 3.353   |
> > | Average  | 0.845| 3.136       | 4.058        | 3.292   | 3.363  | 3.940   | 3.612   |
> >
> > ---
> >
> > **[1]** KISA: A Unified Keyframe Identifier and Skill Annotator for Long-Horizon Robotics Demonstrations
> >
> > **[2]** Universal Visual Decomposer: Long-Horizon Manipulation Made Easy
> >
> > **[3]** VIP: Towards Universal Visual Reward and Representation via Value-Implicit Pre-Training
> >
> > **[4]** R3M: A Universal Visual Representation for Robot Manipulation

---

### Official Review · Reviewer_Mxkp · 2024-07-25
**This paper presents a good dataset for GUI agent**

**Rating:** 6
**Confidence:** 3
**Correctness:** Yes
**Clarity:** Yes

**Review:**

The idea of this work is good, introducing temporal information and covering a wide range of scenarios.  The paper is well-written and should provide a good resource for GUI agents.

**Strengths:**

- Good idea: The paper points out that a robust GUI agent should be capable of perceiving temporal information, and introduces this dataset for dynamic GUI content understanding.
- Rich scenarios: The dataset includes Windows, macOS, Linux, iOS, Android, and XR environments, which is comprehensive.
- Easy to read: The paper is well-structured, giving detailed figures and tables to show the content of the dataset.

**Additional Feedback:**

NA

**Documentation:**

Yes

**Limitations:**

Yes

**Opportunities For Improvement:**

- Inconsistent experimental setups: In Table 4 and Table 5, why the settings of Gemini-Pro-1.5, Qwen-VL-Max, GPT-4V, and GPT-4o is inconsistent. GPT-4o only has human-selected keyframes (H.), but GPT-4V has six different settings (R., E., H., D.C., C.C., and H. + D.C). What is the reason for this inconsistency? I think an aligned setting would be better.
- The necessity of dynamic understanding: I agree that understanding dynamic content is important, but how can you prove that dynamic content understanding is necessary in your dataset?  The results in Table 4 and Table 5 show that the ImageLLMs (using fewer keyframes) do better than VideoLLMs (using more keyframes). Btw, I think there seems no difference between the VideoLLMs and ImageLLM: all their inputs are keyframes, and the numbers of keyframes seem at the same level of magnitude.

**Relation To Prior Work:**

Yes

**Summary And Contributions:**

This paper introduces a new dataset, GUI-World, which covers six GUI scenarios (Windows, macOS, Linux, iOS, Android, and XR) and eight types of GUI-orientated tasks. An important feature of this dataset is that the inputs are videos, and the authors fine-tune a VideoLLM as the GUI agent to understand dynamic GUI content.

---

> ### Author Rebuttal · Authors · 2024-08-15
>
> We would like to express our sincere gratitude for your valuable feedback. We will address each of your concerns and provide explanations step by step:
>
> ---
>
> **Q1:** Inconsistent experimental setups.
>
> **A1:** In our experimental setting, experiments involving GPT-4V that include C (Caption) are ablation studies, which aim to verify whether providing detailed image descriptions in the text prompt would be beneficial for GUI-oriented tasks. Thank you for your suggestion and we plan to present the ablation study results in a separate table to minimize potential misunderstandings.
>
> ---
>
> **Q2.1:** How to prove that dynamic content understanding is necessary in your dataset?
>
> **A2.1:** In our article, we define "dynamic" content as elements that change between consecutive frames. When designing our QA system, we addressed similar issues by selecting pairs of frames from human-annotated keyframes that are similar but not identical. This approach ensures that our dataset includes examples of dynamic content. Regarding the significant performance gap between ImageLLM and VideoLLM, we believe the primary reason is that the open-source VideoLLM uses a less powerful LLM backbone compared to proprietary models. Consequently, even with additional keyframe inputs, VideoLLM's performance falls short of ImageLLM's capabilities.
>
> ---
>
> **Q2.2:** Difference between VideoLLMs and ImageLLMs.
>
> **A2.2:** Current mainstream VideoLLMs indeed differ from ImageLLMs primarily in their ability to process a larger number of input images. Our experiments in manuscript have also demonstrated that using more GUI keyframes leads to better performance. In fact, within the GUI domain, short video clips have been proven to better represent the current state compared to static images **[1] [2]**. Therefore, our dataset serves as a significant contribution in encouraging GUI assistants to utilize short video clips instead of still photos.
>
> ---
>
> We greatly appreciate your valuable insights and will continue to contribute to the GUI field based on your suggestions. We plan to use our proposed dataset to train more powerful GUI expert models, aiming to advance the open-source community's development in the GUI domain.
>
> ---
> **[1]** Cradle: Empowering Foundation Agents Towards General Computer Control
>
> **[2]** AgentStudio: A Toolkit for Building General Virtual Agents

---

> > ### Author Response · Authors · 2024-08-30
> > **Official Comment by Authors**
> >
> > Dear Reviewer Mxkp,
> >
> > We are thankful for your review. As the rebuttal deadline is coming to an end, please let us know if your concerns are well addressed. We are happy to provide further clarification.

---

### Decision · Program_Chairs · 2024-09-26

**Decision:**

Reject

**Comment:**

The dataset comprises diverse GUI scenarios (Windows, macOS, Linux, iOS, Android, and XR) and various tasks, which can benchmark the performance of different MLLMs on GUI tasks. However, I am concerning the necessity and the relevance of the proposed benchmark.  The constructed dataset aims to pushing the ablity of MLLM on GUI-world. Howver, GPT-4o seems to work quite well on the benchmark dataset, as such the difficulty and contributions of the dataset for helping MLLM seem to be not very essential. More, the meaningful task for GUI should be GUI-based manipulation and related task.  while the captioning task for general MLLM seems to be not really important for a GUI-related task. The GUI-related tasks should be more carefully considered and desigend.
As such, I am recommending rejection for this submission.